# A meta learning and task adaptive approach for drug target affinity prediction

Mengxuan Wan[1,2,4], Yanpeng Zhao [1,4], Yixin Zhang[2,4], Huiyan Xu[2,3], Duoyun Yi[2,3], Peng Zan [1,3] ✉, Song He [2] ✉ & Xiaochen Bo [2] ✉

Accurate and robust prediction of drug-target affinity (DTA) plays a critical role in drug discovery. While deep learning has advanced DTA prediction, existing methods struggle with limited training data and poor generalization. In this study, we propose AdaMBind, a novel DTA prediction model based on meta-learning framework with an adaptive task module designed for low-data scenarios. It employs a dynamic "easy-to-hard" task scheduling mechanism to enhance training efficiency and robustness. Experimental results on three benchmark datasets demonstrate that AdaMBind outperforms 8 baseline models in predicting affinity for unseen targets, particularly under few-shot conditions. Under stringent data constraints, the model successfully identifies high-affinity compounds for ESR and TP53, achieving outstanding virtual screening performance. Furthermore, when applied to inhibitor discovery against FLT3 for acute myeloid leukemia, AdaMBind successfully identified candidate compounds with potent inhibitory activity, as verified by preliminary experimental assays. In summary, AdaMBind provides a robust framework for few-shot DTA prediction.

The binding of a drug to its target to modulate biological functions is a fundamental mechanism of therapeutic interventions[1]. Drug-target affinity (DTA), which quantitatively characterizes the interaction strength between drugs and targets, serves as a critical parameter guiding the rational design and optimization of therapeutic agents[2]. Consequently, accurate DTA determination is essential in both de novo drug discovery and drug repurposing strategies.

Conventional experimental methodologies for DTA determination, including isothermal titration calorimetry (ITC) and surface plasmon resonance (SPR), are inherently constrained by substantial time requirements, exorbitant costs, and technical complexities[3,4]. In contrast, computational prediction approaches demonstrate superior efficiency and cost-effectiveness, offering several orders of magnitude acceleration in throughput. Although in silico approaches cannot fully replace experimental validation, they have become indispensable for preliminary screening and prioritizing candidates, thereby accelerating the drug discovery pipeline[5]. Among these, Molecular docking is widely used as a key computational method for predicting binding modes and interaction strengths[6,7]. However, its practical application is often limited by high computational complexity and the limited availability of 3D structural data.

In recent years, deep learning (DL) -based approaches have emerged as pivotal tools for DTA prediction. Current methods can be mainly categorized into three classes. The first class comprises sequence-based approaches, which extract representations directly from the amino acid sequences of protein and drug SMILES strings. For instance, MFR-DTA[8] employed multiple feature extractors to process SMILES and protein sequences, capturing both global and local sequence features. Similarly, Co-VAE[9] employed Gated Convolutional Neural Networks (GatedCNNs) to extract sequence features from drugs and proteins, followed by a variational autoencoder (VAE) to learn latent representations that were subsequently fused and decoded for affinity prediction. TEFDTA[10] leveraged a Transformer module to extract drug features from MACCS fingerprints and then applied a

[1]School of Medicine, Shanghai University, Shanghai, China. [2]Academy of Military Medical Sciences, Beijing, China. [3]Shanghai Key Laboratory of Power Station Automation Technology, School of Mechatronics Engineering and Automation, Shanghai University, Shanghai, China. [4]These authors contributed equally: Mengxuan Wan, Yanpeng Zhao, Yixin Zhang. ✉e-mail: zanpeng@shu.edu.cn; hes1224@163.com; boxc@bmi.ac.cn

Multilayer Perceptron (MLP) regressor to predict binding affinity. PSICHIC[11] introduced physicochemical constraints into sequence-based modeling via graph neural networks, enhancing interpretability and generalization while maintaining broad applicability. The second category consists of graph-based methods, where the atoms and chemical bonds in drug molecules are represented as nodes and edges in a graph, and protein information is predominantly derived from their amino acid sequences. For instance, ColdDTA[12] utilized graph neural networks (GNNs) for drug and 1D convolutional layers for protein sequences, incorporating an attention-based feature fusion module to enhance drug-target interaction modeling. TransVAE-DTA[13] combined a VAE with the Transformer architecture to more effectively capture long-range dependencies in drug-target interactions. The third class comprises pocket-based approaches, in which 3D protein pocket structures are integrated with 2D molecular graphs. For instance, AttentionMGT-DTA[14] employed attention mechanisms to integrate drug-target interactions using pocket and molecular graph representations. CSCo-DTA[15] enhanced prediction by incorporating protein 3D structures, molecular 2D graphs, and protein-ligand interaction networks via contrastive learning. HiSIF-DTA[16] further improved accuracy by integrating protein-protein interaction (PPI) data into protein representations. Despite these advances, pocket-based approaches remain limited when applied to novel or understudied targets that lack experimentally determined structures or well-defined binding pockets. Although some studies have attempted to mitigate this issue by leveraging AlphaFold-predicted structures[17–19], challenges persist for proteins with high conformational flexibility (e.g., membrane proteins or intrinsically disordered proteins), where dynamic binding sites remain difficult to resolve.

Although DL has made significant advances in DTA prediction, its application in real-world scenarios still faces two critical bottlenecks. First, most existing models generally struggle to learn target representations with limited samples. In real-world scenarios, a large number of targets remain insufficiently characterized, resulting in an extreme sparsity of known drug-target interaction data. This data scarcity severely restricts the capacity of models to extract robust features, rendering effective modeling for such targets a major obstacle for current approaches. As a result, effectively modeling such drug-target pairs remains a major bottleneck for current approaches. Second, mainstream approaches often lack mechanisms for target-specific adaptation or meta-knowledge transfer. In practical drug discovery, models are frequently required to predict affinities for previously unseen targets. Such tasks are not only highly target-specific but also demand strong cross-target generalization capabilities of the models. Therefore, a critical practical challenge arises: when only a minimal amount of experimental data is available for a novel target, how to enable the model to leverage such limited information to rapidly develop a target-specific understanding and deliver reliable affinity predictions. This few-shot adaptation capability holds pivotal practical significance for accelerating the early-stage drug discovery pipeline targeting emerging or understudied targets.

Meta-learning, also referred to as "learning to learn", has emerged as a promising framework to address these challenges. Unlike conventional training that merely fits data distributions of specific tasks, meta-learning enables models to acquire shared and transferable "meta knowledge" across diverse tasks. Consequently, when encountering a novel target with extremely limited data, the model can leverage the previously learned meta-knowledge to rapidly fine-tune and adapt using only a few samples, thereby enabling effective prediction of affinities for new targets. Within this paradigm, datasets are organized into sub-tasks. Each sub-task centers on a specific protein and includes all known drugs that interact with it, along with their corresponding affinity values. Recently, some meta-learning-based methods have been proposed for DTA prediction. Contrastive Meta-Learning (CML)[20] builds on the Model Diagnostic Meta-Learning

(MAML) framework by incorporating a contrastive learning module in the forward propagation phase of the query set in order to enhance cross-task relevance exploration for improved adaptation to new tasks. MetaDTA[21] employed attention-based neural processes, enabling the model to refine its predictions for query samples by leveraging information from the support set. ZeroBind[22] formulated drug-target interaction (DTI) prediction as a target-specific meta-learning task, utilizing a Subgraph Information Bottleneck (SIB) module to identify potential binding pockets and employing a task-adaptive self-attention mechanism to assign different weights to tasks within the loss function. Although these methods have shown promising progress, several limitations remain. First, MetaDTA and CML typically assume all tasks are equally important and sample meta-training tasks uniformly at random, overlooking differences in task difficulty. This may lead to the model being dominated by noisy or outlier tasks, thereby impairing overall generalization. While ZeroBind mitigated this issue by dynamically adjusting task weights through the task adaptive self-attention mechanism, its reliance on a static sampling strategy still leaved room for potential interference from noisy tasks during model optimization. Secondly, these methods rely on modeling inter-task correlations. When adapting to a new task, the lack of highly similar training tasks may impair the model's generalization ability. Therefore, it is essential to develop a meta-learning framework that can dynamically identify and prioritize high-value tasks, suppress the influence of noisy tasks, and enable robust generalization to unseen targets under limited data conditions.

Inspired by recent advances in multi-task and meta-learning, a widely accepted view has emerged: not all tasks contribute equally to model training, especially when the data include noisy tasks or are unevenly distributed[23,24]. In such cases, uniform random sampling may mislead the learning process and reduce the model's generalization ability[25]. Against this background, a more effective learning strategy has emerged. The strategy begins with easier tasks and gradually shifts to more challenging ones, following an easy-to-hard approach[26,27], which reflects human learning behavior and can improve both the efficiency and stability of training under limited data conditions. This idea provides valuable insight into the design of adaptive meta-learning frameworks for DTA prediction.

In this work, we introduce AdaMBind, a meta-learning framework with the adaptive task module, designed for accurate affinity prediction between drugs and unseen targets. AdaMBind integrates three key components: a meta-learning module built on the MAML strategy, an adaptive task module, and a label noise injection strategy. This meta-learning module enables AdaMBind to rapidly adapt to previously unseen targets under data-scarce conditions by learning transferable initialization parameters across training tasks. The adaptive task module dynamically prioritizes high-value training tasks by their loss and gradient similarity, while the label noise strategy enhances model robustness and prevents overfitting to noisy affinity annotations. AdaMBind is trained and validated on three widely used benchmark datasets: BindingDB, KIBA, and Davis. Under the meta-learning paradigm, the model is trained on target-specific tasks, each consisting of a specific target along with its corresponding drug and affinity. We evaluate the performance of AdaMBind under different task split methods and different support set size settings. The experimental results show that AdaMBind outperforms the existing eight baseline models in most evaluation metrics, with its advantage being especially pronounced in the few-shot setting. By investigating the operational mechanism of the adaptive task module, we demonstrate that it follows an "easy-to-hard" learning strategy, dynamically prioritizing tasks with lower loss and higher gradient consistency during meta-training, thereby substantially improving learning efficiency and model generalization. We further design a strict few-shot virtual screening experiment on the LIT-PCBA dataset for the ESR and TP53 targets under a limited number of meta-tasks. AdaMBind achieves

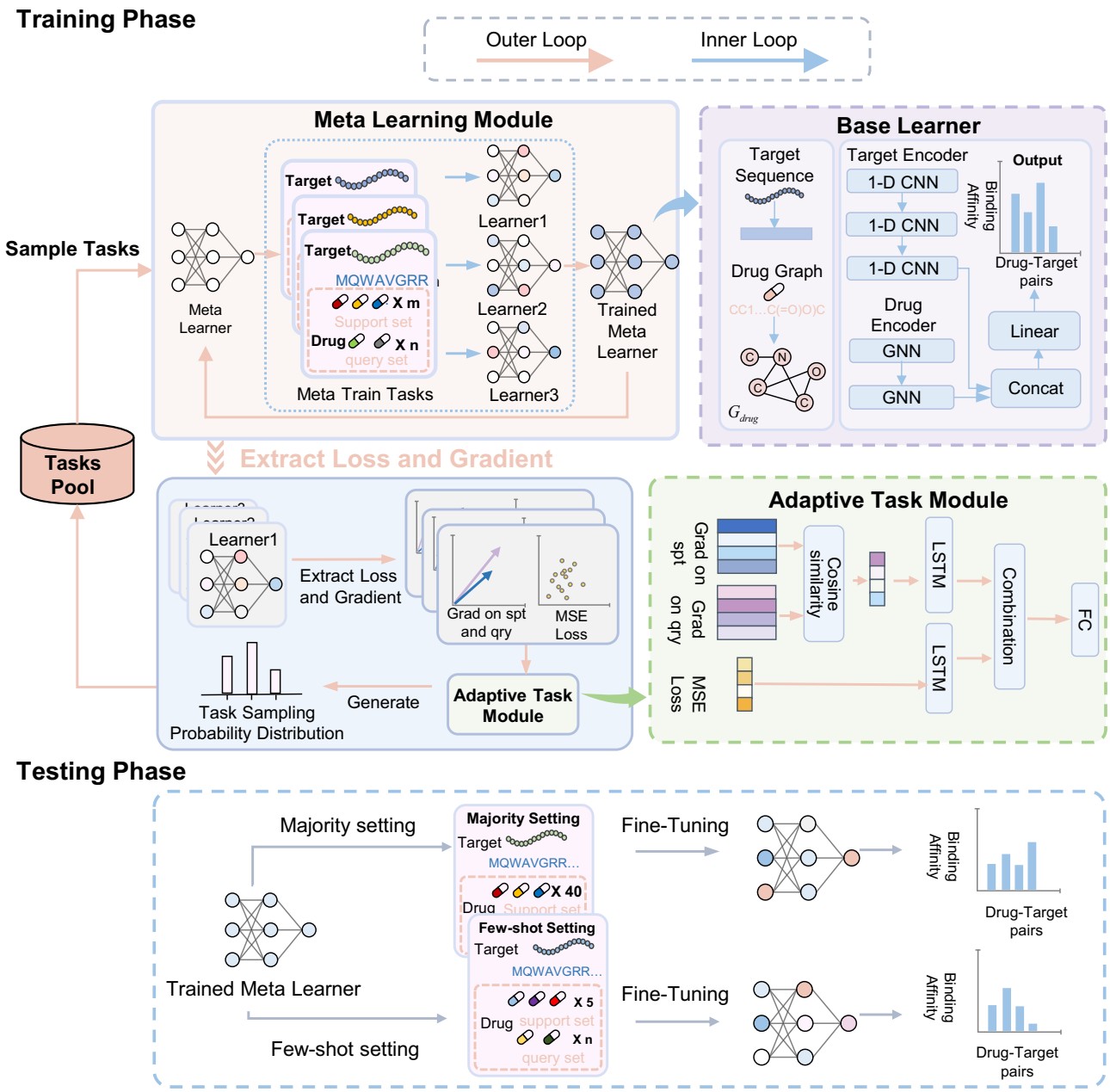

**Fig. 1 | The architecture of AdaMBind.** AdaMBind takes molecular graphs of drugs and amino acid sequences of target proteins as input, and outputs the predicted binding affinity. In the training phase, the data are first divided into multiple sub-tasks based on different targets, with each task containing a set of drugs that interact with the target and their corresponding affinity values. Then, the adaptive task module samples tasks based on their importance scores to construct the meta-training set. During the testing phase, the meta-learner is first fine-tuned on the support set of the testing task, employing either a majority setting (40 DTA pairs in support set) or a few-shot setting (5 DTA pairs in support set), and is subsequently deployed to predict the binding affinity of unknown drug-target pairs, and then used to predict drug-target binding affinities.

outstanding virtual screening performance. Finally, we apply AdaM-Bind to identify potential therapeutic drugs for acute myeloid leukemia (AML) from the DrugBank database. Among the top 20 predicted compounds ranked by affinity, half are supported either by AutoDock docking simulations or by previous literature. Notably, Staurosporine is experimentally validated to inhibit FLT3 activity.

## Results
### Overview of AdaMBind
In this study, we propose a novel DTA prediction model named AdaMBind. By integrating a meta-learning strategy with an adaptive task module, the model aims to solve the problem of effectively extracting drug-target affinity information under limited data and

achieve accurate predictions for unseen targets. Figure 1 illustrates the AdaMBind framework, comprising three modules: a meta-learning module based on the MAML strategy, an adaptive task module, and a label noise strategy.

During the training phase, the raw data are first partitioned into multiple sub-tasks. Each task consists of a unique protein, together with its corresponding interacting drugs and their binding affinities. To prevent the model from being disturbed by noisy tasks during training, the adaptive task module dynamically evaluates the value of each task and performs task sampling. The sampling weights are determined by two key factors: the loss of each task on the query set, and the gradient similarity between the support set and the query set for each task. After sampling, the meta-learning module conducts meta-training

based on the selected tasks. The procedure involves a bi-level optimization loop: in the inner loop, each task-specific base learner is initialized with a set of shared meta-initialization, and undergoes several gradient descent updates on its task-specific support set to achieve rapid adaptation of parameters, followed by performance evaluation on the query set. The outer loop aggregates the losses generated on the query set across all selected tasks, and uses their sum as the objective function to update the meta-initialization via gradient descent. The core objective of this process is to guide the meta-learner to learn a set of initialization parameters that enable fast adaptation to novel tasks, by repeatedly simulating this "rapid adaptation–evaluation" paradigm across multiple tasks. To enhance the model robustness in real-world scenarios and prevent overfitting, we employ a regularization technique that injects noise into the affinity labels during training.

During the testing phase, for an unseen target task, the model initializes the base learner with parameters from the trained meta-learner. Then rapidly adapted the parameters through a few gradient steps on the test task's support set. To simulate different levels of data scarcity in real applications, we evaluated the model under two support set sizes, including a few-shot setting (5 DTA pairs) and a majority setting (40 DTA pairs). Finally, the fine-tuned base learner was employed to predict the affinity values for drug-target pairs in the query set of the test task. For specific details on the model, please see the Methods section.

## Performance comparison of Baselines and AdaMBind under random task split

To address two critical challenges encountered in practical drug discovery: generalization capability for novel targets and learning ability under data-scarce conditions. We developed a systematic evaluation protocol to comprehensively assess AdaMBind's performance in drug target prediction. Our experimental framework was structured hierarchically across two complementary levels. First, two task splitting strategies were employed to examine model generalization: (i) the random task split, in which all tasks were randomly divided into meta-training, meta-validation, and meta-test sets; and (ii) the novel task split based on sequence similarity, where tasks were partitioned based on target-protein sequence similarity using CD-HIT with a stringent 40% sequence-identity threshold. Within each of the above task split scenarios, we further investigated model robustness under two levels of per-task data availability: (i) majority setting, each task contains 40 known DTA pairs in its support set, and (ii) few-shot setting, each task was provided with only 5 known DTA pairs in its support set.

We compared AdaMBind with several baseline methods. These methods encompassed different learning paradigms, including non-meta-learning approaches DeepDTA[28], HiSIF-DTA[16], ColdDTA[12], Co-VAE[9] and PSICHIC[11], as well as meta-learning approaches CML[20], MetaDTA[21], ZeroBind[22]. All models were trained and tested on three benchmark datasets: BindingDB, KIBA, and Davis.

To ensure a fair comparison, we followed the approach of Wang et al. and set aside the support set of each meta-test task as a separate fine-tuning dataset[29]. All non-meta-learning baseline models were fine-tuned on these datasets for 5 steps before making predictions. The results under these two splitting schemes were analyzed in detail below.

The result under random task split is shown in Fig. 2 (Supplementary Table 10–15). As shown in Fig. 2a, c, AdaMBind demonstrated outstanding performance under the majority setting. Specifically, on the KIBA dataset, AdaMBind achieved a significant improvement over the best baseline. The MSE was reduced by 11.76% (from 0.4868 to 0.4295), the Consistency Index (CI) was improved by 6.05% (from 0.7115 to 0.7545), and the $R^2$ reached 0.4285, while the Spearman and Pearson correlation coefficients (PCC) increased by 17.82% (from 0.5541 to 0.6528) and 10.05% (from 0.5960 to 0.6559), respectively.

This performance surpassed all other comparative methods, including HiSIF-DTA, which relies on protein structural representation. On the BindingDB dataset, AdaMBind achieved the best results across all evaluation metrics. Compared to the best baseline model, AdaMBind reduced MSE by 0.67% (from 0.8636 to 0.8578), improved CI by 0.86% (from 0.7776 to 0.7843), increased $R^2$ by 15.03% (from 0.4186 to 0.4815), enhanced the Spearman and pearson correlation coefficient by 9.79% (from 0.5991 to 0.6578) and 6.67% (from 0.6530 to 0.6966). On the Davis dataset, AdaMBind demonstrated the best performance in CI and Spearman correlation metrics, achieving improvements of 0.34% (from 0.8194 to 0.8222) and 2.27% (from 0.6118 to 0.6257) over the best baseline model, respectively. Its MSE was 0.5060, ranking fourth, while its $R^2$ value of 0.4693 was second only to HiSIF (0.5096). The Pearson correlation coefficient exhibited a slight decrease of 0.98% (from 0.6946 to 0.6878).

The advantages of AdaMBind were also evident in the few-shot setting, as illustrated in Fig. 2b, c. On the KIBA dataset, compared to the best baseline, AdaMBind reduced MSE by 14.45% (from 0.6383 to 0.5461), improved CI by 8.90% (from 0.6327 to 0.6891), increased $R^2$ from 0.1210 to 0.2540, and enhanced Spearman and Pearson correlation coefficients by 41.19% (from 0.3577 to 0.5051) and 26.44% (from 0.4127 to 0.5219), respectively. On the BindingDB dataset, compared to the best baseline model, AdaMBind reduced MSE by 10.08% (from 1.3474 to 1.2116), while achieving superior $R^2$ (from 0.2693 to 0.3094). The Spearman and Pearson improved by 6.22% (from 0.5011 to 0.5323) and 6.66% (from 0.5282 to 0.5634). On the Davis dataset, compared to the best baseline model, AdaMBind reduced MSE by 11.53% (from 0.5363 to 0.4744), improved $R^2$ from 0.2495 to 0.2917, and enhanced Pearson correlation coefficient by 4.15% (from 0.5279 to 0.5499), while showing slight decrease in CI performance by −2.43% (from 0.7693 to 0.7507) and slight decrease in Spearman correlation coefficient by −2.54% (from 0.5032 to 0.4904).

In summary, under the random task split, AdaMBind demonstrates exceptional and stable predictive performance in both the majority and few-shot settings, significantly outperforming existing baseline methods

## Performance comparison of Baselines and AdaMBind under novel task split based on sequence similarity

The novel task split is designed to simulate a more challenging real-world scenario, where models are required to predict affinity for entirely new targets with low sequence similarity to those in the training set (see Methods for details of the novel task split). Under this configuration, we compared the generalization performance of AdaMBind against all baseline models.

The result under the novel task split is shown in Fig. 3 (Supplementary Table 16–21). As illustrated in Fig. 3a, c, under the majority setting on the KIBA dataset, compared to the best baseline model, AdaMBind reduced MSE by 22.52% (from 0.4680 to 0.3626), improved CI by 4.97% (from 0.7140 to 0.7495), increased $R^2$ from 0.2821 to 0.3738, and enhanced Spearman and Pearson correlation coefficients by 20.20% (from 0.5224 to 0.6279) and 16.57% (from 0.5352 to 0.6239), respectively. On the BindingDB dataset, compared to the best baseline model, AdaMBind reduced MSE by 1.62% (from 0.8902 to 0.8758), improved $R^2$ by from 0.3837 to 0.4114, and increased the Pearson correlation coefficient by 1.49% (from 0.6313 to 0.6407), while showing slight decreases in CI and Spearman correlation by 0.27% (from 0.7710 to 0.7689) and 0.35% (from 0.6086 to 0.6064), respectively. On the Davis dataset, compared to the best baseline model, AdaMBind improved $R^2$ by 7.00% (from 0.4168 to 0.4460) and Pearson correlation from 0.6517 to 0.6702, while MSE, CI, and Spearman correlation decreased by 2.42% (from 0.4960 to 0.5080), 1.48% (from 0.8187 to 0.8066), and 9.50% (from 0.6606 to 0.5978), respectively.

As shown in Fig. 3b, c, the advantages of AdaMBind were even more pronounced under the few-shot setting. On the KIBA dataset,

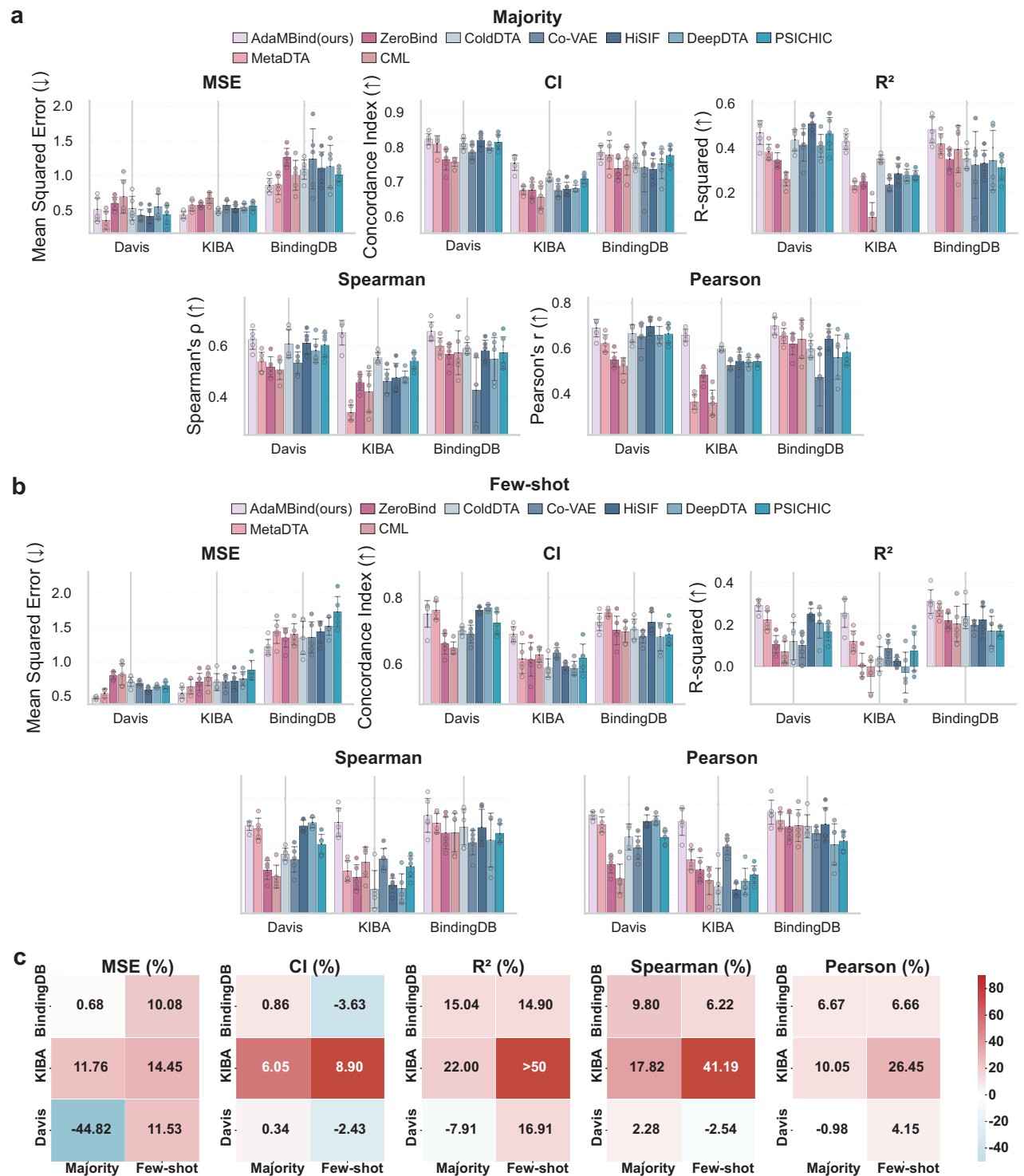

**Fig. 2 | Performance comparison between AdaMBind and baseline models under random task split. a**, **b** Performance evaluation of AdaMBind and baseline models under the majority setting (**a**) and few-shot setting (**b**). In the majority setting, the support set size is 40, while in the few-shot setting, it is 5. The evaluation metrics include MSE, CI, R², Spearman and Pearson. Five independent replications of each method were performed (*n* = 5). Data are expressed as means ± std. **c** Heatmap showing performance gains or losses of AdaMBind compared to its strongest competitor under both majority and few-shot settings. The three subplots correspond to 5 metrics (MSE, CI, R², Spearman, Pearson). The y-axis lists different datasets, and the x-axis represents either the majority or the few-shot setting. Color represents performance difference, where red indicates performance improvement and blue indicates decline. Source data are provided as a Source Data file.

compared to the best baseline model, AdaMBind reduced MSE by 24.83% (from 0.6532 to 0.4910), improves CI by 5.54% (from 0.6451 to 0.6808), increased R² from 0.1055 to 0.2983, and enhanced Spearman and Pearson correlation coefficients by 22.07% (from 0.3864 to 0.4717) and 27.70% (from 0.4341 to 0.5543), respectively. On the BindingDB dataset, compared to the best baseline model, AdaMBind improved R² from 0.2184 to 0.2536. The Spearman and Pearson correlations improved by 3.84% (from 0.4529 to 0.4703)

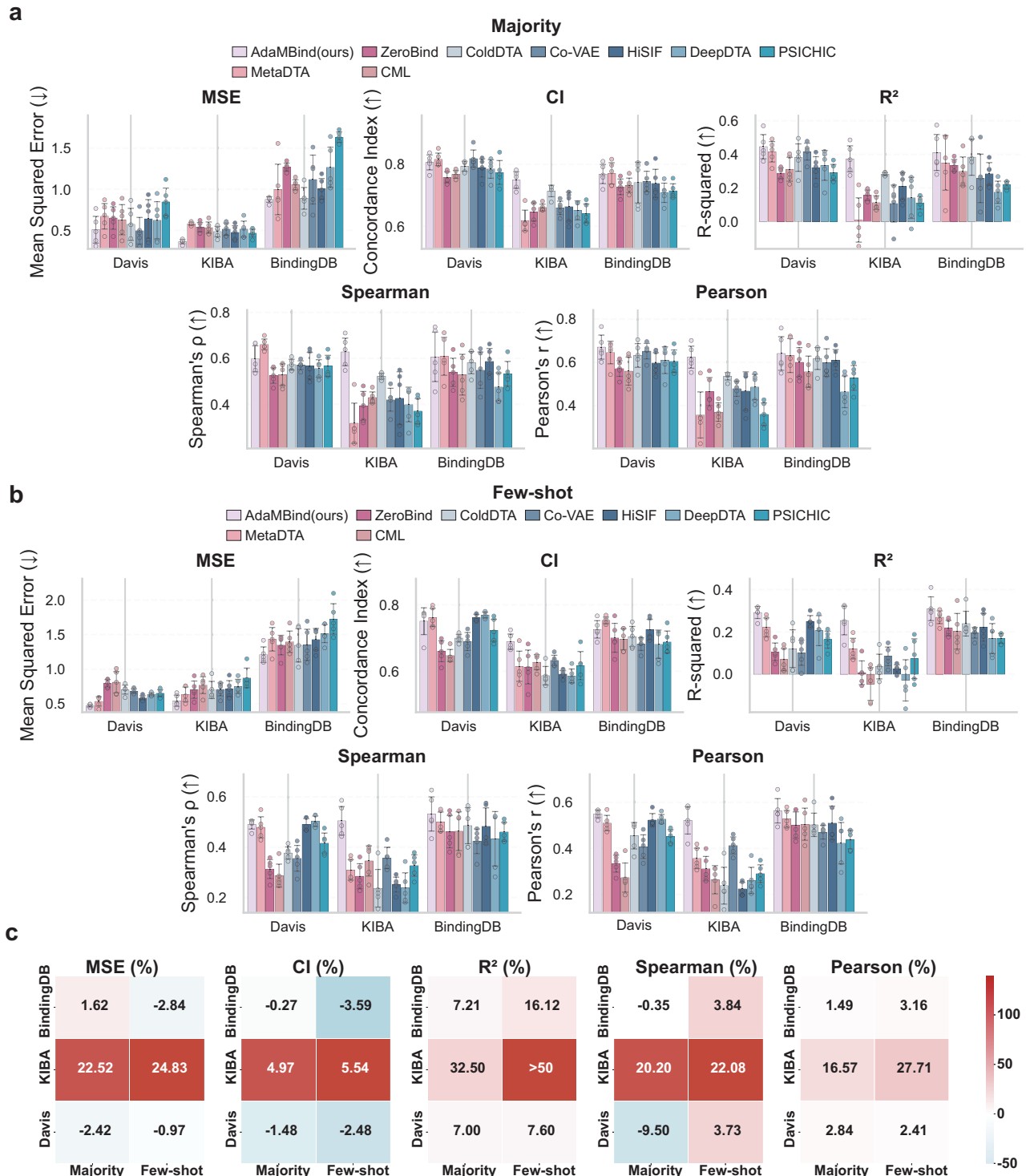

**Fig. 3 | Performance comparison between AdaMBind and baseline models under novel task split based on sequence similarity. a, b** Performance evaluation of AdaMBind and baseline models under the majority setting (**a**) and few-shot setting (**b**). In the majority setting, the support set size is 40, while in the few-shot setting, it is 5. The evaluation metrics include MSE, CI, $R^2$, Spearman and Pearson. Five independent replications of each method were performed ($n = 5$). Data are expressed as means ± std. **c** Heatmap showing performance gains or losses of AdaMBind compared to its strongest competitor under both majority and few-shot settings. The three subplots correspond to 5 metrics (MSE, CI, $R^2$, Spearman, Pearson). The y-axis lists different datasets, and the x-axis represents either the majority or the few-shot setting. Color represents performance difference, where red indicates performance improvement and blue indicates decline. Source data are provided as a Source Data file.

and 3.16% (from 05012 to 0.5171), respectively, while MSE and CI decreased by 2.84% (from 1.2610 to 1.2968) and 3.59% (from 0.7240 to 0.6980). On the Davis dataset, compared to the best baseline model, AdaMBind improved $R^2$ from 0.2347 to 0.2526. The Spearman and Pearson correlations improved by 3.73% (from 0.4749 to 0.4926) and 2.41% (from 0.5224 to 0.5350), respectively, while MSE and CI decreased by 0.97% (from 0.6069 to 0.6128) and 2.48% (from 0.7784 to 0.7591).

In summary, AdaMBind also consistently outperforms baseline models in both majority and few-shot settings under the more challenging novel task split, demonstrating its robust generalization capability for predicting affinity to entirely unseen protein targets.

## Cross-domain generalization evaluation

Furthermore, considering that the distribution characteristics of different datasets may affect model training and generalization capability, we conducted a systematic analysis of the label distributions across three benchmark datasets (BindingDB, KIBA, and Davis). The results (Supplementary Fig. 1) revealed that the affinity distributions of the KIBA (with data densely clustered within the range of [11.20, 11.92]) and Davis (with data densely clustered within the range of [5.00, 5.52]) datasets are highly concentrated, while BindingDB exhibits a moderate level of concentration. Such distribution concentration may cause models to overfit to the main distribution intervals, thereby impairing their predictive performance on out-of-distribution samples. To more comprehensively validate the model's generalization performance across different data distributions, we further designed a cross-domain generalization evaluation experiment under the novel task split. Specifically, the KIBA dataset was used as the source domain, while the BindingDB dataset was used as the target domain for cross-domain transfer validation. The experimental results (Supplementary Table 3) showed that on the KIBA→BindingDB cross-domain transfer path, AdaMBind significantly outperformed all baseline models on most evaluation metrics. Compared to other competitive baselines PSICHIC, AdaMBind demonstrated consistent improvements across multiple metrics. AdaMBind improved CI by 1.34% (from 0.7359 to 0.7458), increased $R^2$ by 12.15% (from 0.3066 to 0.3446), and enhanced Spearman and Pearson correlation coefficients by 2.87% (from 0.5182 to 0.5326) and 4.70% (from 0.5695 to 0.5961), respectively.

In summary, AdaMBind not only consistently outperforms existing baseline methods under both random task split and novel task split, but also fully demonstrates the advantages of the adaptive task meta-learning framework in enhancing out-of-distribution generalization capabilities and cross-domain transfer adaptability.

## Support set size sensitivity analysis

In real-world drug discovery, the number of labeled samples can vary greatly across tasks. To evaluate the performance of AdaMBind under such diverse conditions, we specifically designed a series of experiments to examine the impact of different sizes of support sets on the model performance. Specifically, we set the support set size for each task to 5, 10, 20, 30, and 40, and tested the model on three widely used datasets: BindingDB, KIBA, and Davis. These experiments were conducted under both the random task split and the novel task split.

As shown in Fig. 4a, under the random task split, AdaMBind achieved the best performance across 2 datasets (BindingDB, KIBA) for support set sizes ranging from 5 to 40. On the Davis dataset, it ranked second only to HiSIF when the support size was set to 40. Notably, AdaMBind demonstrated a clear performance advantage even at very small support set sizes, and its performance improved steadily as the number of samples increased. Other meta-learning-based models, such as MetaDTA and CML, showed a similar trend across different support set sizes. However, their overall performance varied considerably across datasets: while they performed well on BindingDB, their improvements on Davis and KIBA were more limited. In contrast, non-meta-learning methods, including HiSIF, ColdDTA, and Co-VAE, performed relatively poorly when the support set size was small (5 or 10 samples). Their performance improved substantially only when the support set was increased to 20 or 30 samples. Under the novel task split based on sequence similarity (Fig. 4b), AdaMBind and the baselines exhibited a trend consistent with that of the random task split. These results suggested that AdaMBind is generally more stable across varying data scales, whereas conventional baseline models are highly sensitive to data volume, relying heavily on large sample sizes to learn effective patterns.

Together, these findings further validated that our model can rapidly and efficiently adapt using only limited sample information, enabling effective knowledge transfer even under data-scarce conditions.

## Ablation study

To validate the contribution of each module in AdaMBind to DTA prediction, we designed and implemented a series of ablation experiments. Specifically, we removed the meta-learning module, adaptive task module, and the label noise strategy used during training, respectively, while keeping the other components unchanged, to assess the importance of each module. All experiments were independently repeated under the majority setting using 5 different random number seeds to ensure robustness. Details of the experimental setup are provided in the Methods.

As shown in Fig. 5, on the BindingDB dataset, the full AdaMBind model achieved the best performance across all evaluation metrics in both random task split and novel task split, confirming that each component contributed to its DTA prediction capability. Among these modules, removing the meta-learning module resulted in the largest performance degradation. This highlighted the critical role of the meta-learning module in predicting affinities for unseen targets. Specifically, MAML shared and updated parameters across multiple target-specific tasks, enabling the model to quickly adapt to new tasks. As a result, under data-limited conditions, the meta-learning module had a more pronounced impact on performance than other components. In addition, removing the adaptive task module also led to a decrease in prediction accuracy. By dynamically adjusting task sampling probabilities, this module helped the model select training tasks that were more compatible with its current state, enhancing adaptability to few-shot and new target scenarios. Moreover, removing the label noise injection strategy caused moderate performance drops across metrics, indicating its role in improving robustness.

In summary, each module of AdaMBind made a positive contribution to its overall performance. The meta-learning module served as the core of the framework, enabling rapid adaptation to new tasks and playing a key role in addressing cold-target and few-shot learning. The adaptive task module further enhanced this adaptability by dynamically adjusting the meta-training sets, allowing the model to better align with task-specific characteristics. Meanwhile, the label noise injection strategy introduced perturbations during training, improving the model's robustness and generalization.

## Adaptive task module leads to better predictive performance

The results of the ablation study indicated that the meta-learning module is the core component enabling the model to quickly adapt to new targets. The adaptive task module further strengthens this adaptability by dynamically constructing meta-training sets. To better understand how the adaptive task module identified and prioritized high-quality tasks during the meta-learning process, we further investigated the characteristics of the tasks selected by the adaptive task module and how they contributed to improved generalization performance.

The adaptive task module is based on the assumption that not all tasks contribute equally during the meta-learning process. To address this, a learnable adapter is introduced to quantify task importance using two key factors: the loss on the query set and the gradient similarity between the support and query sets. As shown in Fig. 6, we visualized the correlation between task sampling weights and these two factors across the three datasets. Tasks with higher query losses— often associated with greater noise or greater difficulty—tended to receive lower sampling weights. In contrast, tasks with higher gradient similarity between support and query sets received higher weights.

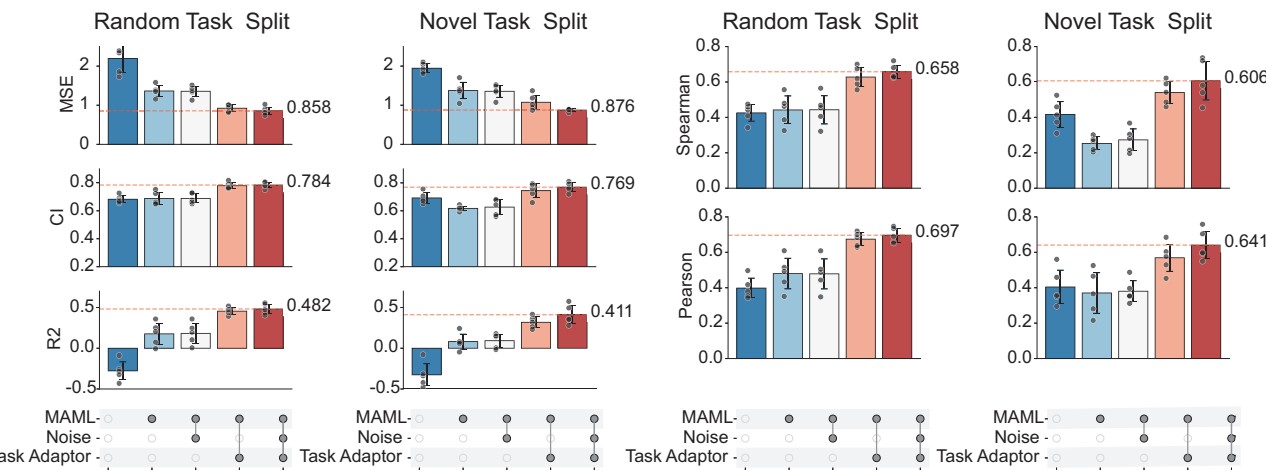

**Fig. 4 | Support set size sensitivity analysis.** Performance comparison under random task split (**a**) and novel task split (**b**). Comparing AdaMBind and baseline models across different support set sizes (5, 10, 20, 30, 40). The evaluation metric is $R^2$. Five independent replications of each method were performed ($n = 5$). Data are expressed as means ± std. Source data are provided as a Source Data file.

**Fig. 5 | Ablation Study.** Ablation study showing the impact of removing key components: the meta-learning module, the adaptive task module, and the label noise injection strategy during training. Model performance is evaluated using MSE, CI, $R^2$, Spearman and Pearson. Five independent replications of each method were performed ($n = 5$). Data are expressed as means ± std. Source data are provided as a Source Data file.

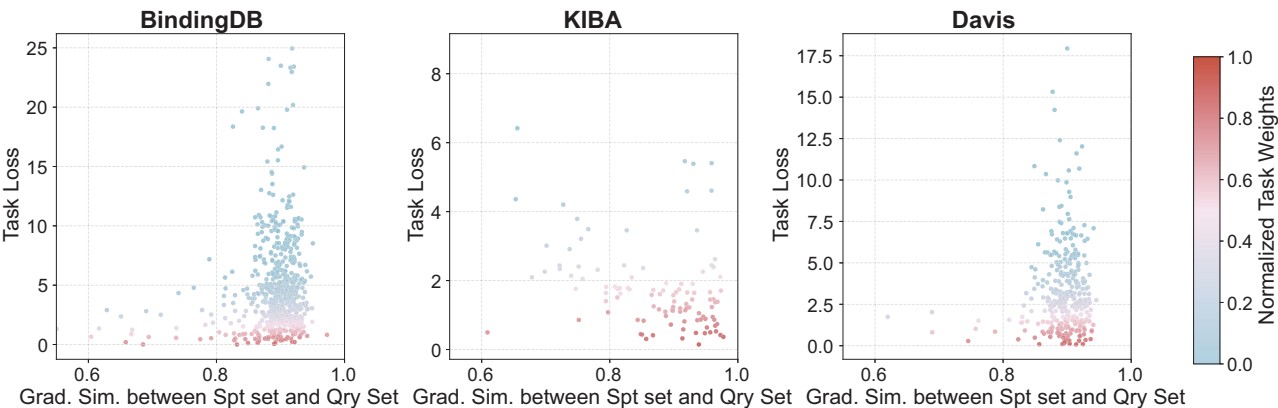

**Fig. 6 | Visualize the weights from the adaptive task module across three benchmark datasets (BindingDB, KIBA, and Davis).** The horizontal axis represents the gradient similarity between the support and query sets of each task, and the vertical axis represents the query loss. Each scatter point corresponds to a specific task, and the color of the point indicates its assigned sampling weight: warmer colors (red/orange) denote higher weights, while cooler colors (blue/cyan) denote lower weights. Source data are provided as a Source Data file.

This suggested that when the optimization directions of the support and query samples align more closely, the model learned more effectively from that task, resulting in better generalization and a higher assigned weight.

These findings were consistent with the theoretical framework proposed by Guy Hacohen et al.[25], which suggests that learning efficiency and generalization can be improved by prioritizing tasks that are more consistent with the target hypothesis, following an "easy-to-hard" learning principle. This also explained why the adaptive task module could guide the meta-learner to focus on more representative and transferable tasks, thereby enhancing the overall effectiveness of meta-learning.

## AdaMBind generalizes robustly beyond training task similarity

A common assumption when evaluating the generalization capability of meta-learning models to new tasks is that the model has learned transferable knowledge from a diverse yet correlated set of training tasks[30]. However, model performance can heavily depend on the similarity in feature distribution between test tasks and training tasks. This dependency may limit the model's applicability in real-world drug discovery scenarios, as novel targets (serving as test tasks) often exhibit significant differences in sequence or structure compared to known targets (serving as training tasks). To systematically evaluate the robustness of AdaMBind against such task distribution shifts, we designed an experiment to quantitatively analyze the relationship between model performance and task similarity. We selected CML, ZeroBind, and MetaDTA as the strong meta-learning baseline for comparison.

Each task was defined as a DTA prediction problem centered around a specific protein target. Consequently, task similarity was jointly determined by its core elements: the biological characteristics of the protein and the chemical characteristics of the associated drugs. To construct a task-level representation, we employed the following methodology: For proteins, we used the pre-trained protein language model ESM-2 to extract deep semantic embedding vectors from their amino acid sequences. For drugs, we used UniMol to extract deep semantic embedding vectors for each molecule. To obtain a task-level representation of the chemical space, we performed element-wise averaging of the UniMol embedding vectors for all drugs within a task, yielding a comprehensive "task-level drug representation". Finally, we concatenated the protein's ESM embedding vector with the averaged drug's UniMol embedding vector to form the final representation vector for that task. Based on this representation, we computed the cosine similarity between the representation vectors of any two tasks.

The cosine similarity, ranging from $[-1, 1]$, measures the directional alignment in the feature space, with values closer to 1 indicating higher similarity between tasks.

Based on the task representation, for each test task, we calculated its similarity to all training tasks and define the set of training tasks with similarity exceeding a threshold (0.6) as $\{S_{sim}\}$, with its size denoted as $N_{sim}$. $N_{sim}$ intuitively reflects the number of "neighbors" of the test task within the training distribution. To quantify the model's dependence on similarity, we defined the generalization $Gap$ as the difference between the actual performance ($R^2$) on the test task and the average performance ($R^2$) on its similar training set $\{S_{sim}\}$. A smaller positive or negative value of $Gap$ indicates that the model's performance on the test task remains close to the average performance of its similar training tasks, regardless of whether the test task has many or few similar "neighbors" in the training set, thus implying a low dependence on task similarity. Conversely, if the $Gap$ increases as the number of $N_{sim}$ grows, it indicates that the model's performance on the test task is highly dependent on the number of similar "neighbors." Specifically, the model performs well when there are many similar tasks in the training set, but its performance significantly declines when there are fewer similar tasks.

To visually reveal the extent to which different models rely on task similarity, we visualized the relationship between the generalization gap $Gap$ (i.e., the difference between the $R^2$ of a test task and the average $R^2$ of its similar training tasks) and the size of the similar training set $N_{sim}$. The results (Fig. 7 and Supplementary Fig. 3) indicated that AdaMBind exhibited remarkable generalization robustness, with its performance being minimally influenced by task similarity. This distinction is particularly pronounced on the challenging KIBA dataset (Fig. 7). The generalization gap $Gap$ for AdaMBind did not show a systematic shift as $N_{sim}$ increases. Linear regression analysis further confirmed this observation: AdaMBind's $Gap$ showed only a weak and statistically non-significant positive correlation with $N_{sim}$ (Pearson's $r = 0.09$, $p$-value $= 0.79$). This implied that regardless of whether a test task had many similar "neighbors" (high $N_{sim}$) or very few (low $N_{sim}$) within the training distribution, AdaMBind's actual performance ($R^2$) on that task remained stable and comparable to the level achieved by its similar training tasks.

In contrast, other meta-learning baselines demonstrated varying degrees of significant dependence on task similarity. Specifically, the $Gap$ for CML, ZeroBind, and MetaDTA all showed a clear positive correlation trend with $N_{sim}$, with correlation coefficients that were statistically significant (CML: Pearson's $r = 0.61$, $p$-value $= 0.05$; MetaDTA: Pearson's $r = 0.49$, $p$-value $= 0.13$) or indicated a strong

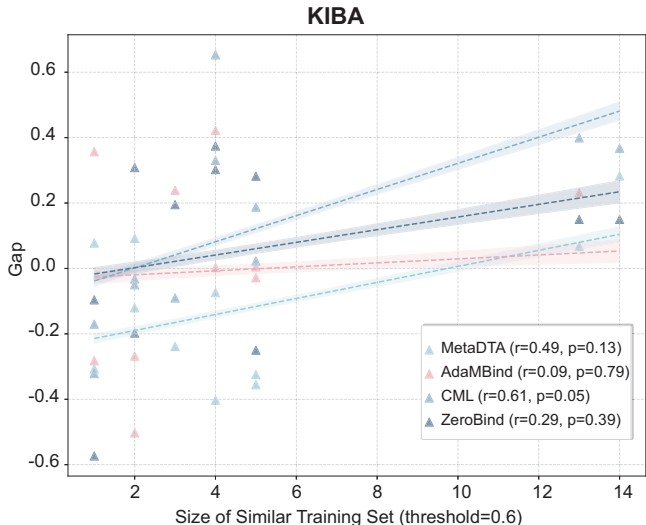

**Fig. 7 | AdaMBind's Generalization Performance in Relation to Task Similarity.** This figure illustrates the relationship between the generalization gap (*Gap*) and the size of the similar training set ($N_{sim}$). Each point represents a test task. The x-axis indicates the number of training tasks whose similarity to the current test task exceeds a threshold of 0.6. The y-axis represents the generalization gap. For each model, the dashed line in the figure represents the least-squares linear fit to the data points (two-sided Student *t* test), and the shaded band denotes the 95% confidence interval of the regression. The slope of the fitted line quantifies how the generalization gap changes with the number of similar training tasks $N_{sim}$: a positive slope indicates that the gap *Gap* widens (performance becomes more dependent on similarity) as more similar tasks become available, whereas a negative slope implies the opposite trend. A smaller absolute value of the generalization gap indicates that the model's performance is minimally influenced by task similarity. Source data are provided as a Source Data file.

correlational trend (ZeroBind: Pearson's *r* = 0.29, *p*-value = 0.39). This pattern revealed a critical limitation: the strong performance of these models heavily relied on the test task having access to a sufficient number of highly similar training tasks as "support". When confronted with a novel target that had few "neighbors" in the training set (low $N_{sim}$ scenario), their generalization performance degraded significantly (*Gap* becomes a large negative value), indicating that the meta-knowledge they had acquired has limited cross-task transferability and struggles to effectively address genuine "cold-start" scenarios.

In summary, by not relying on highly similar training priors, AdaMBind could provide more reliable and stable affinity predictions for these highly challenging novel targets, thereby offering a more powerful computational tool for addressing the cold-start problem in drug discovery.

**Validation of drug discovery against ESR and TP53 targets**

In real-world drug discovery, the data available for model training is often severely limited. This limitation manifests in two key aspects: restricted diversity of protein targets and the scarcity of binding affinity measurements. Insufficient target diversity results in a limited number of meta-tasks during training, which can impair the model's ability to generalize to unseen targets. Meanwhile, inadequate binding data per task increases the difficulty of learning accurate drug-target interaction patterns from small support sets. To evaluate the potential of AdaMBind under such real-world constraints, we designed a virtual screening task to assess its ability to predict high-affinity drug-target interactions under the dual challenges of a "limited number of meta-tasks" and "limited size of the support set".

This experiment was based on the LIT-PCBA dataset constructed by Li et al.[31] A notable characteristic of this dataset is the limited target

diversity, which introduces an additional challenge of "limited number of meta-tasks". This setting is closer to the real drug development scenario where target diversity is scarce, and the number of available training tasks is limited.

As shown in Fig. 8a, following the previous experimental setup, we partitioned the data into 15 tasks based on different target proteins (Supplementary Table 2). Among them, ESR and TP53 were randomly selected as the test tasks, while the remaining 13 tasks were used for training. During the construction of meta-training tasks, we discarded inactive compounds and retained only active ones to better simulate the real-world requirement of identifying high-affinity compounds in drug screening. From each task, 15 active compounds were randomly sampled to form the support sets, the remaining active compounds constituted the query sets. In the meta test tasks, we similarly constructed the support sets with 15 active samples, while the remaining active and inactive compounds collectively formed the query sets for prediction. Specifically, the ESR-targeting task included a total of 3908 compounds, of which only 2.30% were active. Similarly, the TP53-targeting task contained 3409 compounds, with 1.91% classified as active. Under the above setup, we retrained the AdaMBind model based and baselines on the training task and applied it to two testing tasks, aiming to accurately identify high-affinity compounds with potential pharmacological effects from a large number of candidate compounds.

To verify the performance of the model, we first employed the Enrichment Factor (EF)[32] at 1% and 5% as evaluation metrics. Higher EF values indicate that the model can more effectively rank high-affinity active compounds at the top of the prediction list. As shown in Fig. 8b, in the ESR prediction task, AdaMBind achieved EF values of 4.208 (EF@1%) and 4.396 (EF@5%) among the top-ranked predictions. For the TP53 task, AdaMBind also demonstrated superior performance, with EF@1% of 4.197 and EF@5% of 3.687, significantly outperforming other baseline models. We also evaluated the model using the precision@k metric[33]. The results showed that, in the ESR task, when selecting potential active compounds from the top 10 and top 20 predicted molecules, AdaMBind achieved precision rates of 10% and 15%, respectively. For the TP53 task (Fig. 8c), both precision@10 and precision@20 reached 10%. However, most baseline models failed to identify any effective compounds when predicting the top 10 or top 20 candidates. Furthermore, we adopted the BEDROC (α = 80.5) metric[34] for a more comprehensive evaluation. The BEDROC metric, by introducing an exponential weighting function, assigns higher weight to early recognition ability at the front of the ranking list, making it particularly suitable for assessing the effectiveness of identifying highly ranked active compounds in virtual screening. With the parameter α set to 80.5, AdaMBind achieved BEDROC scores of 11.4917% for the ESR task and 9.3461% for the TP53 task. These results were the best among all compared models, further confirming that AdaMBind can effectively prioritize active compounds at the very beginning of the prediction list under data-limited conditions, demonstrating outstanding early recognition capability.

Notably, in recent years, multiple studies have adopted LIT-PCBA as a challenging benchmark for evaluating virtual screening methods. For instance, approaches relying on large-scale pretraining or protein three-dimensional structural information (e.g., S-MolSearch[35], SaBAN[36]) have reported relatively high early enrichment values on this dataset, with EF@1% reaching 6.33-7.36 and BEDROC reaching 6.23%-13.35% (Supplementary Table 5). In contrast, under the strict setting of using only 15 active samples per target as the support set and without relying on external large-scale pretraining data or structural information, AdaMBind still achieved competitive performance in terms of BEDROC and EF@1% across multiple targets (Supplementary Table 4). Furthermore, compared to ensemble docking-based machine learning methods such as MILCDock[37], AdaMBind's EF@1% and BEDROC performance on ESR and TP53 exceeded the average EF@1% and BEDROC

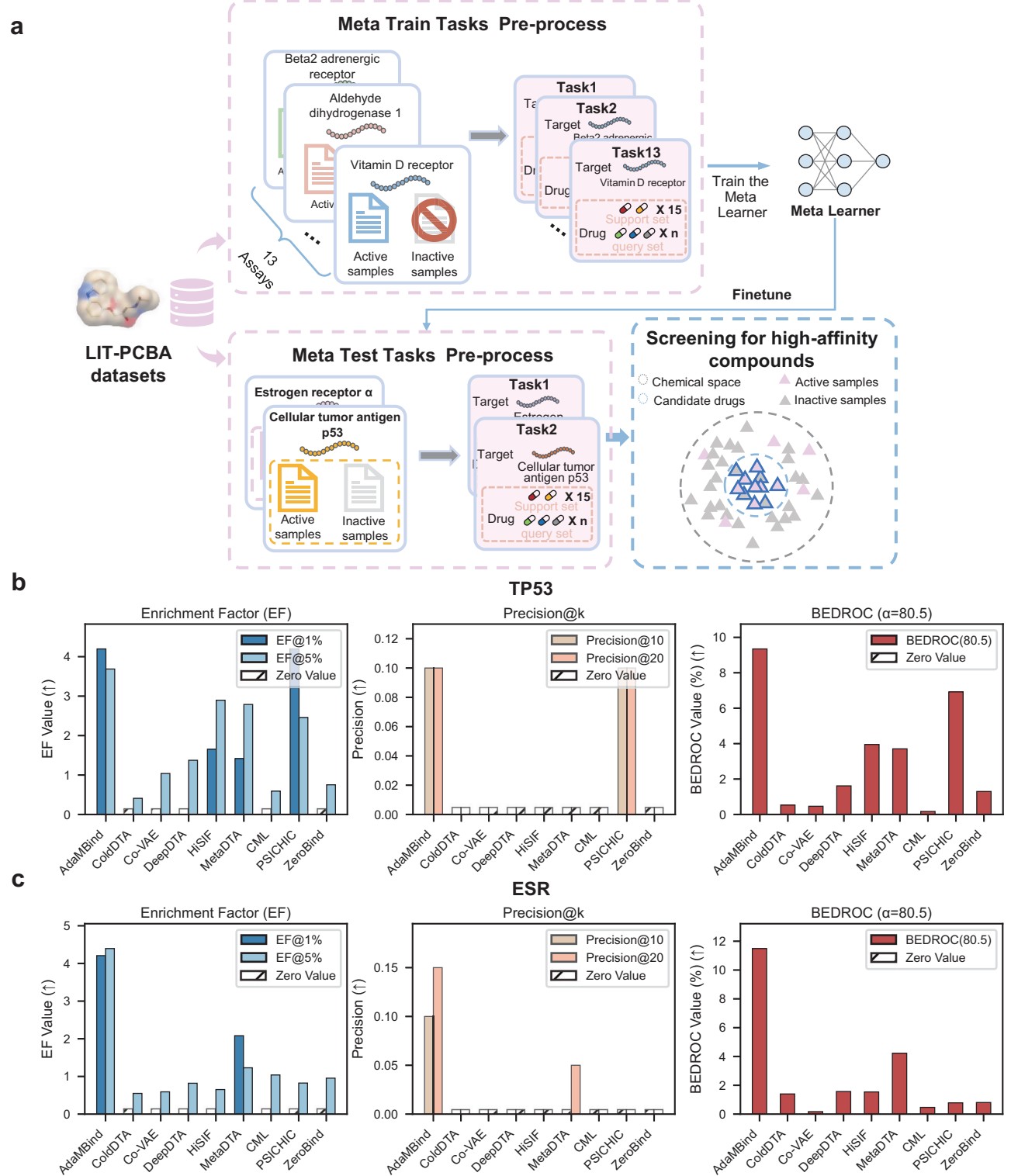

**Fig. 8 | Validation of drug discovery against ESR and TP53 targets. a** The LIT-PCBA dataset was divided into 15 meta-tasks based on target proteins, with 13 tasks used for training and 2 tasks (ESR and TP53) remained for testing. In the training tasks, inactive compounds were discarded, and only active compounds were used, and each support set consists of 15 active samples. For the test tasks, the support set also contains 15 active compounds, while the remaining compounds are used as the query set. The goal of the experiment is to train a model that can efficiently identify high-affinity candidate drugs from a large compound pool under conditions of limited target diversity and few-shot learning. **b, c** Performance comparison between AdaMBind and baseline methods on the ESR (**b**) and TP53 (**c**) targets for identifying active compounds. The left subplots show the enrichment factor at 1% (EF@1%) and at 5% (EF@5%), the mid subplots present precision at the top 10 predictions (precision@10) and at 20 (precision@20), and the right subplots present the BEDROC($\alpha$ = 80.5). Source data are provided as a Source Data file.

**Table 1 | Top 20 drugs ranked by predicted inhibitory potency against FLT3**

| Rank | Drug | CID | Literature Validation | Docking Simulation | Predicted Affinity (pIC$_{50}$) |
|---|---|---|---|---|---|
| 1 | Indolocarbazole Nitrogen Derivative | 44299148 | – | – | 13.2539 |
| 2 | Staurosporine | 44259 | – | √ | 12.7778 |
| 3 | Midostaurin | 9829523 | Stone R M et al.[42] | – | 12.5083 |
| 4 | K-252a | 3035817 | – | – | 12.2893 |
| 5 | Vinblastine | 13342 | – | √ | 11.8991 |
| 6 | Ingenol Mebutate | 6918670 | – | – | 11.5905 |
| 7 | Picrotoxin | 31304 | – | √ | 11.5486 |
| 8 | SCHEMBL1649555 | 133005 | – | – | 11.4881 |
| 9 | CHEMBL282318 | 104826 | – | – | 11.4813 |
| 10 | Tiotropium | 5487427 | – | – | 11.3839 |
| 11 | Vinorelbine | 44424639 | Ramaswamy K et al.[43] | – | 11.3628 |
| 12 | Omadacycline | 54697325 | – | √ | 11.3247 |
| 13 | Fluticasone Furoate | 9854489 | – | – | 11.2953 |
| 14 | SCHEMBL21067566 | 444853 | – | – | 11.2512 |
| 15 | Paclitaxel | 36314 | – | – | 11.2391 |
| 16 | Vincristine | 5978 | – | √ | 11.1512 |
| 17 | 4-hydroxybenzoyl-CoA | 168718 | – | – | 11.1289 |
| 18 | Recinnamine | 32681 | – | √ | 11.1275 |
| 19 | Atracurium Besylate | 47320 | – | – | 11.1261 |
| 20 | Navitoclax | 24978538 | Kivioja J L et al.[44] | – | 11.1082 |

reported by MILCDock across multiple test targets (Supplementary Table 6, 7), while demonstrating more stable cross-target adaptability without exhibiting sharp performance degradation on certain targets (e.g., EF@1% = 0 on the ADRB2 target).

In summary, under the dual challenges of a limited number of meta-tasks and a small support set size, AdaMBind still demonstrated excellent capability in identifying active compounds. These results not only validated the effectiveness and robustness of the model but also highlighted its promising potential for real-world drug discovery applications.

## Application of the discovery of potent inhibitors of FLT3

Acute myeloid leukemia (AML) is a highly heterogeneous hematological malignancy, in which the patient population carrying mutations in the FLT3 gene displays a high relapse rate and poor prognosis, making FLT3 a key target in the therapeutic strategy of AML[38]. However, with the popularity of FLT3 inhibitors in clinical applications, the problem of drug resistance has become increasingly apparent, posing a serious challenge to existing treatment options. Therefore, there is an urgent need to develop new therapeutic approaches to overcome this obstacle.

In this context, this study attempted to identify potential drugs with high affinity for FLT3 from the DrugBank database via AdaMBind[39]. To verify the validity of these predictions, we used molecular docking simulations combined with literature validation to screen out the most promising drug candidates. For the data processing stage, we drew on the dataset collated in the article by Hou et al, from which we screened for known effective FLT3 inhibitors[40]. Subsequently, detailed affinity information of these inhibitors was obtained from the PubChem database[41], and 10 samples were randomly selected to construct the support set, while all compounds contained in the DrugBank database were used as query sets for further analysis.

In this experiment, we utilized AdaMBind trained on the BindingDB dataset to predict the top 20 compounds with the highest binding affinity on the query set, as shown in Table 1, Among these top 20 ligands with the highest binding affinity, Midostaurin[42] has been validated in the literature to indeed show potent inhibition of FLT3;

whereas Vinorelbine[43], and Navitoclax[44] are commonly used in combination regimens in AML therapy.

To further explore potential FLT3 inhibitor candidates, we obtained the 3D structure of FLT3 from the Protein Data Bank (PDB ID: 6JQR) and performed 30 rounds of molecular docking simulations for each of the remaining 16 compounds using the AutoDock tool[45]. The results are shown in Fig. 9b, six compounds, Staurosporine, Vinblastine, Picrotoxin, Vincristine, Omadacycline and Recinnamine, showed strong interaction with FLT3.

Considering the predicted binding affinity, stability of the docking results, and reliability of the binding mode, we then selected Staurosporine for experimental verification. ADP-Glo™ Kinase Assay[46] was used to validate the interaction between Staurosporine and FLT3. For the FLT3 kinase experiment, we selected Quizartinib as the positive control. In our system, the compound showed an IC$_{50}$ (half maximal inhibitory concentration) of 2.58 nM, comparable to its reported FLT3 inhibitory activity (4.2 nM)[47], confirming our system's validity for FLT3 kinase activity evaluation. As shown in Fig. 9d (left), Staurosporine inhibited FAK[48] activity with an IC50 of 0.278 nM and exhibited a lower IC$_{50}$ against FLT3 than Quizartinib, indicating a stronger inhibitory effect on FLT3. Given that the FLT3-ITD mutation is a key driver and therapeutic target in Acute Myeloid Leukemia[49], we further evaluated Staurosporine against this clinically relevant mutant. Figure 9d (right) showed that Staurosporine exhibited potent activity against the FLT3-ITD kinase (IC$_{50}$ = 0.465 nM), which was over 10-fold more potent than Quizartinib (IC$_{50}$ = 5.7 nM). These findings experimentally validated the accuracy of AdaMBind's computational predictions and highlighted its capability to identify highly potent drug-target relationships with therapeutic potential.

To further elucidate the underlying molecular mechanism of the potent inhibitory activity observed, we analyzed the binding mode of Staurosporine to FLT3 through docking simulations. As shown in Fig. 9c, Staurosporine stably occupies the ATP-binding pocket of the FLT3 kinase. Its molecular scaffold engages in extensive hydrophobic interactions with key residues, including Leu818 and Val624. Moreover, the formation of two hydrogen bonds is observed: one between the carbonyl oxygen of Staurosporine and the backbone nitrogen of

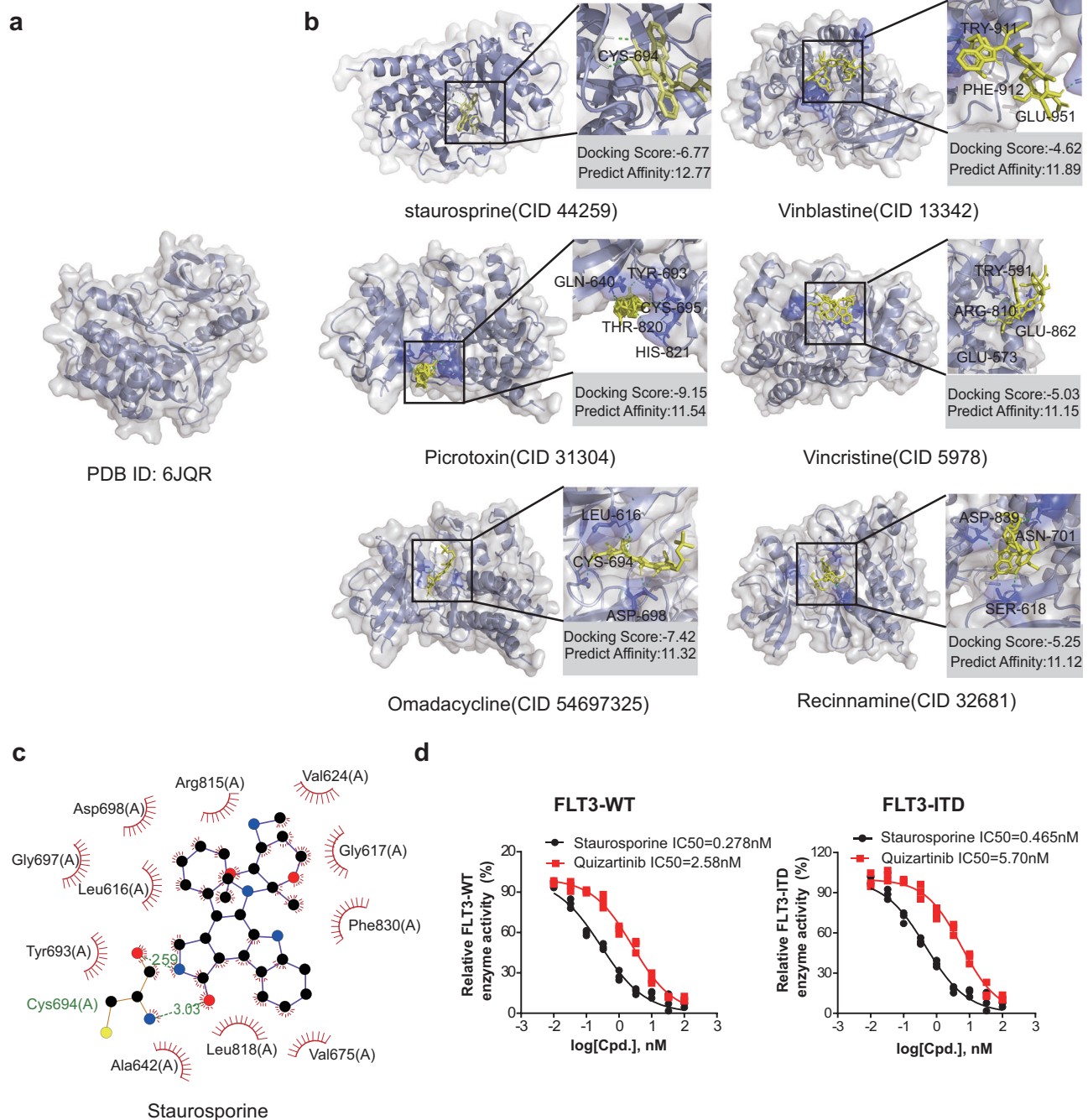

**Fig. 9 | Application of the discovery of potent inhibitors of FLT3. a** Visualization of the FLT3 structure used in AutoDock docking (PDB ID:6JQR). **b** AdaMBind identified several potential inhibitors for the FLT3 target. The 3D binding poses of 6 compounds with the FLT3 protein were generated using AutoDock. Experimental Validation of Predicted Relationships. **c** 2D diagram of the interaction between FLT3 and Staurosporine simulated by AutoDock. **d** Concentration-activity curves of Quizartinib (positive control) and Staurosporine in FLT3 kinase ADP-Glo assays for WT-type (left) and ITD-type (right). Mean ± SEM of three independent experiments is shown ($n = 3$). Source data are provided as a Source Data file.

Cys694 (2.9 Å), and another between its amino group and the side-chain carboxyl of Asp698 (3.1 Å). In addition, CH-π interactions are identified with Tyr693.

Comparative analysis reveals that the binding characteristics of Staurosporine exhibit a complex pattern. In certain respects, its binding mode resembles that of Type I FLT3 inhibitors[50], as it localizes within the ATP-binding pocket without deeply occupying the hydrophobic cavity typically targeted by Type II inhibitors. The hydrogen-bond network formed with the conserved catalytic residues Cys694 and Asp698 aligns with interaction patterns observed in some kinase inhibitors targeting the active conformation. Furthermore, the absence of strong direct interactions with the gatekeeper residue Phe691 distinguishes it from many conventional FLT3 inhibitors, whose efficacy is often compromised by the F691L mutation. This interaction profile, centered on the ATP-binding pocket while displaying unique features, provides a structural rationale for the high affinity predicted by AdaMBind and accounts for the potent inhibitory activity observed in our experimental studies against both wild-type and mutant forms of FLT3.

These findings not only further confirmed the effectiveness of AdaMBind in identifying novel FLT3 inhibitors but also demonstrated its potential in accelerating the drug discovery process.

## Discussion

This study introduces AdaMBind, a meta-learning-based regression model specifically designed for DTA prediction in a few-shot setting. The primary objective of AdaMBind is to address a challenge in early-stage drug discovery: effectively leveraging a minimal amount of available data for a novel target to rapidly develop target-specific understanding and achieve reliable affinity prediction. To enhance its adaptability, we incorporate a dynamic adaptive task module based on an "easy-to-hard" principle, which guides the model to learn from simpler tasks (characterized by lower loss and higher gradient consistency) before gradually transitioning to more complex tasks. Notably, the model does not heavily rely on similarity to training tasks when adapting to new tasks, exhibiting more stable performance and lower sensitivity to training task distribution compared to other meta-learning frameworks.

Experimental results demonstrate that AdaMBind consistently outperforms existing methods across three benchmark datasets (BindingDB, KIBA, and Davis) under varying support set sizes. Its superiority is particularly pronounced in few-shot settings, where it even surpasses baseline models that rely on protein structural information. Ablation studies confirm the essential role of the meta-learning module in enabling rapid adaptation, while the adaptive task module improves generalization by following an "easy-to-hard" learning path, and the label noise injection strategy enhances model robustness. In a rigorous evaluation on the LIT-PCBA dataset, AdaMBind effectively identified high-affinity molecules even under extremely low active compound ratios (2.30% for ESR and 1.91% for TP53), achieving EF@1% values of 4.208 for ESR and 4.197 for TP53, BEDROC($\alpha = 80.5$) values of 11.4917% for ESR and 9.3461% for TP53. In a practical application targeting the FLT3 protein for AML, the model successfully predicted potential inhibitors, including Staurosporine. Experimental validation confirms that Staurosporine exhibits superior inhibitory activity against both FLT3-WT and the clinically relevant FLT3-ITD mutant. The $IC_{50}$ against FLT3-WT and FLT3-ITD is significantly lower than that of the positive control, Quizartinib. These findings collectively underscore the practical utility of AdaMBind in real-world drug discovery scenarios characterized by data scarcity.

Although AdaMBind demonstrates strong performance in cold-start and few-shot learning tasks, several limitations should be noted. First, as this study primarily focuses on the design and validation of the meta-learning framework, we adopted a relatively simple architecture for the base learner, the drug GNN encoder relies only on node features and graph topology without explicitly encoding edge information such as chemical bond types, which may constrain the expressiveness of molecular representations. Second, the current model design does not depend on three-dimensional structural information, which makes it applicable to targets with incomplete structural resolution, but may also limit its predictive capability for highly flexible proteins such as GPCRs and ion channels. The binding affinity of such targets often involves dynamic processes like "induced fit", which may not be fully captured by the static feature fusion mechanism based solely on sequence. Finally, although the model performs well in few-shot scenarios, achieving true zero-shot learning capability remains important for rapidly responding to newly emerging disease targets or rare conditions, representing a key direction for future exploration.

To address these limitations and further advance DTA prediction, future work can proceed along the following directions. At the feature representation level, dynamic structural information derived from molecular dynamics simulations or conformational sampling could be integrated to more accurately characterize the dynamics of drug-target interactions. Simultaneously, enhanced graph neural networks capable of explicitly modeling edge information, such as chemical bonds, could be introduced to improve the fine-grained expressiveness of drug representations. At the algorithmic framework level, more advanced pre-trained protein encoders that incorporate three-

dimensional structures can be integrated with uncertainty quantification mechanisms. This integration would enhance both the reliability and interpretability of predictions. Moreover, to simultaneously achieve numerical prediction accuracy and reliable ranking consistency, we plan to introduce a dual-loss training framework that integrates mean squared error with a ranking-oriented loss function. This method aims to keep the individual predicted value as close as possible to the true value while improving the ranking reliability among overall predictions. At the application paradigm level, zero-shot inference architectures that operate without any support-set fine-tuning could be explored to address zero-shot prediction challenges. Furthermore, thanks to the flexible and model-agnostic design of the AdaMBind framework itself, more powerful base learners for DTA prediction can be conveniently integrated in the future, thereby continuously raising the performance ceiling and generalization capability of the model.

In conclusion, AdaMBind achieves the extraction of transferable knowledge from limited data through the synergistic combination of meta-learning strategies and an adaptive task module. The framework's generality allows it to be integrated with various base learners, offering a powerful and flexible new tool for drug discovery tasks under data-scarce conditions and demonstrating significant potential for addressing the challenge of predicting affinities for unknown targets.

## Methods

### Datasets

In this study, we train our model using three widely recognized benchmark datasets: BindingDB[51], KIBA[52], and Davis[53]. Supplementary Table 1 summarizes these datasets. The Davis dataset includes kinase dissociation constant binding affinities between 68 kinase inhibitors and 442 target proteins. The KIBA dataset contains binding affinity data for 2111 drugs and 229 target proteins. It integrates multiple bioactivity measures, including inhibition constants (Ki), dissociation constants (Kd), and half-maximal inhibitory concentrations ($IC_{50}$), providing a comprehensive view of drug-target interaction strength. The BindingDB dataset comprises binding affinities for 9864 drugs and 1088 protein targets, primarily based on $IC_{50}$ measurements. To improve training stability, we convert the original Kd, Ki and $IC_{50}$ values into their logarithmic forms (pKd, pKi and $pIC_{50}$)[54].

Additionally, in experiments with limited data, we use the LIT-PCBA dataset proposed by Li et al., which includes 15 targets, 7844 active compounds, and 407,381 inactive compounds. These data are carefully selected from PubChem bioassays, and the affinity values for active compounds are all available on PubChem.

### Data split

In this study, we frame the DTA prediction problem within a protein-anchored meta-learning paradigm. We assume a distribution of tasks $p(T)$, where each task $T_i$ corresponds to a unique target protein $t_i$ and comprises all available drugs associated with that target. The task is further divided into a support set $S_i$ and a query set $Q_i$. The support set $S_i = \{(t_i, d_s, y_s)\}_{s=1}^n$ contains pairs of the target $t_i$ and a small number of drugs $d_s$ with affinities $y_s$, while the query set $Q_i = \{(t_i, d_q, y_q)\}_{q=1}^m$ includes the same target $t_i$ with additional drugs $d_q$ with affinities $y_q$. In this way, the full set of tasks $p(T)$ can be partitioned into meta-training set $\{T_i\}_{i=1}^{N_{tr}}$, meta-validation set $\{T_i\}_{i=1}^{N_{val}}$, and meta-testing set $\{T_i\}_{i=1}^{N_{test}}$.

In our implementation, to systematically assess the model's ability to generalize under different levels of task novelty, we adopted two distinct task split strategies:

(i) Random task split: where all tasks are randomly partitioned into meta-training, meta-validation, and meta-testing task sets at a ratio of 8:1:1.

(ii) Novel task split: where tasks are partitioned based on protein sequence similarity using CD-HIT at a 40% identity threshold,

proteins with sequence identity $\geq 40\%$ were grouped into the same cluster, while those with identity $< 40\%$ were placed into different clusters. The resulting clusters were then allocated to meta-training, meta-validation, and meta-testing sets in an 8:1:1 ratio. This ensures that proteins from different sets share low sequence similarity ($< 40\%$) and belong to distinct clusters. Consequently, the meta-testing tasks contain targets that are structurally and sequentially dissimilar to those encountered during meta-training[22,55].

Beyond task novelty, the amount of data available for adaptation within each task also critically affects performance. Therefore, from the perspective of per-task data availability, we further established two learning scenarios with different support set sizes:

(i) Majority setting: where each task's support set contains 40 labeled DTA pairs.
(ii) Few-shot setting: where each task's support set contains 5 labeled DTA pairs.

### The representations of protein and drug

We represent each drug as a molecular graph, where edges indicate atomic bonds and nodes correspond to atoms, each described by a multi-dimensional binary feature vector. These atomic features include five key properties: atom symbol, number of neighboring atoms, number of neighboring hydrogen atoms, implicit valence, and whether the atom is part of an aromatic ring. Using the open-source toolkit RDKit[56], we convert SMILES strings into their corresponding molecular graphs and extract atom-level features. As a result, each compound is mathematically represented as a graph $G_d = \{V_d, E_d\}$, where $V_d$ is the set of node features and $E_d$ is the set of edges.

For protein targets, we encode their amino acid sequences using one-hot vectors, where each amino acid type is mapped to an integer based on its corresponding alphabetical symbol.

### Meta-learning module

In this section, we describe the meta-learning framework used in AdaMBind. We adopt gradient-based methods over metric-based methods[57]. This approach is designed to learn a shared parameter initialization that can be rapidly fine-tuned for novel tasks. Specially, in AdaMBind, we employ the Model-Agnostic Meta-Learning (MAML) framework as the core meta-learning approach.[58] The MAML operates via the synergistic collaboration between a base learner and a meta-learner: the base learner enables fast adaptation to novel tasks, while the meta-learner acquires task-agnostic meta-knowledge across tasks to facilitate such rapid adaptation.

The base learner acts as a task-specific prediction model. We can define a base learner $f_\theta(\cdot)$ to predict affinities for each $T_i$ in meta-training set $\{T_i\}_{i=1}^{N_{tr}}$:

$$y = f_\theta(t, d) \tag{1}$$

The $f_\theta(\cdot)$ adopts a simple architecture: For drug graphs, a two-layer graph neural network[59] is applied to extract structural features. For protein sequences, a three-layer one-dimensional convolutional neural network is applied to capture local patterns. The extracted features are then fused and passed through a fully connected layer to output the predicted binding affinity.

During the k-th meta update in the meta-training phase, each base learner $f_\theta(\cdot)$ is initialized with a shared meta-initialization $\theta_0^{(k)}$ and then updated to $\theta_i^{(k)}$ via $n$ steps of gradient descent on the support set $S_i$ with $\alpha$ as the inner-loop learning rate:

$$\theta_i^{(k)} = \theta_0^{(k)} - \alpha \nabla_\theta Loss(S_i; \theta_0^{(k)}) \tag{2}$$

To evaluate and improve the meta-initialization, we measure the performance of $\theta_i^{(k)}$ on their query sets $Q_i$, and use the aggregated loss to optimize the meta-initialization $\theta_0^{(k+1)}$ as (outer-loop optimization):

$$\theta_0^{(k+1)} = \theta_0^{(k)} - \beta \cdot \nabla_\theta E_{T_{tr} \sim p(T_{tr})}[Loss(Q_i; \theta_i^{(k)})] \tag{3}$$

In practice, due to memory constraints, we can't put the whole meta-training set $\{T_i\}_{i=1}^{N_{tr}}$ at once. As a result, during each meta update, the model samples a batch of $B$ tasks $\{T_i\}_{i=1}^{B}$ from the entire training task distribution by uniform sampling. However, the uniform sampling may cause the model to learn from noisy or less informative tasks.

### Adaptive task module

To address these limitations, we adopt the adaptive task module proposed by Yao et al.[60] to generate sampling probabilities for tasks, which offers a principled way to guide task selection.

During the k-th meta update, we can define an adapter $g_\varphi(\cdot)$ that generates task sampling weights $\omega_i$ with parameter $\varphi$ for each candidate task $T_i \in \{T_i\}_{i=1}^{B}$:

$$\omega_i^{(k)} = g(T_i, \theta_0^{(k)}; \phi^{(k)}) \tag{4}$$

To quantify the importance of task, we consider two key factors: the loss on the query set $Loss(Q_i; \theta_i^{(k)})$, where $\theta_i^{(k)}$ represents the specific-task model parameters for candidate task $T_i$ in k-th meta update; (2) the gradient similarity between the support and query sets of the candidate task $T_i$, i.e., $\langle \nabla_{\theta_0^{(k)}} Loss(Q_i; \theta_0^{(k)}), \nabla_{\theta_0^{(k)}} Loss(S_i; \theta_0^{(k)}) \rangle$. This similarity reflects the generalization gap from the support set to the query set. A large gradient similarity, combined with a high query loss, indicates that the task is relatively difficult. Finally, the Eq.(4) can be reformulated as:

$$\omega_i^{(k)} = g(Loss(Q_i; \theta_i^{(k)}), \langle \nabla_{\theta_0^{(k)}} Loss(Q_i; \theta_0^{(k)}), \nabla_{\theta_0^{(k)}} Loss(S_i; \theta_0^{(k)}) \rangle; \varphi^{(k)}) \tag{5}$$

A higher value of the computed score indicates a greater sampling probability for the task. Once the sampling weights are obtained, we use them to sample $N_{slct}$ tasks from the candidate $B$ tasks to form the selected task set $\{T_i'\}_{i=1}^{N_{slct}}$ for meta-training. In practice, $g_\varphi(\cdot)$ employs two bidirectional LSTM layers[61] as feature extractors to encode the loss information and the gradient similarity, respectively.

### Label noise strategy

To enhance the model's robustness and generalization capability in noisy real-world scenarios, we incorporate random label noise during the meta-training phase. Specifically, we inject uniform distribution noise into the affinity labels of both support and query sets:

$$\tilde{y} = y + \varepsilon, \varepsilon \sim U(-\sigma, \sigma) \tag{6}$$

Where $y$ represents the original affinity value, $\tilde{y}$ denotes the noisy label, and $\sigma$ controls the noise intensity. This noise injection strategy serves as an effective regularization technique that prevents the model from overfitting to potentially noisy training labels and encourages learning more robust feature representations[62].

### Model optimization

In this section, we explore the co-optimization mechanism between the adapter parameters $\varphi$ and the meta-initialization $\theta_0$ during meta-training. Unlike conventional approaches that directly optimize both parameters on meta-training tasks, we introduce a separate validation set $\{T_i\}_{i=1}^{N_{val}}$ containing $N_{val}$ tasks, using its performance as the optimization guidance. Specifically, we employ a bi-level optimization

framework to identify the optimal adapter parameters $\varphi^{(k)}$ that minimize the loss on validation tasks, while the corresponding meta-initialization $\theta_0^*$ are obtained by optimizing the training loss on sampled tasks. The mathematical formulation of this optimization problem is as follows:

$$\varphi^* = \arg\min_{\varphi} E_{T_{val} \sim p(T_{val})}[Loss_{val}(T_{val}; \theta_0^*(\varphi))]$$
$$with\ \theta_0^*(\varphi) = \arg\min_{\theta_0} E_{T_{slct} \sim p(T_{slct})}[Loss_{slct}(T_{slct}; \theta_0)] \tag{7}$$

To overcome the computational burden of exact inner-loop optimization, we adopt a one-step gradient update strategy proposed by Hu et al.[63] to approximate the optimal parameters $\theta_0^*(\varphi)$. At the k-th meta update, the approximate meta-parameters can be formulated as:

$$\theta_0^*(\varphi) \approx \tilde{\theta}_0^{(k+1)}(\varphi^{(k)}) = \theta_0^{(k)} - \beta \cdot E_{T_{slct} \sim p(T_{slct})}[\nabla_\theta Loss(T_{slct}; \theta_0^{(k)}, \varphi^{(k)})] \tag{8}$$

Based on this approximation, we transform the original bi-level optimization problem into the following operational iterative process:

$$\varphi^{(k+1)} = \varphi^{(k)} - \beta \cdot \nabla_\varphi E_{T_{val} \sim p(T_{val})}\left[Loss(T_{val}; \tilde{\theta}_0^{(k+1)}(\varphi^{(k)}))\right] \tag{9}$$

After updating the adapter parameters, we resample $N_{slct}$ tasks to form $\{T_i'\}_{i=1}^{N_{slct}}$ from the candidate task pool $\{T_i\}_{i=1}^B$ using the new $\varphi^{(k+1)}$ and subsequently update the meta-initialization:

$$\theta_0^{(k+1)}(\varphi^{(k)}) = \theta_0^{(k)} - \beta \cdot E_{T'_{slct} \sim p(T'_{slct})}[\nabla_\theta Loss(T_{slct}'; \theta_0^{(k)}, \varphi^{(k+1)})] \tag{10}$$

The detailed step-by-step training procedure is summarized in Supplementary Algorithm 1.

## Evaluation metrics

For evaluation, we employed 5 metrics: mean squared error (MSE), concordance index (CI)[64], and coefficient of determination (R²), Spearman rank correlation coefficient (Spearman), and Pearson correlation coefficient (Pearson).

1.  MSE: It measures the average squared difference between predicted affinity and true affinity:

$$MSE = \frac{1}{N}\sum_{i=1}^{N}(y_i - \hat{y}_i)^2 \tag{11}$$

2.  CI: It evaluates the ranking consistency of predictions, where Z is the number of comparable pairs, and 1(·) is the indicator function:

$$CI = \frac{1}{Z}\sum_{y_i > y_j} 1(\hat{y}_i > \hat{y}_j) \tag{12}$$

3.  R²: It quantifies the proportion of variance in the true values explained by the model:

$$R^2 = 1 - \frac{\sum_{i=1}^{N}(y_i - \hat{y}_i)^2}{\sum_{i=1}^{N}(y_i - \bar{y}_i)^2} \tag{13}$$

4.  Spearman: It assesses the monotonic relationship between predicted affinity and true affinity, where is the difference $d_i$ between the rank of predicted affinity and true affinity:

$$\rho = 1 - \frac{6\sum_{i=1}^{N}d_i^2}{N(N^2 - 1)} \tag{14}$$

5.  Pearson: $r$ measures the linear correlation between predictions and true values:

$$r = \frac{\sum_{i=1}^{N}(y_i - \bar{y})(\hat{y}_i - \bar{\hat{y}})}{\sqrt{\sum_{i=1}^{N}(y_i - \bar{y})^2 \sum_{i=1}^{N}(\hat{y}_i - \bar{\hat{y}})^2}} \tag{15}$$

We also evaluated early recognition capability on the LIT-PCBA dataset using five virtual screening metrics: EF@1% and EF@5% to measure enrichment in the top 1% and 5% of the ranked list, respectively; Precision@10 and Precision@20 to assess the proportion of active compounds among the top 10 and 20 predictions; and BEDROC($\alpha$ = 80.5) to quantify weighted early recognition performance with an emphasis on the initial ranking positions.

6.  EF@k%: It measures the concentration of active compounds within the top k% of the model's ranked predictions relative to random selection. It is defined as

$$EF@k\% = \frac{N_{actives}^{topk\%}}{(k/100) \times N_{actives}} \tag{16}$$

Where $N_{actives}$ is the total number of active compounds in the dataset, and $N_{actives}^{topk\%}$ denotes the number of actives appearing in the top k% of the ranked list.

7.  Precision@n: It measures the proportion of active compounds among the top n ranked predictions. The metric is defined as.

$$Pr\ ecision@n = \frac{N_{actives}^{top\,n}}{n} \tag{17}$$

where $N_{actives}^{topn}$ represents the number of active compounds within the first n positions of the model's ranked output.

8.  BEDROC($\alpha$ = 80.5): It measures the ability of a model to prioritize active compounds at the beginning of a ranked list by applying an exponential weighting scheme that emphasizes top ranks. The metric is defined as:

$$BEDROC(\alpha) = \frac{\sum_{i=1}^{N}e^{-\alpha r_i/N}}{R_a \cdot \frac{1-e^{-\alpha}}{e^{\alpha/N}-1}} \times \frac{R_a \sinh(\alpha/2)}{\cosh(\alpha/2) - \cosh(\alpha/2 - \alpha R_a)}$$
$$+ \frac{1}{1 - e^{\alpha(1-R_a)}} \tag{18}$$

where $R_a$ is the proportion of active compounds, $N$ is the total number of compounds (including inactives), and $r_i$ denotes the rank position of the i-th active compound. The parameter $\alpha$ controls the focus on early ranks.

## BaseLines

To comprehensively evaluate the performance of AdaMBind in drug-target affinity prediction, we compared it against 8 widely used baseline methods: DeepDTA, HiSIF-DTA, ColdDTA, Co-VAE, PSICHIC, MetaDTA, CML and ZeroBind.

**DeepDTA.** A non-meta-learning DTA method that takes drug SMILES and protein amino acid sequences as input. It uses CNNs to extract features and fully connected layers to predict binding affinity.

**HiSIF-DTA.** A non-meta-learning DTA method that inputs 3D protein structures and molecular graphs of drugs. It uses GNNs for encoding and integrates PPI networks to enhance protein representations to predict affinity.

**ColdDTA**. A non-meta-learning DTA method that takes protein sequences and molecular graphs as input. It uses a GNN for drug features, a CNN for protein features, and an attention module to model interactions.

**Co-VAE**. A non-meta-learning DTA method that inputs protein sequences and molecular graphs. It uses GatedCNN for feature extraction, followed by a VAE for latent representation learning and fusion.

**MetaDTA**. A meta-learning-based DTA method that takes drug sequences as input and divides the dataset into target-specific tasks. It uses an attention-based neural process as the base learner and applies MAML for few-shot learning.

**ZeroBind**. A meta-learning-based DTI method that inputs protein structures and drug graphs. It uses GNNs to encode drug graphs and protein graphs, a Subgraph Information Bottleneck (SIB) module to identify binding pockets, and a task-adaptive self-attention mechanism for protein-specific modeling. For the purpose of affinity prediction, we replace its original classification output head with a regression head, enabling the model to predict continuous binding affinity values rather than binary interaction labels.

**CML**. A meta-learning-based DTA method that inputs drug SMILES and target sequences. It uses a contrastive learning block (CLB) to explore correlations among drug-target pairs, and applies MAML for fast adaptation to new drugs and targets.

**PSICHIC**. A non-meta-learning DTA method that inputs protein sequences and drug SMILES strings. It initializes ligand and protein graphs, uses GNNs to encode their representations, and learns interaction fingerprints to predict binding affinity.

In the base learner of AdaMBind, we explored four graph neural network architectures (GCN, GAT, GIN, and a hybrid GAT_GCN model) for drug feature extraction. To determine the optimal encoder for each dataset, we conducted hyperparameter tuning experiments under the majority setting. As shown in Supplementary Table 8, GCN achieved the best performance on the KIBA dataset, while the GAT_GCN combination performed best on BindingDB and Davis. Therefore, AdaMBind used dataset-specific optimal graph encoders in all subsequent experiments.

For the baseline implementations, we closely followed the original hyperparameter settings reported in the respective papers. To ensure a fair comparison, all models were trained and evaluated using the same data splitting strategy described in this study.

## Model implements and experiment settings

All experiments were conducted using the PyTorch framework[65], with key dependencies including RDKit (version 2024.3.3), PyTorch (2.3.1), NumPy (1.24.4), and Pandas (2.2.2). To ensure statistical robustness and reproducibility, all reported results were obtained from five independent experimental runs, each executed with a distinct random seed. In our study, the five different random seeds were employed to control stochasticity across the entire experimental pipeline, encompassing both (a) task-level data split (i.e., the assignment of tasks to meta-training, meta-validation, and meta-testing sets) and (b) model-level randomness (i.e., neural network parameter initialization and stochastic operations during training). This approach enables a thorough and rigorous assessment of model robustness, preventing the evaluation results from being potentially influenced by a single specific data partition or favorable initialization.

During meta-training, the outer loop was run for 10 epochs, with each epoch comprising 100 meta updates. Within each meta-update, the base learner was trained on the support set for 5 epochs in the inner loop. Mean squared error (MSELoss) was used as the loss function, and model parameters were optimized with the Adam optimizer. Detailed hyperparameter configurations (including inner learning rates, outer learning rate, batch size, etc.) and the corresponding performance results are provided in Supplementary Information (Supplementary Table 9 and Supplementary Fig. 2).

## In vitro kinase activity assays

The FLT3 (Cat#V4064) and the FLT3-ITD (Cat#VA7168) were purchased from Promega. Compounds Quizartinib (Cat# T2066), Staurosporine (Cat# T6680) were purchased from Targetmol, USA. All compounds were tested from 100 nM, with 3-fold dilution for 10 points.

In vitro kinase activity was assessed via Promega's ADP - Glo assay (#V9101). For Flt3: Dilute enzyme, substrate, ATP, and compounds in 1X reaction buffer (40 mM Tris, pH 7.5; 2 mM $MnCl_2$; 100 μM sodium vanadate). In a 384-well plate, add 1 μl compound (or 5% DMSO), 2 μl FLT3 enzyme (15 ng/well), and 2 μl substrate/ATP mix (20 μM ATP final). Incubate at 25 °C for 60 min. Add 5 μl ADP - Glo Reagent, incubate 25 °C for 40 min. Then add 10 μl Kinase Detection Reagent, incubate 25 °C for 30 min. Measure luminescence (0.5 s integration) on a SpectraMax iD3 (Molecular Devices). The 50% effective concentration was calculated using Prism 8 by fitting the following equation:

$$Y = Bottom + (Top - Bottom)/(1 + 10^{(\log IC50 - X)} \times HillSlope) \quad (19)$$

where X is a log of concentration, Y is a response, and top and bottom are the resp onses of controls, each assay was repeated at least three times, and we computed the mean and standard deviation for the values.

## Reporting summary

Further information on research design is available in the Nature Portfolio Reporting Summary linked to this article.

## Data availability

The source data of three datasets used to train and evaluate the model is provided in https://github.com/Moohyun-w/AdaMBind/tree/main/data. The source data of the LIT-PCBA dataset is provided in https://drugdesign.unistra.fr/LIT-PCBA/. The support set data used to construct the FLT3 inhibitor prediction task are available at https://doi.org/10.5281/zenodo.13635393[66]. Source data are provided with this paper through https://doi.org/10.6084/m9.figshare.30963823. Source data are provided in this paper.

## Code availability

The source code and data of this study are available at https://github.com/Moohyun-w/AdaMBind. The specific version of the code associated with this publication is archived in Zenodo and is accessible via 10.5281/zenodo.18595084[67]. Data are analyzed using numpy v2.2.6 (https://numpy.org/), pandas v2.3.2 (https://pandas.pydata.org/), Seaborn V0.13.2 (https://seaborn.pydata.org/). Structures are visualized by Pymol v3.1.6 (https://www.pymol.org/) and LigPlot[68] v2.1 (https://www.ebi.ac.uk/). Molecular docking simulations are performed using AutoDock4 v4.2.6 (https://autodock.scripps.edu/).

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

## Acknowledgements

This work was supported by the National Key R&D Program of China (Grant No. 2023YFC2604400, 2024YFA1307700), Shanghai Natural Science Foundation (Grant No. 25ZR1402171), the National Natural Science Foundation of China (62573425), PostGraduate Innovation Fund of Interdiscipline and New Medicine from School of Medicine of Shanghai University. Funding was provided to S.H. (2023YFC2604400, 62573425), X.B. (2024YFA1307700), and Y. Zhao (25ZR1402171).

## Author contributions

M.W., Y. Zhao, H.X., S.H. and X.B. conceived the study; M.W., Y. Zhang performed the experiments; M.W., Y. Zhao., H.X., and D.Y. conducted the surveys and collated the data; M.W., Y. Zhao, and Y. Zhang performed the writing-primer preparation; S.H. and X.B. performed the writing-reviewing and editing, and S.H., P.Z., and X.B. supervised the study. All authors have read and agreed to the published version of the manuscript.

## Competing interests

The authors declare no competing interests.
