## [Transparent Peer Review file · Nature Communications]

A meta learning and task adaptive approach for drug target affinity prediction

Corresponding Author: Professor Song He

Version 0:

Reviewer comments:

Reviewer #1

(Remarks to the Author)

In this manuscript, the authors propose the AdaMBind, which is a meta-learning model for drug-target affinity prediction that integrates a task adaptation module to dynamically prioritize learning tasks. It demonstrates superior performance in few-shot settings, significantly outperforming baselines on multiple datasets, and successfully identifies novel FLT3 inhibitors validated experimentally. However, I believe this manuscript is not yet sufficiently polished for publication, due to issues such as limited experimental comparisons, insufficient descriptions of experimental details, and language imperfections. My comments are as follows:

Major

1. Lines 109–112 describe that “CML (Contrastive Meta-Learning) extends the MAML framework by integrating a contrastive learning module in the forward propagation of the query set to better capture cross-task relevance and improve adaptation to new tasks”. It is unclear why CML was not included as a baseline in this study. I suggest the authors clarify this choice and, if possible, include CML for comparative evaluation.
2. Chen et al. proposed the ZeroBind framework, which achieves zero-shot drug-target prediction through meta-learning and task adaptive attention module, and leverages IB-subgraph learning to automatically identify potential binding pockets, significantly outperforming existing methods across multiple experiments and real-world COVID-19 scenarios. I think the authors should discuss the differences between AdaMBind and ZeroBind, and include a comparative evaluation in the experimental setup.
3. I recommend that the authors consider including the PSICHIC as an additional non-meta-learning baseline. PSICHIC is specifically designed to integrate physicochemical principles when decoding protein–ligand interaction fingerprints directly from sequence data. Unlike prior sequence-based models, it uniquely incorporates these physicochemical constraints, achieving state-of-the-art accuracy and enhanced interpretability. Given that PSICHIC represents a highly advanced sequence-based DTA prediction model, its inclusion would provide a more comprehensive and rigorous comparative evaluation of the proposed approach.
4. It would be helpful for the authors to present the label distributions of the BindingDB, KIBA, and Davis datasets. As far as I am aware, the affinity distributions in KIBA and Davis are highly concentrated. The authors should provide a detailed discussion of this issue, as it may have significant implications for model training, evaluation, and generalization.
5. It is unclear why Pearson correlation was not included among the evaluation metrics, as it is commonly used in related studies.
6. The details of the relevant hyperparameters are not clearly reported. I recommend that the authors provide a table or similar format specifying the optimal hyperparameter settings for transparency and reproducibility.
7. Line 767 mentions that “This noise was sampled from a uniform distribution in the range $[-0.5, 0.5]$.” Intuitively, this level of noise seems relatively large compared with the model's MSE. Did the authors conduct any experiments to test the effect of different noise levels or distributions on model performance? It would be helpful to include such an analysis.
8. Regarding the task-adaptive module, it is unclear in what form the inputs are provided. Moreover, the rationale for using an LSTM is not evident, as there does not appear to be any sequential or temporal relationship in the data. Could the authors clarify this design choice and, if possible, visualize or provide the learned weights to help interpret the module's behavior?
9. The manuscript mentions that the validation set is used to determine the parameters of the adaptive task module. Should this validation set be further divided into support and query subsets, similar to standard meta-learning practice? Clarification on this point would help understand the training and evaluation procedure.
10. The dataset splitting does not appear to consider sequence similarity (e.g., clustering with CD-HIT). The authors may

refer to relevant literature and consider incorporating such a procedure to ensure that the training, validation, and test sets are appropriately non-redundant, which would provide a more rigorous evaluation of model generalization.

11. Regarding the use of five different random seeds, it is unclear whether these correspond to different dataset splits or only to variations in network initialization and training. The authors should clarify this, as it impacts the interpretation of the reported performance variability.

12. It is unclear how task similarity is defined in this study. Additionally, the use of molecule-averaged fingerprints warrants further explanation—what is the physical or chemical significance of averaging the fingerprints across molecules?

13. When modeling molecules with GNNs, it is unclear why edge information (e.g., bond types) is not considered.

14. Regarding the LIT-PCBA dataset and the associated experimental setup, it is unclear whether the training data consist of only 13 tasks \times 15 active samples. Since virtual screening is a common task, the authors should clarify the performance levels of the reported EF and precision values, and discuss how they compare with existing studies. Additionally, it is unclear why BEDROC ($\alpha = 80.5$), a widely used metric for early recognition in virtual screening, was not considered. Including this metric or discussing its omission would strengthen the evaluation.

15. I suggest providing the predicted affinity values for all compounds listed in Table 1. It is also unclear why experimental validation was not performed for the other compounds. A more detailed analysis of the binding modes would help interpret the predictions and strengthen the study.

Minor

1. In line 32, "IC50" is incorrectly written as "lC50". This typo appears throughout the manuscript, and the authors should correct all instances to ensure consistency and accuracy.

2. In line 205, it is stated that "In the testing phase, the trained meta-learner is used to predict drug–target binding affinities in the majority setting (40 known DTA pairs) or few-shot setting (5 known DTA pairs)." This description seems inaccurate. Shouldn't the meta-learner be fine-tuned on the support set during testing, as is standard in meta-learning? Clarification on this point is needed.

3. In line 608, the manuscript mentions "additional drugs whose affinities are unknown." However, the samples in the query set appear to have known labels. The authors should clarify this apparent inconsistency.

4. In Figure 1, "combanation" is a typo and should be corrected to "combination."

5. In line 317, "Meta-Learng" is a typo and should be corrected to "Meta-Learning."

6. In line 155, "Li dataset" should be corrected to "Li's dataset" to indicate proper possession.

7. Throughout the manuscript, "Lit-PCBA" should be corrected to "LIT-PCBA" to maintain consistent capitalization of the dataset name.

8. In line 668, "Huaxiu Yao et al." should be corrected to "Yao et al."

9. In line 713, it is mentioned that "we converted the original Kd and IC50 values into their logarithmic forms (pKd and pIC50)." The authors should also include Ki values.

10. In line 412, it is mentioned "and Davis datasets (Supplementary Fig. 2)." However, other supplementary information is not referenced in the main text, and there appears to be no Supplementary Fig. 1. The authors should ensure that all supplementary figures are properly cited and included.

11. In line 727, "coefficient of determination (r^2)" should be written as " R^2 " to follow standard notation.

12. In line 762, it is mentioned that "During meta-training, the outer loop was run for 10 episodes, and each episode included 5 epochs of training in the inner loop." This number of training iterations seems relatively small. The authors should clarify whether this is sufficient and, if possible, provide training curves to support the choice of training schedule.

(Remarks on code availability)

I did not run the code, README file seems ok

Reviewer #2

(Remarks to the Author)

This manuscript presents AdaMBind, an innovative drug-target affinity prediction model that integrates the MAML meta-learning framework with a task adaptation mechanism. By adopting an "easy-to-hard" learning strategy, AdaMBind effectively enhances model performance under low-data conditions. The authors conduct comprehensive evaluations across three benchmark datasets, under both majority- and few-shot settings, demonstrating consistent superiority over existing baseline methods. Notably, the study extends beyond computational validation by applying AdaMBind to identify candidate FLT3 inhibitors, culminating in experimental confirmation that Staurosporine exhibits potent inhibitory activity against FLT3 with a significantly improved IC₅₀ compared to the positive control. This real-world application underscores the practical utility of the proposed method. Overall, the integration of meta-learning with adaptive task selection is both innovative and well-motivated. The experimental validation is thorough, and the manuscript is clearly written. However, the authors should consider the following comments.

Comments

1. The manuscript needs more details on hyperparameter optimization. The authors should clarify their hyperparameter selection process and training procedures, especially regarding how they optimized the parameters of different components of AdaMBind. This will clarify the effects of hyperparameter tuning on model performance across various datasets.

2. As meta-learning frameworks are highly sensitive to hyperparameter settings, the authors are strongly encouraged to include a comprehensive list of all key hyperparameters in the Supplementary Materials. Moreover, more statements or illustrations should be added to clarify why the meta-learning framework helps the prediction.

3. Label noise injection is mentioned in the method section as a means to enhance robustness, but the potential biases it may introduce are not discussed. It is recommended to incorporate an analysis of the impact of noise levels in the ablation studies.

4. Although AdaMBind does not require 3D structural information, which is advantageous for targets with unknown or poorly resolved structures, it may encounter difficulties when applied to highly flexible proteins, such as GPCRs or ion channels, where conformational dynamics play a crucial role in ligand binding. The authors are encouraged to discuss these limitations and boundary conditions in the "Discussion" section. Additionally, they could briefly outline potential future directions, such as incorporating conformational ensembles, molecular dynamics features, or uncertainty estimation, to position the work within the evolving landscape of DTA prediction.
5. Figure 4a presents the ablation study results, but does not include error bars or statistical measures. The authors should report variance metrics (e.g., standard deviation over five independent runs) to support the reliability of the observed performance differences.
6. In the context of the FLT3 inhibitor validation, the PDB ID of the FLT3 structure employed in the docking simulation should be explicitly stated.
7. The manuscript employs Enrichment Factor (EF@1%, EF@5%) as a key metric in the validation of ESR/TP53, but it fails to define the EF.
8. To enhance the scholarly relevance and positioning of this work, the authors are advised to update the reference list to include recent advances from 2023 to 2025, particularly in the areas of meta-learning for biomedicine and DTA modeling.
9. Two instances of the word "Table" appear on line 500.
10. Several grammatical errors were observed (e.g., inconsistent tenses in the abstract). A thorough proofreading is recommended.

(Remarks on code availability)

The authors provided comprehensive code accompanied by well-documented instructions, which offers readers a good opportunity for reproducibility of their work.

In addition, I suggest that the authors provide case codes showing how to apply AdaMBind in the Table of Contents.

Reviewer #3

(Remarks to the Author)

The manuscript by Wan et al, provides a tool "AdaMbind" for accurate and robust drug target affinity prediction.

Authors have developed a new tool for DTA prediction to address the limitations of the current methods. Two core modules were developed – meta learning and task adaptive. Authors evaluated it under majority and few shot learning setting, using the two settings allows for detection of drug-target patterns and drug target binding pairs. ADAMBind outperformed as compared to existing models. Further testing in AML from drug bank identified staurosporine as FLT3 inhibitor which may hold relevance if significant beyond the kinase activity assay.

Concerns:

The model though outperforms current models still has dependency on the known structures or binding affinities. This should be elaborated more.

Figure 2. shows the outperformance of AdaMbind as compared to other models, but difference in improvement varies in different comparisons.

Number of unconventional abbreviations used makes it difficult to read the manuscript.

T

he studies on FLT3 to identify compounds with interactions with FLT 3 is interesting, but will need testing in context of FLT3-ITD positive and FLT3-WT staus to establish its therapeutic relevance in AML.

(Remarks on code availability)

NA

Version 1:

Reviewer comments:

Reviewer #1

(Remarks to the Author)

The authors extensively revised the manuscript and added more experiments and baselines, and addressed all my comments.

(Remarks on code availability)

Reviewer #2

(Remarks to the Author)

Based on the authors' detailed and comprehensive responses to my comments, I am pleased to recommend the acceptance of the manuscript titled "AdaMBind: A Drug-Target Affinity Prediction Model Integrating MAML Meta-Learning with a Task Adaptation Mechanism" for publication. The authors have addressed all my concerns thoroughly and have made significant

improvements to the manuscript, enhancing its clarity, rigor, and overall impact.

Below, I provide a more detailed evaluation of the key improvements:

First, the authors provided a meticulous, point-by-point response to every comment. The authors conducted additional experiments (e.g., the ablation study on noise robustness) and provided extensive new details (e.g., the hyperparameter tuning process) to substantiate their methodology. This significantly boosts the reproducibility and reliability of the reported results.

Second, the inclusion of a comprehensive hyperparameter table and a detailed description of the grid search process effectively addresses the initial lack of clarity regarding parameter optimization. The addition of a Jupyter notebook with a complete workflow demonstration to their code repository is a good practice that will greatly benefit the research community and facilitate the adoption of their method.

Third, the revised Discussion section now thoughtfully addresses the model's limitations, particularly concerning conformationally flexible proteins, and outlines clear, promising future directions. Furthermore, the inclusion of recent literature in the Introduction effectively positions the study within the current state-of-the-art, enhancing its relevance.

In conclusion, the integration of meta-learning with an adaptive task selection mechanism remains innovative and well-motivated. The revisions have thoroughly addressed all methodological concerns, bolstered the experimental evidence, and improved the manuscript's presentation and scholarly context. I have no further reservations and recommend acceptance in its current form.

(Remarks on code availability)

Reviewer #3

(Remarks to the Author)

My critiques has been addressed by the authors. Authors have done additional work to address the comments and in its current form the manuscript have improved significantly from the previous version.

(Remarks on code availability)

NA

Response to Referees Letter

Title:AdaMBind: Meta-learning and task adaptation based drug-target affinity prediction

Response to Reviewers' Comments

Reviewer #1:

In this manuscript, the authors propose the AdaMBind, which is a meta-learning model for drug-target affinity prediction that integrates a task adaptation module to dynamically prioritize learning tasks. It demonstrates superior performance in few-shot settings, significantly outperforming baselines on multiple datasets, and successfully identifies novel FLT3 inhibitors validated experimentally. However, I believe this manuscript is not yet sufficiently polished for publication, due to issues such as limited experimental comparisons, insufficient descriptions of experimental details, and language imperfections. My comments are as follows:

Thank you for the efforts you have put into reviewing our manuscript. In accordance with your suggestions, we have revised the manuscript from four aspects (**Parts 1-4** below).

1) Newly added models and the performance comparative study.

a. For comparison with AdaMBind, we introduced CML (Contrastive Meta-Learning), ZeroBind, and PSICHIC as baseline models. [**Major Comment 1-3**]. (Page 56, Line 1017-1031, Page 13-26, Line 230-452 in revised manuscript).

b. We added the Pearson Correlation Coefficient as an evaluation metric across all relevant experiments [**Major Comment 5**].

2) Expanded description of experimental implementation and methodological design.

a. **Expanded description of the hyperparameter settings.** We provided a detailed table specifying the optimal hyperparameter configurations for experiments. In

addition, we provided detailed information on the hyperparameter selection process and training procedures [**Major Comment 6**]. (Table S9 and Figure S2 in the revised Supplementary Information)

b. Newly added ablation study on noise robustness. We conducted an ablation study to test the model's performance under different noise levels and distributions. [**Major Comment 7**]. (Table S9 and Figure S2 in the revised Supplementary Information)

c. Expanded description of rationale and implementation details of the adaptive task module. We provided the input format for the task adaptation module and the rationale for using LSTMs, and visualized the module's behavior to explain it. [**Major Comment 8**]. Furthermore, we clarified how the validation set determines the parameters of the adaptive task module [**Major Comment 9**]. (Page 50, Line 934-940 in revised manuscript).

d. Clarification of the application of the random seeds. We clarified that five random seeds collectively control both data splits and model initialization/training randomness, allowing for a comprehensive assessment of model robustness [**Major Comment 11**]. (Page 57-58, Line 1046-1063 in revised manuscript).

e. Clarification of task similarity characterization. We explained the rationale behind the definition of task similarity in this study. In addition, we replaced averaged molecular fingerprints with the more physicochemically meaningful averaged UniMol representation, in order to generate a more accurate consensus representation of the task's chemical space. [**Major Comment 12**] (Page 28-32, Line 485-564 in revised manuscript).

f. Explanation of GNN does not take into account edge information. We clarified that the core contribution of AdaMBind lies in proposing a novel meta-learning framework to address few-shot DTA prediction. Therefore, we constructed a structurally simple yet stable base learner that relies solely on atom-level node features and graph topology. In addition, AdaMBind features a model-agnostic and flexible design, allowing for seamless compatibility with more sophisticated molecular encoders in the future. [**Major Comment 13**] (Page 45, Line 779-798 in revised

manuscript).

3) Supplementation of experimental results and analyses

We enriched the presentation and discussion of results to provide deeper insights and a more rigorous evaluation.

a. Newly added description of dataset characterization. We analyzed the label distributions across all three datasets and their impact on model training and evaluation. In addition, to further assess the generalization implications, we performed cross-domain experiments that explicitly tested the model's ability to generalize across different data distributions. **[Major Comment 4].** (Page 21, Line 354-382 in revised manuscript)

b. Newly added novel task split scenarios for rigorous generalization evaluation. We incorporated **novel task split scenarios** to evaluate model generalization. This split is based on protein sequence similarity clustering (using CD-HIT), implementing a more rigorous dataset division strategy to ensure low similarity between the training, validation, and test sets. **[Major Comment 10]** (Page 17-21, Line 302-352 and Page 22-26, Line 383-452 in revised manuscript)

c. Enhanced virtual screening evaluation on LIT-PCBA dataset. Firstly, we clarified the experimental setup for the LIT-PCBA benchmark, explicitly stating the composition of the training data. Secondly, the performance levels of the reported Enrichment Factor (EF) and BEDROC are now discussed in the context of existing studies. Finally, the widely used BEDROC ($\alpha=80.5$) metric for early recognition has been added to the evaluation. **[Major Comment 14]** (Page 35-37, Line 622-646 in revised manuscript).

d. Clarification of experimental validation about potent inhibitors of FLT3. The predicted affinity values for all compounds listed in Table 1 are now provided in the supplementary information. The rationale for selecting specific compounds for experimental validation has been elaborated to clarify the selection process **[Major Comment 15].**(Page 40-41, Line 691-730 in revised manuscript)

4) Made corrections of inaccurate expressions, grammar, and representations of figures and tables. [Minor Comments 1-12]

Below, we provide a point-by-point response to each suggestion. All modifications in the revised manuscript are highlighted in red text.

Major

Points

1. Lines 109–112 describe that “CML (Contrastive Meta-Learning) extends the MAML framework by integrating a contrastive learning module in the forward propagation of the query set to better capture cross-task relevance and improve adaptation to new tasks”. It is unclear why CML was not included as a baseline in this study. I suggest the authors clarify this choice and, if possible, include CML for comparative evaluation.

Response:

Thank you for the valuable suggestion. We added CML as an additional baseline model for comparative evaluation and subsequently analyzed the experimental results.

(1) Implementation and Supplementary Details of Baseline Models

Thank you for the valuable comment regarding the exclusion of Contrastive Meta-Learning (CML) from our initial baseline studies. We would like to clarify that the original authors of CML have not released their code publicly, and our attempts to contact them via email have received no response. For the purpose of comparison, we reimplemented CML based on the descriptions in the original paper. In accordance with the reviewer’s suggestion, we have now added comparative experiments with CML in the revised manuscript. The performance results on the three datasets under the random task split are presented in Table R38-Table R43 (corresponding to Fig.2 in the revised manuscript). In addition, in response to your major comment 10, we considered sequence similarity and designed a novel task split strategy to evaluate the model’s generalization capability. The results under the novel task split are also provided in Table R53-Table R58 (corresponding to Fig.3 in the revised manuscript).

(2) Comparison with Original Results

To ensure the accuracy and comparability of the model reproduction, we strictly followed the data splitting scheme and hyperparameter settings described in the original CML paper to retrain our independently reproduced model. Based on the Davis dataset

and using the three core performance metrics from the original study (CI, r_m^2 , Pearson), we systematically compared the reproduced results with the originally reported results. As shown in Table R1, the differences between the reproduced results and the original results for these three key metrics all fall within an acceptable range of experimental error. This confirms the reliability of the present reproduction study and attests to the reproducibility of the original research methodology. In the revised version, we have clarified that CML was re-implemented by us. (corresponding to Line 1024-1028 on Page 56 in the revised manuscript)

Table R1 Comparison with Original CML Results on Davis dataset

Methods	CI(↑)	r_m^2 (↑)	Pearson(↑)
Original CML results	0.799 (0.011)	0.301 (0.013)	0.563 (0.010)
Reproduced CML Results	0.762 (0.020)	0.289 (0.048)	0.559 (0.076)

Five independent replications of each method were performed ($n = 5$). Data are expressed as means (std).

(3) Comparative evaluation of AdaMBind and CML

We adopted two task-partitioning strategies to systematically evaluate the performance of AdaMBind and CML under different levels of task novelty.

The random task split strategy partitions all tasks randomly into meta-training, meta-validation, and meta-test sets at a ratio of 8:1:1.

The novel task split strategy clusters all tasks based on protein sequence similarity using CD-HIT with a stringent 40% sequence identity threshold, and allocates the resulting clusters to training, validation, and test sets (see Major Comment 10 for details). This partitioning ensures that the proteins in the test set show substantial sequence divergence from those in the training set, thereby constructing a more challenging evaluation scenario. This scenario simulates the real-world requirement for predicting the affinity of novel targets with low sequence similarity and markedly divergent structures.

Based on these two task-split strategies, the experimental results are analyzed as follows.

a. Performance comparison of CML and AdaMBind under random task split

Under the majority setting (Table R2-Table R4), AdaMBind demonstrates consistent and substantial performance advantages over CML across all three datasets.

Table R2 Performance on Davis in random task split (Majority)

Methods	MSE(↓)	CI(↑)	R2(↑)	Spearman(↑)	Pearson(↑)
CML	0.5295 (0.2405)	0.7850 (0.1238)	0.3517 (0.0307)	0.5788 (0.0393)	0.5892 (0.0351)
AdaMBind(ours)	0.5060 (0.1672)	0.8222 (0.0160)	0.4693 (0.0538)	0.6257 (0.0389)	0.6878 (0.0385)

The best results are highlighted in bold. Five independent replications of each method were performed (n = 5). Data are expressed as means (std).

Table R3 Performance on KIBA in random task split (Majority)

Methods	MSE(↓)	CI(↑)	R2(↑)	Spearman(↑)	Pearson(↑)
CML	0.6734 (0.0805)	0.6549 (0.0330)	0.0879 (0.0655)	0.4196 (0.0813)	0.3597 (0.0550)
AdaMBind(ours)	0.4295 (0.0566)	0.7545 (0.0245)	0.4285 (0.0392)	0.6528 (0.0545)	0.6559 (0.0297)

The best results are highlighted in bold. Five independent replications of each method were performed (n = 5). Data are expressed as means (std).

Table R4 Performance on BindingDB in random task split (Majority)

Methods	MSE(↓)	CI(↑)	R2(↑)	Spearman(↑)	Pearson(↑)
CML	1.0017 (0.0814)	0.7595 (0.0194)	0.3946 (0.0506)	0.5735 (0.0340)	0.6392 (0.0352)
AdaMBind(ours)	0.8578 (0.1077)	0.7843 (0.0221)	0.4815 (0.0657)	0.6578 (0.0413)	0.6966 (0.0430)

The best results are highlighted in bold. Five independent replications of each method were performed (n = 5). Data are expressed as means (std).

Specifically, on the Davis dataset, CML achieved MSE (0.5259), CI (0.7850), R² (0.3517), Spearman (0.5788), and Pearson (0.5892). In comparison, AdaMBind attained MSE (0.5060), CI (0.8222), R² (0.4693), Spearman (0.6257), and Pearson (0.6878). This represents an improvement of 4.4% in MSE, 4.7% in CI, 33.4% in R², 8.1% in Spearman, and 16.7% in Pearson.

On the KIBA dataset, CML achieved MSE (0.6734), CI (0.6549), R² (0.0879), Spearman (0.4196), and Pearson (0.3597), while AdaMBind achieved MSE (0.4295), CI (0.7545), R² (0.4285), Spearman (0.6528), and Pearson (0.6559), corresponding to relative improvements of 36.2% in MSE, 15.2% in CI, 387.5% in R², 55.6% in Spearman, and 82.3% in Pearson correlation.

On the BindingDB dataset, CML obtained MSE (1.0017), CI (0.7595), R² (0.3946), Spearman (0.5735), and Pearson (0.6392). AdaMBind improved these results to MSE (0.8578), CI (0.7843), R² (0.4815), Spearman (0.6578), and Pearson (0.6966), reflecting improvements of 14.4% in MSE, 3.3% in CI, 22.0% in R², 14.7% in Spearman, and 9.0% in Pearson correlation.

Similarly, under the few-shot setting (Table R5-Table R7), AdaMBind also demonstrates better performance over CML across all three datasets.

Table R5 Performance on Davis in random task split (Few-shot)

Methods	MSE(↓)	CI(↑)	R2(↑)	Spearman(↑)	Pearson(↑)
CML	0.8138 (0.1723)	0.6474 (0.0204)	0.0689 (0.0589)	0.2874 (0.0510)	0.2728 (0.0708)
AdaMBind(ours)	0.4744 (0.0216)	0.7507 (0.0450)	0.2732 (0.0314)	0.4904 (0.0205)	0.5499 (0.0187)

The best results are highlighted in bold. Five independent replications of each method were performed (n = 5). Data are expressed as means (std).

Table R6 Performance on KIBA in random task split (Few-shot)

Methods	MSE(↓)	CI(↑)	R2(↑)	Spearman(↑)	Pearson(↑)
CML	0.7713 (0.1363)	0.6272 (0.0253)	-0.0463 (0.0864)	0.3457 (0.0669)	0.2636 (0.0686)
AdaMBind(ours)	0.5715 (0.0806)	0.6802 (0.0185)	0.2379 (0.0769)	0.4847 (0.0504)	0.5062 (0.0653)

The best results are highlighted in bold. Five independent replications of each method were performed (n = 5). Data are expressed as means (std).

Table R7 Performance on BindingDB in random task split (Few-shot)

Methods	MSE(↓)	CI(↑)	R2(↑)	Spearman(↑)	Pearson(↑)
CML	1.3982 (0.1782)	0.6965 (0.0356)	0.2029 (0.0941)	0.4637 (0.0862)	0.5050 (0.0780)
AdaMBind(ours)	1.2116 (0.1255)	0.7260 (0.0292)	0.3094 (0.0629)	0.5323 (0.0758)	0.5634 (0.0590)

The best results are highlighted in bold. Five independent replications of each method were performed (n = 5). Data are expressed as means (std).

On the Davis dataset, CML reported MSE (0.8138), CI (0.6474), R² (0.0689), Spearman (0.2874), and Pearson (0.2728). AdaMBind significantly outperformed with MSE (0.4744), CI (0.7507), R² (0.2732), Spearman (0.4904), and Pearson (0.5499), corresponding to improvements of 41.7% in MSE, 16.0% in CI, 70.6% in Spearman, and 101.6% in Pearson correlation.

On the KIBA dataset under few-shot conditions, CML achieved MSE (0.7713), CI (0.6272), R² (-0.0463), Spearman (0.3457), and Pearson (0.2636), while AdaMBind attained MSE (0.5715), CI (0.6802), R² (0.2379), Spearman (0.4847), and Pearson (0.5062), representing gains of 25.9% in MSE, 8.5% in CI, 40.2% in Spearman and 92.1% in Pearson correlation.

On the BindingDB dataset under the few-shot setting, CML produced MSE (1.3982), CI (0.6965), R² (0.2029), Spearman (0.4637), and Pearson (0.5050). AdaMBind improved these metrics to MSE (1.2116), CI (0.7260), R² (0.3094), Spearman (0.5323), and Pearson (0.5634), demonstrating relative increases of 13.4% in MSE, 4.2% in CI, 52.5% in R², 14.8% in Spearman, and 11.6% in Pearson correlation.

b. Performance comparison of CML and AdaMBind under Novel Task Split

Under the majority setting (Table R8-Table R10), AdaMBind demonstrated superior performance to CML across all three datasets.

Table R8 Performance on Davis in novel task split (Majority)

Methods	MSE(↓)	CI(↑)	R2(↑)	Spearman(↑)	Pearson(↑)
CML	0.6301 (0.2050)	0.7680 (0.0141)	0.3117 (0.0798)	0.5288 (0.0525)	0.5592 (0.0723)
AdaMBind(ours)	0.5080 (0.1823)	0.8066 (0.0271)	0.4460 (0.0808)	0.5978 (0.0633)	0.6702 (0.0623)

The best results are highlighted in bold. Five independent replications of each method were performed (n = 5). Data are expressed as means (std).

Table R9 Performance on KIBA in novel task split (Majority)

Methods	MSE(↓)	CI(↑)	R2(↑)	Spearman(↑)	Pearson(↑)
CML	0.5332 (0.0773)	0.6605 (0.0121)	0.1120 (0.0458)	0.4289 (0.0274)	0.3698 (0.0476)
AdaMBind(ours)	0.3626 (0.0368)	0.7495 (0.0307)	0.3738 (0.1017)	0.6279 (0.0739)	0.6239 (0.0701)

The best results are highlighted in bold. Five independent replications of each method were performed (n = 5). Data are expressed as means (std).

Table R10 Performance on BindingDB in novel task split (Majority)

Methods	MSE(↓)	CI(↑)	R2(↑)	Spearman(↑)	Pearson(↑)
CML	1.0545 (0.068)	0.7323 (0.0260)	0.2987 (0.0964)	0.5287 (0.0993)	0.5567 (0.0805)
AdaMBind(ours)	0.8758 (0.0374)	0.7689 (0.0348)	0.4114 (0.1199)	0.6064 (0.1205)	0.6407 (0.0869)

The best results are highlighted in bold. Five independent replications of each method were performed (n = 5). Data are expressed as means (std).

On the Davis dataset, CML achieved the following performance: MSE (0.6301), CI (0.7680), R² (0.3117), Spearman correlation (0.5288), and Pearson correlation (0.5592). In comparison, AdaMBind attained MSE (0.5080), CI (0.8066), R² (0.4460), Spearman (0.5978), and Pearson (0.6702), which corresponds to relative improvements of 19.4% in MSE, 5.0% in CI, 43.1% in R², 13.0% in Spearman, and 19.9% in Pearson correlation.

On the KIBA dataset under the majority setting, CML yielded MSE (0.5332), CI (0.6605), R² (0.1120), Spearman (0.4289), and Pearson (0.3698), whereas AdaMBind achieved MSE (0.3626), CI (0.7495), R² (0.3738), Spearman (0.6279), and Pearson (0.6239), representing improvements of 32.0% in MSE, 13.5% in CI, 46.4% in Spearman, and 68.7% in Pearson correlation.

On the BindingDB dataset, CML obtained MSE (1.0545), CI (0.7323), R² (0.2987), Spearman (0.5287), and Pearson (0.5567). AdaMBind improved these results to MSE (0.8758), CI (0.7689), R² (0.4114), Spearman (0.6064), and Pearson (0.6407), reflecting improvements 17.0% in MSE, 5.0% in CI, 37.7% in R², 14.7% in Spearman, and 15.1% in Pearson correlation.

In few-shot setting (Table R11-Table R13), AdaMBind still demonstrates better performance over CML across all three datasets.

Table R11 Performance on Davis in novel task split (Few-shot)

Methods	MSE(↓)	CI(↑)	R2(↑)	Spearman(↑)	Pearson(↑)
CML	0.7499 (0.1971)	0.6556 (0.0190)	0.0627 (0.0579)	0.3028 (0.0582)	0.2957 (0.0654)
AdaMBind(ours)	0.6128 (0.2205)	0.7591 (0.0186)	0.2526 (0.1101)	0.4926 (0.0971)	0.5350 (0.0900)

The best results are highlighted in bold. Five independent replications of each method were performed (n = 5). Data are expressed as means (std).

Table R12 Performance on KIBA in novel task split (Few-shot)

Methods	MSE(↓)	CI(↑)	R2(↑)	Spearman(↑)	Pearson(↑)
CML	0.7863 (0.1335)	0.5992 (0.0418)	-0.1383 (0.0940)	0.2675 (0.1068)	0.2296 (0.0798)
AdaMBind(ours)	0.4910 (0.1403)	0.6808 (0.0442)	0.2983 (0.1101)	0.4717 (0.0971)	0.5543 (0.0900)

The best results are highlighted in bold. Five independent replications of each method were performed (n = 5). Data are expressed as means (std).

Table R13 Performance on BindingDB in novel task split (Few-shot)

Methods	MSE(↓)	CI(↑)	R2(↑)	Spearman(↑)	Pearson(↑)
CML	1.4191 (0.2578)	0.6926 (0.0232)	0.1898 (0.0859)	0.4529 (0.0451)	0.5012 (0.0631)
AdaMBind(ours)	1.2968 (0.2389)	0.6980 (0.0169)	0.2536 (0.0352)	0.4703 (0.0505)	0.5171 (0.0421)

The best results are highlighted in bold. Five independent replications of each method were performed (n = 5). Data are expressed as means (std).

On the Davis dataset, CML achieved MSE (0.7499), CI (0.6556), R² (0.0627), Spearman (0.3028), and Pearson (0.2957). AdaMBind attained MSE (0.6128), CI (0.7591), R² (0.2526), Spearman (0.4926), and Pearson (0.5350), corresponding to improvements of 18.3% in MSE, 15.8% in CI, 62.7% in Spearman, and 80.9% in Pearson correlation.

On the KIBA dataset, CML achieved MSE (0.7863), CI (0.5992), R² (-0.1383), Spearman (0.2675), and Pearson (0.2296), while AdaMBind attained MSE (0.4910), CI (0.6808), R² (0.2983), Spearman (0.4717), and Pearson (0.5543), representing

improvements of 37.6% in MSE, 13.6% in CI, 76.3% in Spearman correlation.

On the BindingDB dataset, CML achieved MSE (1.4191), CI (0.6926), R^2 (0.1898), Spearman (0.4529), and Pearson (0.5012). AdaMBind attained MSE (1.2968), CI (0.6980), R^2 (0.2536), Spearman (0.4703), and Pearson (0.5171), demonstrating relative increases of 8.6% in MSE, 0.8% in CI, 33.6% in R^2 , 3.8% in Spearman, and 3.2% in Pearson correlation.

Overall, across both task split strategies, AdaMBind consistently and significantly outperforms CML on all datasets (Davis, KIBA, BindingDB) under both sample settings (majority and few-shot), demonstrating its stronger generalization capability. Particularly in the more challenging few-shot novel task scenario, the advantage of AdaMBind becomes even more pronounced.

2. Chen et al. proposed the ZeroBind framework, which achieves zero-shot drug-target prediction through meta-learning and task adaptive attention module, and leverages IB-subgraph learning to automatically identify potential binding pockets, significantly outperforming existing methods across multiple experiments and real-world COVID-19 scenarios. I think the authors should discuss the differences between AdaMBind and ZeroBind, and include a comparative evaluation in the experimental setup.

Response:

Thank you for the valuable suggestion. We added ZeroBind as an additional baseline model for comparative evaluation and subsequently conducted an in-depth discussion on the methodological differences between AdaMBind and ZeroBind.

(1) Incorporation of ZeroBind as baseline

It should be noted that ZeroBind is designed for drug target interaction prediction, which is a classification model. To ensure a fair comparison, we maintained the original architecture of ZeroBind but replaced its output head with a regression layer, then trained and evaluated ZeroBind under identical dataset splits. The performance results on the three datasets under the random task split are presented in Table R38-Table R43 (corresponding to Fig. 2 in the revised manuscript). Similarly, the results under the novel task split are also provided in Table R53-Table R58 (corresponding to Fig.3 in

the revised manuscript).

(2) Extended analysis of the difference between ZeroBind and AdaMBind

We conducted a comparative analysis of ZeroBind and CML from two perspectives: the task sampling strategy in meta-learning framework and the input representation.

A. Fundamental difference in task sampling strategy

ZeroBind: Static sampling, dynamic weighting. ZeroBind follows a two-stage process. In the sampling stage (static), a batch of tasks is sampled uniformly at random from a fixed task pool at the beginning of each training episode, giving every task an equal probability of selection. Then, in the weighting stage (dynamic), after the loss for each task is computed, a “task-adaptive self-attention module” assigns different weights to each task’s loss. These weighted losses are subsequently used to update the meta-learner. The core idea can be summarized as: While every task is given equal attention during sampling, we assign different levels of confidence to its learning outcomes.

AdaMBind: Dynamic sampling, dynamic construction. By contrast, AdaMBind employs a dynamic selection mechanism. During the sampling stage (dynamic), before each training episode, all candidate tasks are proactively evaluated and assigned a sampling probability via query set loss and gradient similarity. The training batch is then sampled in a biased manner according to this probability distribution. Consequently, in the construction stage (dynamic), each training batch is dynamically assembled from tasks that are currently deemed “more valuable”. The idea is: we should prioritize feeding the model with task samples that most effectively facilitate its learning.

The rationale behind our choice of this strategy lies in its inherent capability to proactively identify and suppress the sampling probability of noisy or outlier tasks, thereby preventing them from interfering with the meta-learner's optimization process and directly enhancing the model's generalization robustness.

B. Difference in dependency on protein structural information

Another fundamental distinction lies in their dependency on protein structural information. ZeroBind exhibits strong reliance on 3D structural data, whereas

AdaMBind depends solely on sequence information. This divergence in input requirements constitutes a key factor influencing their respective applicability and computational efficiency.

ZeroBind: Strong dependency on 3D structure. ZeroBind core innovation, the Subgraph Information Bottleneck (SIB) module, necessitates representing the protein as a 3D structural graph, where nodes correspond to amino acid residues and edges are formed based on spatial proximity. This enables the model to identify potential binding pockets from such structural representations. However, a limitation of this approach is that its performance may be constrained for proteins lacking high-resolution experimental structures or those with low-confidence AlphaFold-predicted models, such as membrane proteins or intrinsically disordered regions. Additionally, processing 3D structural graphs involves higher computational overhead.

AdaMBind: Relies solely on sequence information. AdaMBind uses the amino acid sequence of the protein as input, extracting features through a 1D CNN without any dependency on 3D structural information. This design confers a key advantage: broader applicability, as sequence information is available for virtually all known proteins. Consequently, AdaMBind can be rapidly and widely deployed for any target with a known sequence, **proving particularly valuable in the early stages of novel target discovery when reliable structural data is often unavailable.** We adopted the strategy of relying solely on sequence information based on in-depth consideration of real-world drug discovery scenarios.

The analysis between the two methods has been added to **the *Instruction*** of the revised manuscript. (The corresponding revision can be seen Line 123-128, Line 131-134 on Page 7 of the revised manuscript)

(3) Comparative evaluation of AdaMBind and ZeroBind

Similar to **comment 1**, we adopted two task-partitioning strategies to systematically evaluate the performance of AdaMBind and ZeroBind under different levels of task novelty. Including the **random task split strategy and novel task split strategy.**

a. Performance comparison of ZeroBind and AdaMBind under random task split

Table R14 Performance on Davis in random task split (Majority)

Methods	MSE(↓)	CI(↑)	R2(↑)	Spearman(↑)	Pearson(↑)
ZeroBind	0.5923 (0.1865)	0.7637 (0.0452)	0.3448 (0.0584)	0.5179 (0.0454)	0.5489 (0.0513)
AdaMBind(ours)	0.5060 (0.1672)	0.8222 (0.0160)	0.4693 (0.0538)	0.6257 (0.0389)	0.6878 (0.0385)

The best results are highlighted in bold. Five independent replications of each method were performed (n = 5). Data are expressed as means (std).

Table R15 Performance on KIBA in random task split (Majority)

Methods	MSE(↓)	CI(↑)	R2(↑)	Spearman(↑)	Pearson(↑)
ZeroBind	0.5727 (0.0706)	0.6755 (0.0287)	0.2484 (0.0299)	0.4562 (0.0495)	0.4810 (0.0277)
AdaMBind(ours)	0.4295 (0.0566)	0.7545 (0.0245)	0.4285 (0.0392)	0.6528 (0.0545)	0.6559 (0.0297)

The best results are highlighted in bold. Five independent replications of each method were performed (n = 5). Data are expressed as means (std).

Table R16 Performance on BindingDB in random task split (Majority)

Methods	MSE(↓)	CI(↑)	R2(↑)	Spearman(↑)	Pearson(↑)
ZeroBind	1.2613 (0.1306)	0.7379 (0.0316)	0.3491 (0.0557)	0.5675 (0.0463)	0.6176 (0.0531)
AdaMBind(ours)	0.8578 (0.1077)	0.7843 (0.0221)	0.4815 (0.0657)	0.6578 (0.0413)	0.6966 (0.0430)

The best results are highlighted in bold. Five independent replications of each method were performed (n = 5). Data are expressed as means (std).

Under the majority setting (Table R14-Table R16), AdaMBind outperformed ZeroBind across all metrics on all three datasets.

On the Davis dataset, ZeroBind achieved the following performance: MSE (0.5923), CI (0.7637), R^2 (0.3448), Spearman(0.5179), and Pearson (0.5489). In comparison, AdaMBind attained MSE (0.5060), CI (0.8222), R^2 (0.4693), Spearman (0.6257), and Pearson (0.6878), which corresponds to relative improvements of 14.6% in MSE, 7.7% in CI, 36.1% in R^2 , 20.8% in Spearman, and 25.3% in Pearson correlation.

On the KIBA dataset under the same majority setting, ZeroBind yielded MSE (0.5727), CI (0.6755), R^2 (0.2484), Spearman (0.4562), and Pearson (0.4810), whereas

AdaMBind achieved MSE (0.4295), CI (0.7545), R² (0.4285), Spearman (0.6528), and Pearson (0.6559), representing improvements of 25.0% in MSE, 11.7% in CI, 72.5% in R², 43.1% in Spearman, and 36.4% in Pearson correlation.

On the BindingDB dataset, ZeroBind obtained MSE (1.2613), CI (0.7379), R² (0.3491), Spearman (0.5675), and Pearson (0.6176). AdaMBind improved these results to MSE (0.8578), CI (0.7843), R² (0.4815), Spearman (0.6578), and Pearson (0.6966), reflecting improvements of 32.0% in MSE, 6.3% in CI, 37.9% in R², 15.9% in Spearman, and 12.8% in Pearson correlation.

Under the few-shot setting (Table R17-Table R19), AdaMBind also outperformed ZeroBind across all metrics on all three datasets.

Table R17 Performance on Davis in random task split (Few-shot)

Methods	MSE(↓)	CI(↑)	R2(↑)	Spearman(↑)	Pearson(↑)
ZeroBind	0.7971 (0.0422)	0.6603 (0.0388)	0.1059 (0.0491)	0.3135 (0.0566)	0.3335 (0.0479)
AdaMBind(ours)	0.4744 (0.0216)	0.7507 (0.0450)	0.2732 (0.0314)	0.4904 (0.0205)	0.5499 (0.0187)

The best results are highlighted in bold. Five independent replications of each method were performed (n = 5). Data are expressed as means (std).

Table R18 Performance on KIBA in random task split (Few-shot)

Methods	MSE(↓)	CI(↑)	R2(↑)	Spearman(↑)	Pearson(↑)
ZeroBind	0.7054 (0.1195)	0.6127 (0.0614)	0.0068 (0.0747)	0.2849 (0.0538)	0.3118 (0.0520)
AdaMBind(ours)	0.5715 (0.0806)	0.6802 (0.0185)	0.2379 (0.0769)	0.4847 (0.0504)	0.5062 (0.0653)

The best results are highlighted in bold. Five independent replications of each method were performed (n = 5). Data are expressed as means (std).

Table R19 Performance on BindingDB in random task split (Few-shot)

Methods	MSE(↓)	CI(↑)	R2(↑)	Spearman(↑)	Pearson(↑)
ZeroBind	1.3503 (0.1684)	0.7007 (0.0516)	0.2177 (0.0455)	0.4623 (0.0874)	0.5009 (0.0812)
AdaMBind(ours)	1.2116 (0.1255)	0.7260 (0.0292)	0.3094 (0.0629)	0.5323 (0.0758)	0.5634 (0.0590)

The best results are highlighted in bold. Five independent replications of each method were performed (n = 5). Data are expressed as means (std).

On the Davis dataset, ZeroBind reported MSE (0.7971), CI (0.6603), R^2 (0.1059), Spearman (0.3135), and Pearson (0.3335). AdaMBind achieved MSE (0.4744), CI (0.7507), R^2 (0.2732), Spearman (0.4904), and Pearson (0.5499), corresponding to improvements of 40.5% in MSE, 13.7% in CI, 56.4% in Spearman, and 64.9% in Pearson correlation.

On the KIBA dataset under few-shot conditions, ZeroBind achieved MSE (0.7054), CI (0.6127), R^2 (0.0068), Spearman (0.2849), and Pearson (0.3118), while AdaMBind attained MSE (0.5715), CI (0.6802), R^2 (0.2379), Spearman (0.4847), and Pearson (0.5062), representing improvements of 19.0% in MSE, 11.0% in CI, along with a substantial improvement in R^2 from near-zero to a positive value, as well as enhancements of 70.1% in Spearman and 62.4% in Pearson correlation.

On the BindingDB dataset under the few-shot setting, ZeroBind achieved MSE (1.3503), CI (0.7007), R^2 (0.2177), Spearman (0.4623), and Pearson (0.5009). AdaMBind improved these metrics to MSE (1.2116), CI (0.7260), R^2 (0.3094), Spearman (0.5323), and Pearson (0.5634), demonstrating relative increases of 10.3% in MSE, 3.6% in CI, 42.1% in R^2 , 15.1% in Spearman, and 12.4% in Pearson correlation.

b. Performance comparison of ZeroBind and AdaMBind under Novel Task Split

Under the majority setting (Table R20-Table R22), similar to random partitioning, AdaMBind outperformed ZeroBind across all metrics on all three datasets.

Table R20 Performance on Davis in novel task split (Majority)

Methods	MSE(↓)	CI(↑)	R2(↑)	Spearman(↑)	Pearson(↑)
ZeroBind	0.6506 (0.1684)	0.7582 (0.0265)	0.2882 (0.0480)	0.5248 (0.0479)	0.5713 (0.0442)
AdaMBind(ours)	0.5080 (0.1823)	0.8066 (0.0271)	0.4460 (0.0808)	0.5978 (0.0633)	0.6702 (0.0623)

The best results are highlighted in bold. Five independent replications of each method were performed ($n = 5$). Data are expressed as means (std).

Table R21 Performance on KIBA in novel task split (Majority)

Methods	MSE(↓)	CI(↑)	R2(↑)	Spearman(↑)	Pearson(↑)
ZeroBind	0.5398 (0.0667)	0.6461 (0.0348)	0.1567 (0.0482)	0.3928 (0.0973)	0.4647 (0.0822)
AdaMBind(ours)	0.3626 (0.0368)	0.7495 (0.0307)	0.3738 (0.1017)	0.6279 (0.0739)	0.6239 (0.0701)

The best results are highlighted in bold. Five independent replications of each method were performed (n = 5). Data are expressed as means (std).

Table R22 Performance on BindingDB in novel task split (Majority)

Methods	MSE(↓)	CI(↑)	R2(↑)	Spearman(↑)	Pearson(↑)
ZeroBind	1.2703 (0.2417)	0.7274 (0.0358)	0.3349 (0.1257)	0.5391 (0.1106)	0.6001 (0.0978)
AdaMBind(ours)	0.8758 (0.0374)	0.7689 (0.0348)	0.4114 (0.1199)	0.6064 (0.1205)	0.6407 (0.0869)

The best results are highlighted in bold. Five independent replications of each method were performed (n = 5). Data are expressed as means (std).

On the Davis dataset, ZeroBind achieved the following performance: MSE (0.6506), CI (0.7582), R² (0.2882), Spearman correlation (0.5248), and Pearson correlation (0.5713). In comparison, AdaMBind attained MSE (0.5080), CI (0.8066), R² (0.4460), Spearman (0.5978), and Pearson (0.6702), which corresponds to relative improvements of 21.9% in MSE, 6.4% in CI, 54.8% in R², 13.9% in Spearman, and 17.3% in Pearson correlation.

On the KIBA dataset, ZeroBind achieved MSE (0.5398), CI (0.6461), R² (0.1567), Spearman (0.3928), and Pearson (0.4647), whereas AdaMBind achieved MSE (0.3626), CI (0.7495), R² (0.3738), Spearman (0.6279), and Pearson (0.6239), representing improvements of 32.8% in MSE, 16.0% in CI, 138.5% in R², 59.9% in Spearman, and 34.3% in Pearson correlation.

On the BindingDB dataset under the majority setting, ZeroBind obtained MSE (1.2703), CI (0.7274), R² (0.3349), Spearman (0.5391), and Pearson (0.6001). AdaMBind improved these results to MSE (0.8758), CI (0.7689), R² (0.4114), Spearman (0.6064), and Pearson (0.6407), representing improvements of 31.1% in MSE, 5.7% in CI, 22.8% in R², 12.5% in Spearman, and 6.8% in Pearson correlation.

In the few-shot setting (Table R23-Table R26), AdaMBind still outperformed

ZeroBind across all metrics on all three datasets.

Table R23 Performance on Davis in novel task split (Few-shot)

Methods	MSE(↓)	CI(↑)	R2(↑)	Spearman(↑)	Pearson(↑)
ZeroBind	0.7723 (0.1552)	0.6285 (0.0277)	0.0935 (0.0746)	0.3260 (0.0654)	0.3135 (0.0789)
AdaMBind(ours)	0.6128 (0.2205)	0.7591 (0.0186)	0.2526 (0.1101)	0.4926 (0.0971)	0.5350 (0.0900)

The best results are highlighted in bold. Five independent replications of each method were performed (n = 5). Data are expressed as means (std).

Table R24 Performance on KIBA in novel task split (Few-shot)

Methods	MSE(↓)	CI(↑)	R2(↑)	Spearman(↑)	Pearson(↑)
ZeroBind	0.7316 (0.1137)	0.5549 (0.0256)	0.0011 (0.1320)	0.2324 (0.0685)	0.2681 (0.0714)
AdaMBind(ours)	0.4910 (0.1403)	0.6808 (0.0442)	0.2983 (0.1101)	0.4717 (0.0971)	0.5543 (0.0900)

The best results are highlighted in bold. Five independent replications of each method were performed (n = 5). Data are expressed as means (std).

Table R25 Performance on BindingDB in novel task split (Few-shot)

Methods	MSE(↓)	CI(↑)	R2(↑)	Spearman(↑)	Pearson(↑)
ZeroBind	1.2610 (0.3165)	0.6817 (0.0361)	0.1443 (0.0587)	0.4519 (0.0787)	0.4656 (0.0803)
AdaMBind(ours)	1.2968 (0.2389)	0.6980 (0.0169)	0.2536 (0.0352)	0.4703 (0.0505)	0.5171 (0.0421)

The best results are highlighted in bold. Five independent replications of each method were performed (n = 5). Data are expressed as means (std).

On the Davis dataset, ZeroBind achieved MSE (0.7723), CI (0.6285), R² (0.0935), Spearman (0.3260), and Pearson (0.3135). AdaMBind achieved MSE (0.6128), CI (0.7591), R² (0.2526), Spearman (0.4926), and Pearson (0.5350), corresponding to improvements of 20.7% in MSE, 20.8% in CI, 170.2% in R², 51.1% in Spearman, and 70.7% in Pearson correlation.

On the KIBA dataset under few-shot conditions, ZeroBind achieved MSE (0.7316), CI (0.5549), R² (0.0011), Spearman (0.2324), and Pearson (0.2681), while AdaMBind

attained MSE (0.4910), CI (0.6808), R^2 (0.2983), Spearman (0.4717), and Pearson (0.5543), representing substantial gains of 32.9% in MSE, 22.7% in CI, along with a remarkable improvement in R^2 from near-zero to a positive value, as well as enhancements in Spearman and Pearson correlation.

On the BindingDB dataset under the few-shot setting, ZeroBind achieved MSE (1.2610), CI (0.6817), R^2 (0.1443), Spearman (0.4519), and Pearson (0.4656). AdaMBind enhanced these metrics to MSE (1.2968), CI (0.6980), R^2 (0.2536), Spearman (0.4703), and Pearson (0.5171), demonstrating relative increases of -2.8% in MSE, 2.4% in CI, 75.8% in R^2 , 4.1% in Spearman, and 11.1% in Pearson correlation.

In summary, under both random task splitting and the more challenging novel task splitting strategies, and regardless of whether in the majority or few-shot settings, **AdaMBind consistently and significantly outperforms ZeroBind across all evaluation metrics on the three benchmark datasets** (Davis, KIBA, and BindingDB). Compared to ZeroBind, AdaMBind demonstrates superior prediction accuracy, stronger generalization capability, and greater robustness to task distribution shifts.

3. I recommend that the authors consider including the PSICHIC as an additional non-meta-learning baseline. PSICHIC is specifically designed to integrate physicochemical principles when decoding protein–ligand interaction fingerprints directly from sequence data. Unlike prior sequence-based models, it uniquely incorporates these physicochemical constraints, achieving state-of-the-art accuracy and enhanced interpretability. Given that PSICHIC represents a highly advanced sequence-based DTA prediction model, its inclusion would provide a more comprehensive and rigorous comparative evaluation of the proposed approach.

Response:

Thank you for this valuable suggestion. We have incorporated PSICHIC as an additional non-meta-learning baseline in our comparative experiments and subsequently analyzed the experimental results.

(1) Incorporation of PSICHIC as Baseline

It should be noted that the original PSICHIC model is a general-purpose protein-

ligand interaction predictor trained on the PDBBind v2016 and v2020 datasets. To ensure a fair comparison, we retrained the PSICHIC model using the same training/test splits as employed in our study. The performance results on the three datasets under the random task split are presented in Table R38-Table R43 (corresponding to Fig.2 in the revised manuscript). Similarly, the results under the novel task split are also provided in Table R53-Table R58 (corresponding to Fig.3 in the revised manuscript).

(2) Comparative evaluation of AdaMBind and PSICHIC

Similar to comment 1, we adopted two task-partitioning strategies to systematically evaluate the performance of AdaMBind and PSICHIC under different levels of task novelty. Including the **random task split strategy and novel task split strategy**.

a. Performance comparison of PSICHIC and AdaMBind under random task split

Under the majority setting (Table R26-Table R28), besides the MSE on the Davis dataset, AdaMBind outperformed PSICHIC on all other metrics across all three datasets.

Table R26 Performance on Davis in random task split (Majority)

Methods	MSE(↓)	CI(↑)	R2(↑)	Spearman(↑)	Pearson(↑)
PSICHIC	0.4299 (0.1584)	0.8142 (0.0146)	0.4646 (0.0671)	0.6037 (0.0517)	0.6624 (0.0465)
AdaMBind(ours)	0.5060 (0.1672)	0.8222 (0.0160)	0.4693 (0.0538)	0.6257 (0.0389)	0.6878 (0.0385)

The best results are highlighted in bold. Five independent replications of each method were performed (n = 5). Data are expressed as means (std).

Table R27 Performance on KIBA in random task split (Majority)

Methods	MSE(↓)	CI(↑)	R2(↑)	Spearman(↑)	Pearson(↑)
PSICHIC	0.5592 (0.0503)	0.7077 (0.0344)	0.2773 (0.0348)	0.5409 (0.0596)	0.5404 (0.0373)
AdaMBind(ours)	0.4295 (0.0566)	0.7545 (0.0245)	0.4285 (0.0392)	0.6528 (0.0545)	0.6559 (0.0297)

The best results are highlighted in bold. Five independent replications of each method were performed (n = 5). Data are expressed as means (std).

Table R28 Performance on BindingDB in random task split (Majority)

Methods	MSE(↓)	CI(↑)	R2(↑)	Spearman(↑)	Pearson(↑)
PSICHIC	1.0048 (0.1154)	0.7758 (0.0301)	0.3119 (0.0781)	0.5732 (0.0846)	0.5821 (0.0653)
AdaMBind(ours)	0.8578 (0.1077)	0.7843 (0.0221)	0.4815 (0.0657)	0.6578 (0.0413)	0.6966 (0.0430)

The best results are highlighted in bold. Five independent replications of each method were performed (n = 5). Data are expressed as means (std).

On the Davis dataset, PSICHIC achieved the following performance: MSE (0.4299), CI (0.8142), R² (0.4646), Spearman correlation (0.6037), and Pearson correlation (0.6624). In comparison, AdaMBind attained MSE (0.5060), CI (0.8222), R² (0.4693), Spearman (0.6257), and Pearson (0.6878), which corresponds to a relative increase of 17.7% in MSE, improvements of 1.0% in CI, 1.0% in R², 3.6% in Spearman, and 3.8% in Pearson correlation.

On the KIBA dataset under the same majority setting, PSICHIC achieved MSE (0.5592), CI (0.7077), R² (0.2773), Spearman (0.5409), and Pearson (0.5404), whereas AdaMBind achieved MSE (0.4295), CI (0.7545), R² (0.4285), Spearman (0.6528), and Pearson (0.6559), representing gains of 23.2% in MSE, 6.6% in CI, 54.5% in R², 20.7% in Spearman, and 21.4% in Pearson correlation.

On the BindingDB dataset under the majority setting, PSICHIC obtained MSE (1.0048), CI (0.7758), R² (0.3119), Spearman (0.5732), and Pearson (0.5821). AdaMBind improved these results to MSE (0.8578), CI (0.7843), R² (0.4815), Spearman (0.6578), and Pearson (0.6966), reflecting enhancements of 14.6% in MSE, 1.1% in CI, 54.4% in R², 14.8% in Spearman, and 19.7% in Pearson correlation.

Under the few-shot setting (Table R29-Table R31), **AdaMBind outperformed PSICHIC across all metrics on all three datasets.**

Table R29 Performance on Davis in random task split (Few-shot)

Methods	MSE(↓)	CI(↑)	R2(↑)	Spearman(↑)	Pearson(↑)
PSICHIC	0.6539 (0.0431)	0.7232 (0.0323)	0.1634 (0.0587)	0.4166 (0.0441)	0.4525 (0.0254)
AdaMBind(ours)	0.4744 (0.0216)	0.7507 (0.0450)	0.2732 (0.0314)	0.4904 (0.0205)	0.5499 (0.0187)

The best results are highlighted in bold. Five independent replications of each method were performed (n = 5). Data are expressed as means (std).

Table R30 Performance on KIBA in random task split (Few-shot)

Methods	MSE(↓)	CI(↑)	R2(↑)	Spearman(↑)	Pearson(↑)
PSICHIC	0.8755 (0.1245)	0.6169 (0.0317)	0.0732 (0.0884)	0.3276 (0.0579)	0.2899 (0.0534)
AdaMBind(ours)	0.5715 (0.0806)	0.6802 (0.0185)	0.2379 (0.0769)	0.4847 (0.0504)	0.5062 (0.0653)

The best results are highlighted in bold. Five independent replications of each method were performed (n = 5). Data are expressed as means (std).

Table R31 Performance on BindingDB in random task split (Few-shot)

Methods	MSE(↓)	CI(↑)	R2(↑)	Spearman(↑)	Pearson(↑)
PSICHIC	1.7273 (0.2318)	0.6880 (0.0467)	0.1696 (0.0231)	0.4612 (0.0417)	0.4378 (0.0570)
AdaMBind(ours)	1.2116 (0.1255)	0.7260 (0.0292)	0.3094 (0.0629)	0.5323 (0.0758)	0.5634 (0.0590)

The best results are highlighted in bold. Five independent replications of each method were performed (n = 5). Data are expressed as means (std).

On the Davis dataset, PSICHIC achieved MSE (0.6539), CI (0.7232), R² (0.1634), Spearman (0.4166), and Pearson (0.4525). AdaMBind attained MSE (0.4744), CI (0.7507), R² (0.2732), Spearman (0.4904), and Pearson (0.5499), corresponding to improvements of 27.5% in MSE, 3.8% in CI, 67.2% in R², 17.7% in Spearman, and 21.5% in Pearson correlation.

On the KIBA dataset, PSICHIC achieved MSE (0.8755), CI (0.6169), R² (0.0732), Spearman (0.3276), and Pearson (0.2899), while AdaMBind attained MSE (0.5715), CI (0.6802), R² (0.2379), Spearman (0.4847), and Pearson (0.5062), representing gains of 34.7% in MSE, 10.3% in CI, 48.0% in Spearman, and 74.6% in Pearson correlation.

On the BindingDB dataset, PSICHIC achieved MSE (1.7273), CI (0.6880), R² (0.1696), Spearman (0.4612), and Pearson (0.4378). AdaMBind enhanced these metrics to MSE (1.2116), CI (0.7260), R² (0.3094), Spearman (0.5323), and Pearson (0.5634), demonstrating relative increases of 29.9% in MSE, 5.5% in CI, 82.4% in R², 15.4% in

Spearman, and 28.7% in Pearson correlation.

b. Performance comparison of AdaMBind and PSICHIC under novel task split

Under the majority setting (Table R32-Table R34), AdaMBind outperformed PSICHIC across all metrics on all three datasets.

Table R32 Performance on Davis in novel task split (Majority)

Methods	MSE(↓)	CI(↑)	R2(↑)	Spearman(↑)	Pearson(↑)
PSICHIC	0.8463 (0.2298)	0.7739 (0.0487)	0.2915 (0.0681)	0.5668 (0.0513)	0.6051 (0.0544)
AdaMBind(ours)	0.5080 (0.1823)	0.8066 (0.0271)	0.4460 (0.0808)	0.5978 (0.0633)	0.6702 (0.0623)

The best results are highlighted in bold. Five independent replications of each method were performed (n = 5). Data are expressed as means (std).

Table R33 Performance on KIBA in novel task split (Majority)

Methods	MSE(↓)	CI(↑)	R2(↑)	Spearman(↑)	Pearson(↑)
PSICHIC	0.4680 (0.0465)	0.6411 (0.0369)	0.1113 (0.0549)	0.3702 (0.0842)	0.3602 (0.0761)
AdaMBind(ours)	0.3626 (0.0368)	0.7495 (0.0307)	0.3738 (0.1017)	0.6279 (0.0739)	0.6239 (0.0701)

The best results are highlighted in bold. Five independent replications of each method were performed (n = 5). Data are expressed as means (std).

Table R34 Performance on BindingDB in novel task split (Majority)

Methods	MSE(↓)	CI(↑)	R2(↑)	Spearman(↑)	Pearson(↑)
PSICHIC	1.634 (0.0483)	0.7146 (0.0357)	0.2185 (0.0987)	0.5328 (0.0816)	0.5282 (0.0709)
AdaMBind(ours)	0.8758 (0.0374)	0.7689 (0.0348)	0.4114 (0.1199)	0.6064 (0.1205)	0.6407 (0.0869)

The best results are highlighted in bold. Five independent replications of each method were performed (n = 5). Data are expressed as means (std).

On the Davis dataset, PSICHIC achieved the following performance: MSE (0.8463), CI (0.7739), R² (0.2915), Spearman correlation (0.5668), and Pearson correlation (0.6051). In comparison, AdaMBind attained MSE (0.5080), CI (0.8066), R² (0.4460), Spearman (0.5978), and Pearson (0.6702), which corresponds to relative improvements of 40.0% in MSE, 4.2% in CI, 53.0% in R², 5.5% in Spearman, and 10.8%

in Pearson correlation.

On the KIBA dataset under the same majority setting, PSICHIC achieved MSE (0.4680), CI (0.6411), R² (0.1113), Spearman (0.3702), and Pearson (0.3602), whereas AdaMBind achieved MSE (0.3626), CI (0.7495), R² (0.3738), Spearman (0.6279), and Pearson (0.6239), representing gains of 22.5% in MSE, 16.9% in CI, 69.6% in Spearman, and 73.2% in Pearson correlation.

On the BindingDB dataset under the majority setting, PSICHIC obtained MSE (1.6340), CI (0.7146), R² (0.2185), Spearman (0.5328), and Pearson (0.5282). AdaMBind improved these results to MSE (0.8758), CI (0.7689), R² (0.4114), Spearman (0.6064), and Pearson (0.6407), reflecting enhancements of 46.4% in MSE, 7.6% in CI, 88.3% in R², 13.8% in Spearman, and 21.3% in Pearson correlation.

In the few-shot setting (Table R35-Table R37), AdaMBind also outperformed PSICHIC across all metrics on all three datasets.

Table R35 Performance on Davis in novel task split (Few-shot)

Methods	MSE(↓)	CI(↑)	R2(↑)	Spearman(↑)	Pearson(↑)
PSICHIC	0.7207 (0.1697)	0.7023 (0.0418)	0.2295 (0.0899)	0.4086 (0.0786)	0.4535 (0.0741)
AdaMBind(ours)	0.6128 (0.2205)	0.7591 (0.0186)	0.2526 (0.1101)	0.4926 (0.0971)	0.5350 (0.0900)

The best results are highlighted in bold. Five independent replications of each method were performed (n = 5). Data are expressed as means (std).

Table R36 Performance on KIBA in novel task split (Few-shot)

Methods	MSE(↓)	CI(↑)	R2(↑)	Spearman(↑)	Pearson(↑)
PSICHIC	0.9733 (0.1550)	0.5375 (0.0396)	-0.1230 (0.1546)	0.1078 (0.0897)	0.1158 (0.0757)
AdaMBind(ours)	0.4910 (0.1403)	0.6808 (0.0442)	0.2983 (0.1101)	0.4717 (0.0971)	0.5543 (0.0900)

The best results are highlighted in bold. Five independent replications of each method were performed (n = 5). Data are expressed as means (std).

Table R37 Performance on BindingDB in novel task split (Few-shot)

Methods	MSE(↓)	CI(↑)	R2(↑)	Spearman(↑)	Pearson(↑)
PSICHIC	1.8581 (0.2548)	0.6341 (0.0250)	0.1207 (0.0878)	0.3100 (0.0891)	0.3662 (0.0752)

Methods	MSE(↓)	CI(↑)	R2(↑)	Spearman(↑)	Pearson(↑)
AdaMBind(ours)	1.2968 (0.2389)	0.6980 (0.0169)	0.2536 (0.0352)	0.4703 (0.0505)	0.5171 (0.0421)

The best results are highlighted in bold. Five independent replications of each method were performed (n = 5). Data are expressed as means (std).

On the Davis dataset, PSICHIC achieved MSE (0.7207), CI (0.7023), R² (0.2295), Spearman (0.4086), and Pearson (0.4535). AdaMBind attained MSE (0.6128), CI (0.7591), R² (0.2526), Spearman (0.4926), and Pearson (0.5350), corresponding to improvements of 15.0% in MSE, 8.1% in CI, 10.1% in R², 20.6% in Spearman, and 18.0% in Pearson correlation.

On the KIBA dataset, PSICHIC achieved MSE (0.9733), CI (0.5375), R² (-0.1230), Spearman (0.1078), and Pearson (0.1158), while AdaMBind attained MSE (0.4910), CI (0.6808), R² (0.2983), Spearman (0.4717), and Pearson (0.5543), representing gains of 49.6% in MSE, 26.7% in CI, along with a substantial turnaround from negative to positive R², as well as improvements of Spearman and Pearson correlation.

On the BindingDB dataset, PSICHIC achieved MSE (1.8581), CI (0.6341), R² (0.1207), Spearman (0.3100), and Pearson (0.3662). AdaMBind enhanced these metrics to MSE (1.2968), CI (0.6980), R² (0.2536), Spearman (0.4703), and Pearson (0.5171), demonstrating relative increases of 30.2% in MSE, 10.1% in CI, 51.7% in Spearman, and 41.2% in Pearson correlation.

In the revised manuscript, we have incorporated the results of the aforementioned three models (Comments 1-3) into the model comparison. (Table R38-Table R43 and Table R53-Table R58)

a. Performance comparison of Baselines and AdaMBind under random task split.

As shown in Table R38-Table R40 (corresponding to **Figure R1a,c**), AdaMBind demonstrated outstanding performance under the **majority setting** (with a support set size of 40). Specifically, on the KIBA dataset, AdaMBind achieved a significant improvement over the best baseline, ColdDTA. The MSE was reduced by 11.37% (from 0.4868 to 0.4295), the CI (Consistency Index) was improved by 6.04% (from 0.7115 to

0.7545), and the R^2 reached 0.4285, while the Spearman and Pearson correlation coefficients (PCC) increased by 17.81% (from 0.5541 to 0.6528) and 10.05% (from 0.5960 to 0.6559), respectively. This performance surpassed all other comparative methods, including HiSIF-DTA, which relies on protein structural feature extraction. On the BindingDB dataset, AdaMBind achieved the best results across all evaluation metrics. Compared to the best baseline MetaDTA, AdaMBind reduced MSE by 0.67% (from 0.8636 to 0.8578), improved CI by 0.86% (from 0.7776 to 0.7843), increased R^2 by 15.03% (from 0.4186 to 0.4815), enhanced the Spearman and Pearson correlation coefficient by 9.79% (from 0.5991 to 0.6578) and 6.67% (from 0.6530 to 0.6966). On the Davis dataset, AdaMBind demonstrated the best performance in CI and Spearman correlation metrics, achieving improvements of 0.34% (from 0.8194 to 0.8222) and 2.27% (from 0.6118 to 0.6257) over the best baseline HiSIF, respectively. Its MSE was 0.5060, ranking fourth, while its R^2 value of 0.4693 was second only to HiSIF (0.5096). The Pearson correlation coefficient exhibited a slight decrease of 0.97% (from 0.6946 to 0.6878).

As shown in Table R41-Table R43 (corresponding to **Figure R1b,c**), under the few-shot setting, AdaMBind also demonstrates outstanding performance. On the KIBA dataset, compared to the best baseline Co-VAE, AdaMBind reduced MSE by 10.47% (from 0.6383 to 0.5715), improved CI by 7.51% (from 0.6327 to 0.6802), increased R^2 from 0.1210 to 0.2379, and enhanced Spearman and Pearson correlation coefficients by 35.50% (from 0.3577 to 0.4847) and 22.65% (from 0.4127 to 0.5062), respectively. On the BindingDB dataset, compared to the best baseline ColdDTA, AdaMBind reduced MSE by 10.07% (from 1.3474 to 1.2116), while achieving superior R^2 (from 0.2693 to 0.3094). The Spearman and Pearson improved by 6.22% (from 0.5011 to 0.5323) and 6.66% (from 0.5282 to 0.5634). On the Davis dataset, compared to the best baseline MetaDTA, AdaMBind reduced MSE by 11.54% (from 0.5363 to 0.4744), improved R^2 from 0.2495 to 0.2732, and enhanced Pearson correlation coefficient by 4.16% (from 0.5279 to 0.5499), while showing slight decrease in CI performance by -2.42% (from 0.7693 to 0.7507) and slight decrease in Spearman correlation coefficient by -2.54% (from 0.5032 to 0.4904).

Figure R1 Performance comparison between AdaMBind and baseline models under random task split. a, b Performance evaluation of AdaMBind and baseline models under the majority setting (a) and few-shot setting (b). **c** Heatmap showing performance gains or losses of AdaMBind compared to its strongest competitor under both majority and few-shot settings.

b. Performance comparison of Baselines and AdaMBind under novel task split based on sequence similarity.

As shown in Table R53-Table R55 (corresponding to Figure R2a,c), under the majority setting on the KIBA dataset, compared to the best baseline ColdDTA, AdaMBind reduced MSE by 22.52% (from 0.4680 to 0.3626), improved CI by 4.97% (from 0.7140 to 0.7495), increased R^2 from 0.2821 to 0.3738, and enhanced Spearman and Pearson correlation coefficients by 20.20% (from 0.5224 to 0.6279) and 16.57% (from 0.5352 to 0.6239), respectively. On the BindingDB dataset, compared to the best baseline MetaDTA, AdaMBind reduced MSE by 1.62% (from 0.8902 to 0.8758), improved R^2 by from 0.3837 to 0.4114, and increased the Pearson correlation coefficient by 1.49% (from 0.6313 to 0.6407), while showing slight decreases in CI and Spearman correlation by 0.27% (from 0.7710 to 0.7689) and 0.36% (from 0.6086 to 0.6064), respectively. On the Davis dataset, compared to the best baseline Co-VAE, AdaMBind improved R^2 by 7.00% (from 0.4168 to 0.4460) and Pearson correlation from 0.6517 to 0.6702, while MSE, CI, and Spearman correlation decreased by 2.42% (from 0.4960 to 0.5080), 1.48% (from 0.8187 to 0.8066), and 9.51% (from 0.6606 to 0.5978), respectively.

As shown in Table R56-Table R58 (corresponding to Figure R2b,c), the advantages of AdaMBind became more pronounced. On the KIBA dataset, compared to the best baseline MetaDTA, AdaMBind reduced MSE by 28.71% (from 0.6887 to 0.4910), improves CI by 5.53% (from 0.6451 to 0.6808), increased R^2 from 0.1055 to 0.2983, and enhanced Spearman and Pearson correlation coefficients by 22.07% (from 0.3864 to 0.4717) and 27.69% (from 0.4341 to 0.5543), respectively. On the BindingDB dataset, compared to the best baseline MetaDTA, AdaMBind improved R^2 from 0.2184 to 0.2536. The Spearman and Pearson correlations improved by 3.84% (from 0.4529 to 0.4703) and 3.17% (from 0.5012 to 0.5171), respectively, while MSE and CI decreased by 2.84% (from 1.2610 to 1.2968) and 3.59% (from 0.7240 to 0.6980). On the Davis dataset, compared to the best baseline MetaDTA, AdaMBind improved R^2 from 0.2347 to 0.2526. The Spearman and Pearson correlations improved by 3.73% (from 0.4749 to 0.4926) and 2.41% (from 0.5224 to 0.5350), respectively, while MSE and CI decreased

by 0.97% (from 0.6069 to 0.6128) and 2.48% (from 0.7784 to 0.7591).

Figure R2 Performance comparison between AdaMBind and baseline models under random task split. a, b Performance evaluation of AdaMBind and baseline models under the

majority setting (a) and few-shot setting (b). **c** Heatmap showing performance gains or losses of AdaMBind compared to its strongest competitor under both majority and few-shot settings.

In summary, under the random task split, AdaMBind demonstrates exceptional and stable predictive performance in both the majority and few-shot settings, significantly outperforming existing baseline methods.

Table R38 Performance on Davis in random task split (Majority)

Methods	MSE(↓)	CI(↑)	R2(↑)	Spearman(↑)	Pearson(↑)
DeepDTA	0.5487 (0.1840)	0.7979 (0.0081)	0.4096 (0.0514)	0.5814 (0.0467)	0.6567 (0.0378)
ColdDTA	0.5248 (0.1710)	0.8099 (0.0151)	0.4348 (0.0476)	0.6093 (0.0553)	0.6637 (0.0381)
Co-VAE	0.4217 (0.0860)	0.7857 (0.0241)	0.4137 (0.0729)	0.5329 (0.0440)	0.6524 (0.0554)
HiSIF	0.4097 (0.2165)	0.8194 (0.0212)	0.5096 (0.0315)	0.6118 (0.0660)	0.6946 (0.0449)
MetaDTA	0.3494 (0.1839)	0.8084 (0.0154)	0.3811 (0.0456)	0.5376 (0.0712)	0.6205 (0.0335)
CML	0.5295 (0.2405)	0.7850 (0.1238)	0.3517 (0.0307)	0.5788 (0.0393)	0.5892 (0.0351)
PSICHIC	0.4299 (0.1584)	0.8142 (0.0146)	0.4646 (0.0671)	0.6037 (0.0517)	0.6624 (0.0465)
ZeroBind	0.5923 (0.1865)	0.7637 (0.0452)	0.3448 (0.0584)	0.5179 (0.0454)	0.5489 (0.0513)
AdaMBind(ours)	0.5060 (0.1672)	0.8222 (0.0160)	0.4693 (0.0538)	0.6257 (0.0389)	0.6878 (0.0385)

The best results are highlighted in bold, while the second-best results are underlined. Five independent replications of each method were performed (n = 5). Data are expressed as means (std).

Table R39 Performance on KIBA in random task split (Majority)

Methods	MSE(↓)	CI(↑)	R2(↑)	Spearman(↑)	Pearson(↑)
DeepDTA	0.5427 (0.0472)	0.6806 (0.0094)	0.2776 (0.0265)	0.4779 (0.0231)	0.5386 (0.0199)
ColdDTA	0.4868 (0.0335)	0.7115 (0.0094)	0.3512 (0.0164)	0.5541 (0.0211)	0.5960 (0.0105)
Co-VAE	0.5706 (0.0665)	0.6753 (0.0191)	0.2342 (0.0290)	0.4610 (0.0472)	0.5242 (0.0203)
HiSIF	0.5268 (0.0561)	0.6785 (0.0251)	0.2849 (0.0518)	0.4749 (0.0619)	0.5434 (0.0473)
MetaDTA	0.5648 (0.0943)	0.6747 (0.0106)	0.2308 (0.0257)	0.3368 (0.0512)	0.3637 (0.0345)
CML	0.6734 (0.0805)	0.6549 (0.0330)	0.0879 (0.0655)	0.4196 (0.0813)	0.3597 (0.0550)
PSICHIC	0.5592 (0.0503)	0.7077 (0.0344)	0.2773 (0.0348)	0.5409 (0.0596)	0.5404 (0.0373)
ZeroBind	0.5727 (0.0706)	0.6755 (0.0287)	0.2484 (0.0299)	0.4562 (0.0495)	0.4810 (0.0277)
AdaMBind(ours)	0.4295 (0.0566)	0.7545 (0.0245)	0.4285 (0.0392)	0.6528 (0.0545)	0.6559 (0.0297)

The best results are highlighted in bold, while the second-best results are underlined. Five independent replications of each method were performed (n = 5). Data are expressed as means (std).

Table R40 Performance on BindingDB in random task split (Majority)

Methods	MSE(↓)	CI(↑)	R2(↑)	Spearman(↑)	Pearson(↑)
DeepDTA	1.1268 (0.4365)	0.7513 (0.0401)	0.3443 (0.1693)	0.5486 (0.0848)	0.5594 (0.0975)
ColdDTA	1.0833 (0.1407)	0.7540 (0.0147)	0.3513 (0.0442)	0.5919 (0.0220)	0.5956 (0.0370)
Co-VAE	1.2379 (0.2714)	0.7412 (0.0641)	0.3231 (0.1357)	0.4257 (0.1282)	0.4713 (0.1257)
HiSIF	1.1076 (0.2401)	0.7363 (0.0322)	0.3317 (0.0650)	0.5817 (0.0467)	0.6387 (0.0594)
MetaDTA	0.8636 (0.1417)	0.7776 (0.0564)	0.4186 (0.0506)	0.5991 (0.0340)	0.6530 (0.0352)
CML	1.0017 (0.0814)	0.7595 (0.0194)	0.3946 (0.0506)	0.5735 (0.0340)	0.6392 (0.0352)
PSICHIC	1.0048 (0.1154)	0.7758 (0.0301)	0.3119 (0.0781)	0.5732 (0.0846)	0.5821 (0.0653)
ZeroBind	1.2613 (0.1306)	0.7379 (0.0316)	0.3491 (0.0557)	0.5675 (0.0463)	0.6176 (0.0531)
AdaMBind(ours)	0.8578 (0.1077)	0.7843 (0.0221)	0.4815 (0.0657)	0.6578 (0.0413)	0.6966 (0.0430)

The best results are highlighted in bold, while the second-best results are underlined. Five independent replications of each method were performed (n = 5). Data are expressed as means (std).

Table R41 Performance on Davis in random task split (Few-shot)

Methods	MSE(↓)	CI(↑)	R2(↑)	Spearman(↑)	Pearson(↑)
DeepDTA	0.6363 (0.0249)	0.7693 (0.0098)	0.2062 (0.0079)	0.5032 (0.0232)	0.5279 (0.0214)
ColdDTA	0.7073 (0.0725)	0.6997 (0.0126)	0.1210 (0.0995)	0.3772 (0.0269)	0.4548 (0.0640)
Co-VAE	0.6856 (0.0377)	0.6896 (0.0301)	0.1013 (0.0659)	0.3561 (0.0572)	0.4075 (0.0573)
HiSIF	0.5877 (0.0301)	0.7613 (0.0164)	0.2495 (0.0346)	0.4908 (0.0363)	0.5232 (0.0395)
MetaDTA	0.5363 (0.0570)	0.7618 (0.0354)	0.2213 (0.0536)	0.4794 (0.0482)	0.5099 (0.0398)
CML	0.8138 (0.1723)	0.6474 (0.0204)	0.0689 (0.0589)	0.2874 (0.0510)	0.2728 (0.0708)
PSICHIC	0.6539 (0.0431)	0.7232 (0.0323)	0.1634 (0.0587)	0.4166 (0.0441)	0.4525 (0.0254)
ZeroBind	0.7971 (0.0422)	0.6603 (0.0388)	0.1059 (0.0491)	0.3135 (0.0566)	0.3335 (0.0479)
AdaMBind(ours)	0.4744 (0.0216)	0.7507 (0.0450)	0.2732 (0.0314)	0.4904 (0.0205)	0.5499 (0.0187)

The best results are highlighted in bold, while the second-best results are underlined. Five independent replications of each method were performed (n = 5). Data are expressed as means (std).

Table R42 Performance on KIBA in random task split (Few-shot)

Methods	MSE(↓)	CI(↑)	R2(↑)	Spearman(↑)	Pearson(↑)
DeepDTA	0.7534 (0.1138)	0.5860 (0.0232)	-0.0290 (0.1088)	0.2391 (0.0665)	0.2606 (0.0639)
ColdDTA	0.7002 (0.1598)	0.5887 (0.0752)	0.0376 (0.0353)	0.2418 (0.0957)	0.2437 (0.1023)
Co-VAE	0.7100 (0.1048)	0.6327 (0.0171)	0.0844 (0.0491)	0.3577 (0.0472)	0.4127 (0.0412)
HiSIF	0.7193 (0.1647)	0.5914 (0.0126)	0.0256 (0.2496)	0.2517 (0.0337)	0.2247 (0.0291)
MetaDTA	0.6383 (0.1515)	0.6150 (0.0644)	0.1210 (0.0721)	0.3097 (0.0482)	0.3563 (0.0565)
CML	0.7713 (0.1363)	0.6272 (0.0253)	-0.0463 (0.0864)	0.3457 (0.0669)	0.2636 (0.0686)
PSICHIC	0.8755 (0.1245)	0.6169 (0.0317)	0.0732 (0.0884)	0.3276 (0.0579)	0.2899 (0.0534)
ZeroBind	0.7054 (0.1195)	0.6127 (0.0614)	0.0068 (0.0747)	0.2849 (0.0538)	0.3118 (0.0520)
AdaMBind(ours)	0.5715 (0.0806)	0.6802 (0.0185)	0.2379 (0.0769)	0.4847 (0.0504)	0.5062 (0.0653)

The best results are highlighted in bold, while the second-best results are underlined. Five independent replications of each method were performed (n = 5). Data are expressed as means (std).

Table R43 Performance on BindingDB in random task split (Few-shot)

Methods	MSE	CI	R2	Spearman	Pearson
DeepDTA	1.6575 (0.1462)	0.7177 (0.0476)	0.2031 (0.0796)	0.4878 (0.1211)	0.5153 (0.1005)
ColdDTA	1.3474 (0.2696)	0.7042 (0.0342)	0.2390 (0.0652)	0.4858 (0.0800)	0.5022 (0.0566)
Co-VAE	1.3554 (0.2501)	0.6818 (0.0213)	0.1958 (0.0282)	0.4237 (0.0254)	0.4701 (0.0332)
HiSIF	1.4322 (0.1766)	0.7252 (0.0307)	0.2215 (0.0751)	0.4839 (0.0873)	0.5103 (0.0794)
MetaDTA	1.4342 (0.1611)	0.7534 (0.0172)	0.2693 (0.0523)	0.5011 (0.0423)	0.5282 (0.0398)
CML	1.3982 (0.1782)	0.6965 (0.0356)	0.2029 (0.0941)	0.4637 (0.0862)	0.5050 (0.0780)
PSICHIC	1.7273 (0.2318)	0.6880 (0.0467)	0.1696 (0.0231)	0.4612 (0.0417)	0.4378 (0.0570)
ZeroBind	1.3503 (0.1684)	0.7007 (0.0516)	0.2177 (0.0455)	0.4623 (0.0874)	0.5009 (0.0812)
AdaMBind(ours)	1.2116 (0.1255)	0.7260 (0.0292)	0.3094 (0.0629)	0.5323 (0.0758)	0.5634 (0.0590)

The best results are highlighted in bold, while the second-best results are underlined. Five independent replications of each method were performed (n = 5). Data are expressed as means (std).

4. It would be helpful for the authors to present the label distributions of the BindingDB, KIBA, and Davis datasets. As far as I am aware, the affinity distributions in KIBA and Davis are highly concentrated. The authors should provide a detailed discussion of this issue, as it may have significant implications for model training, evaluation, and generalization.

Response:

Thank you for raising this important point. We conducted a systematic analysis from two perspectives. First, we conducted a detailed analysis of the label distributions across all three datasets and examined how such distributions impact the model training and evaluation process. Second, to further assess the generalization implications, we performed cross-domain experiments that explicitly tested the model’s ability to generalize across different data distributions.

(1) Extended analysis of data distribution

We employed bar charts to illustrate the label distributions across the three datasets, as depicted in Figure R3. In addition, we conducted a systematic analysis of three datasets across three dimensions: **distribution concentration, distribution shape, and**

kurtosis.

First, we analyzed the distribution concentration based on the coefficient of variation (CV) and interquartile range (IQR). CV is defined as the ratio of the standard deviation (σ) to the absolute value of the mean ($|\mu|$): $CV = \sigma / |\mu| \times 100\%$. A lower CV indicates that the data points are closely clustered around the mean, reflecting high distributional homogeneity, whereas a higher CV suggests greater dispersion and variability. The IQR reflects the spread of the middle 50% of the data.

Analysis indicated variations in distribution concentration across the three datasets. The KIBA dataset (CV=7.14%) and Davis dataset (CV=16.41%) exhibited highly concentrated distributions, whereas the BindingDB dataset (CV=23.65%) demonstrates relatively moderate concentration. In addition, the KIBA dataset exhibited an interquartile range of merely 0.72, indicating that half the data points cluster within the narrow range [11.20, 11.92]. The Davis dataset demonstrated an even lower interquartile range (0.52), with data pointed highly concentrated within the interval [5.00, 5.52].

Second, we analyzed the distribution skewness. All three datasets exhibited right-skewed distributions (skewness > 0), with the Davis dataset showing the highest skewness (2.30), indicating a greater proportion of low-affinity values.

Third, we analyzed the kurtosis of three datasets. Both KIBA (4.51) and Davis (5.16) demonstrated leptokurtic distributions (kurtosis > 3), further confirming the concentrated nature of the data.

Based on the aforementioned analysis, we draw the following conclusions regarding the label distributions across the three datasets. The affinity value distributions in the KIBA and Davis datasets exhibited significant centralized characteristics, demonstrated by low CV and IQR values, right-skewed distributions, and high kurtosis. The BindingDB dataset exhibited a relatively broader distribution, though it still demonstrates moderate centralization.

This distributional characteristic poses significant challenges for model training. First, models are prone to overfitting to the main distribution intervals, which affects their predictive capability for out-of-distribution samples. Second, traditional

regression metrics such as MSE may produce misleading results. Finally, models may struggle to learn discriminative feature representations. We have added the above analyses to the manuscript. (The corresponding revision can be seen Lines 354-363 on Page 21)

In this context, the choice of evaluation metrics becomes particularly crucial. We adopted a multi-metric comprehensive evaluation strategy, combining regression error metrics (MSE), R^2 , with ranking correlation metrics (CI, Spearman, Pearson) to more comprehensively reflect the model’s performance under different distribution characteristics.

Figure R3 Affinity distribution across Davis, KIBA, and BindingDB datasets

(2) Performance comparison under cross-domain generalization evaluation

To address the potential issue of poor model generalization performance arising from data distribution centralization, we designed cross-domain transfer experiments based on majority settings to systematically validate the model’s cross-domain generalization capabilities.

Specifically, we followed the novel task split scheme described in the main text. The model was first meta-trained on the KIBA meta-training and meta-validation tasks for 10 outer-loop epochs. After meta-training, the model was adapted to each test task in the BindingDB meta-test set by fine-tuning on its respective support set for 10 inner-loop epochs, and finally evaluated on the corresponding query set. This cross-database transfer setting imposes stringent requirements on the model’s ability to learn universal representations across different data distributions and to rapidly adapt to unseen tasks from a distinct domain.

Table R44 demonstrated that AdaMBind outperformed all baseline models on the KIBA→BindingDB cross-domain transfer path. Specifically, AdaMBind achieved the best performance across multiple key metrics, including a CI of 0.7458, an R^2 of 0.3446, a Spearman correlation coefficient of 0.5326, and a Pearson correlation coefficient of 0.5961, while its MSE of 0.9623 ranked second. It should be noted that, compared to the results obtained from training and testing directly on the BindingDB dataset, all models exhibited a certain degree of performance degradation in the cross-domain setting, which was an expected consequence of distribution shift between domains. Nevertheless, AdaMBind consistently maintained its leading position under these challenging cross-domain conditions, demonstrating superior robustness and generalization capability. This advantage further highlighted the effectiveness of the meta-learning framework in capturing general interaction patterns across diverse protein families and compound structures. The results have been added to the revised supplementary information. (The corresponding revision can be seen in Table S3 of the supplementary information)

In summary, the cross-domain generalization experiment further confirms the strong out-of-distribution generalization capability of AdaMBind, providing empirical support for its application in real-world drug discovery scenarios involving data distribution shifts and cold-start problems. We have added the above experiments to the manuscript. (The corresponding revision can be seen Line 363-382 on Page 21-22 of the revised manuscript)

Table R44 Comparison of cross-domain transfer performance on KIBA→BindingDB

Methods	MSE(↓)	CI(↑)	R2(↑)	Spearman(↑)	Pearson(↑)
ColdDTA	0.9937 (0.0380)	0.7213 (0.0217)	0.2842 (0.1054)	0.4630 (0.0835)	0.5503 (0.0679)
Co-VAE	1.0283 (0.2440)	0.6853 (0.0486)	0.2176 (0.0787)	0.3834 (0.1395)	0.5212 (0.0914)
HiSIF	1.2328 (0.1207)	0.7209 (0.0305)	0.2772 (0.1366)	0.4387 (0.1081)	0.5280 (0.0982)
PSICHIC	0.9604 (0.1513)	0.7359 (0.0452)	0.3066 (0.1180)	0.5182 (0.0587)	0.5695 (0.0957)
DeepDTA	2.5974 (0.2652)	0.6024 (0.0401)	-0.2424 (0.2709)	0.2623 (0.1381)	0.1882 (0.1331)
MetaDTA	1.1047 (0.4644)	0.7592 (0.0436)	0.2824 (0.2743)	0.4936 (0.0935)	0.5639 (0.1254)
CML	1.9214 (0.0694)	0.7000 (0.0248)	0.1847 (0.1036)	0.3526 (0.1149)	0.3845 (0.1480)
ZeroBind	0.9598 (0.2069)	0.7241 (0.0423)	0.2239 (0.0705)	0.4613 (0.1251)	0.5230 (0.0994)
AdaMBind	0.9623 (0.1003)	0.7458 (0.0374)	0.3446 (0.1410)	0.5326 (0.1244)	0.5961 (0.1019)

The best results are highlighted in bold, while the second-best results are underlined. Five independent replications of each method were performed (n = 5). Data are expressed as means (std).

5. It is unclear why Pearson correlation was not included among the evaluation metrics, as it is commonly used in related studies.

Response:

Thank you for this important suggestion. We fully agree on the significance of the Pearson correlation coefficient in related studies and have supplemented this metric in the revised version. We now include the **Pearson correlation coefficient alongside Spearman correlation, Concordance Index (CI), Mean Squared Error (MSE), and R²** as core evaluation metrics to provide a more comprehensive performance analysis. The Pearson correlation coefficient effectively assesses the linear relationship between predicted and true values, while Spearman correlation is more robust to monotonic relationships, and their combination offers a more complete reflection of the model's predictive capability.

Table R45 Spearman and Pearson on the Davis dataset in the random task split

Methods	Majority		Few-shot	
	Spearman(\uparrow)	Pearson(\uparrow)	Spearman(\uparrow)	Pearson(\uparrow)
DeepDTA	0.5814 (0.0467)	0.6567 (0.0378)	0.5032 (0.0232)	0.5279 (0.0214)
ColdDTA	0.6093 (0.0553)	0.6637 (0.0381)	0.3772 (0.0269)	0.4548 (0.0640)
Co-VAE	0.5329 (0.0440)	0.6524 (0.0554)	0.3561 (0.0572)	0.4075 (0.0573)
HiSIF	0.6118 (0.0660)	0.6946 (0.0449)	0.4908 (0.0363)	0.5232 (0.0395)
MetaDTA	0.5376 (0.0712)	0.6205 (0.0335)	0.4794 (0.0482)	0.5099 (0.0398)
CML	0.5788 (0.0393)	0.5892 (0.0351)	0.2874 (0.0510)	0.2728 (0.0708)
PSICHIC	0.6037 (0.0517)	0.6624 (0.0465)	0.4166 (0.0441)	0.4525 (0.0254)
ZeroBind	0.5179 (0.0454)	0.5489 (0.0513)	0.3135 (0.0566)	0.3335 (0.0479)
AdaMBind(ours)	0.6257 (0.0389)	0.6878 (0.0385)	0.4904 (0.0205)	0.5499 (0.0187)

The best results are highlighted in bold, while the second-best results are underlined. Five independent replications of each method were performed ($n = 5$). Data are expressed as means (std).

Table R46 Spearman and Pearson on the KIBA dataset in the random task split

Methods	Majority		Few-shot	
	Spearman(\uparrow)	Pearson(\uparrow)	Spearman(\uparrow)	Pearson(\uparrow)
DeepDTA	0.4779 (0.0231)	0.5386 (0.0199)	0.2391 (0.0665)	0.2606 (0.0639)
ColdDTA	0.5541 (0.0211)	0.5960 (0.0105)	0.2418 (0.0957)	0.2437 (0.1023)
Co-VAE	0.4610 (0.0472)	0.5242 (0.0203)	0.3577 (0.0472)	0.4127 (0.0412)
HiSIF	0.4749 (0.0619)	0.5434 (0.0473)	0.2517 (0.0337)	0.2247 (0.0291)
MetaDTA	0.3368 (0.0512)	0.3637 (0.0345)	0.3097 (0.0482)	0.3563 (0.0565)
CML	0.4196 (0.0813)	0.3597 (0.0550)	0.3457 (0.0669)	0.2636 (0.0686)
PSICHIC	0.5409 (0.0596)	0.5404 (0.0373)	0.3276 (0.0579)	0.2899 (0.0534)
ZeroBind	0.4562 (0.0495)	0.4810 (0.0277)	0.2849 (0.0538)	0.3118 (0.0520)
AdaMBind(ours)	0.6528 (0.0545)	0.6559 (0.0297)	0.4847 (0.0504)	0.5062 (0.0653)

The best results are highlighted in bold, while the second-best results are underlined. Five independent replications of each method were performed ($n = 5$). Data are expressed as means (std).

Table R47 Spearman and Pearson on the BindingDB dataset in the random task split

Methods	Majority		Few-shot	
	Spearman(↑)	Pearson(↑)	Spearman(↑)	Pearson(↑)
DeepDTA	0.5486 (0.0848)	0.5594 (0.0975)	0.4878 (0.1211)	0.5153 (0.1005)
ColdDTA	0.5919 (0.0220)	0.5956 (0.0370)	0.4858 (0.0800)	0.5022 (0.0566)
Co-VAE	0.4257 (0.1282)	0.4713 (0.1257)	0.4237 (0.0254)	0.4701 (0.0332)
HiSIF	0.5817 (0.0467)	0.6387 (0.0594)	0.4839 (0.0873)	0.5103 (0.0794)
MetaDTA	0.5991 (0.0340)	0.6530 (0.0352)	0.5011 (0.0423)	0.5282 (0.0398)
CML	0.5735 (0.0340)	0.6392 (0.0352)	0.4637 (0.0862)	0.5050 (0.0780)
PSICHIC	0.5732 (0.0846)	0.5821 (0.0653)	0.4612 (0.0417)	0.4378 (0.0570)
ZeroBind	0.5675 (0.0463)	0.6176 (0.0531)	0.4623 (0.0874)	0.5009 (0.0812)
AdaMBind(ours)	0.6578 (0.0413)	0.6966 (0.0430)	0.5323 (0.0758)	0.5634 (0.0590)

The best results are highlighted in bold, while the second-best results are underlined. Five independent replications of each method were performed (n = 5). Data are expressed as means (std).

As shown in **Table R44-Table R47**, under the random task split setting, AdaMBind demonstrated significant advantages in both Spearman and Pearson correlation metrics across both the majority and few-shot settings.

On the Davis dataset, in the majority setting, AdaMBind achieved a Spearman of 0.6257, ranking first, and Pearson of 0.6878, second only to HiSIF (0.6946). In the few-shot setting, it attained a Pearson of 0.5499, outperforming all baselines (best baseline: 0.5279).

On the KIBA dataset, in the majority setting, AdaMBind's Spearman (0.6528) and Pearson (0.6559) significantly surpassed all baselines. In the few-shot setting, it also delivered strong performance (Spearman 0.4847, Pearson 0.5062, ranking first).

On the BindingDB dataset, AdaMBind achieved the best Spearman (0.6578) and Pearson (0.6966) in the majority setting. In the few-shot setting, its Pearson (0.5634) and Spearman (0.5323) outperformed all baselines.

As shown in Table R48-Table R50, in the more challenging novel task split, AdaMBind exhibited even greater robustness and generalization capability across both

data scenarios.

Table R48 Spearman and Pearson on the Davis dataset in the novel task split

Methods	Majority		Few-shot	
	Spearman(\uparrow)	Pearson(\uparrow)	Spearman(\uparrow)	Pearson(\uparrow)
DeepDTA	0.5560 (0.0425)	0.6097 (0.0691)	0.4687 (0.0304)	0.4946 (0.0517)
ColdDTA	0.5722 (0.0261)	0.6316 (0.0613)	0.4503 (0.0476)	0.4795 (0.0927)
Co-VAE	0.5704 (0.0291)	0.6517 (0.0215)	0.4043 (0.0083)	0.4358 (0.0222)
HiSIF	0.5673 (0.0654)	0.5966 (0.0533)	0.4618 (0.0835)	0.4784 (0.0972)
MetaDTA	0.6606 (0.0542)	0.6454 (0.0506)	0.4749 (0.0281)	0.5224 (0.0529)
CML	0.5288 (0.0525)	0.5592 (0.0723)	0.3028 (0.0582)	0.2957 (0.0654)
ZeroBind	0.5248 (0.0479)	0.5713 (0.0442)	0.3260 (0.0654)	0.3135 (0.0789)
PSICHIC	0.5668 (0.0513)	0.6051 (0.0544)	0.4086 (0.0786)	0.4535 (0.0741)
AdaMBind(ours)	0.5978 (0.0633)	0.6702 (0.0623)	0.4926 (0.0971)	0.5350 (0.0900)

The best results are highlighted in bold, while the second-best results are underlined. Five independent replications of each method were performed ($n = 5$). Data are expressed as means (std).

Table R49 Spearman and Pearson on the KIBA dataset in the novel task split

Methods	Majority		Few-shot	
	Spearman(\uparrow)	Pearson(\uparrow)	Spearman(\uparrow)	Pearson(\uparrow)
DeepDTA	0.3985 (0.0845)	0.4857 (0.0661)	0.2775 (0.0571)	0.2937 (0.0576)
ColdDTA	0.5224 (0.0071)	0.5352 (0.0172)	0.3340 (0.0372)	0.3709 (0.0072)
Co-VAE	0.4194 (0.0353)	0.4781 (0.0131)	0.1065 (0.0867)	0.1121 (0.0907)
HiSIF	0.4260 (0.1290)	0.4669 (0.0995)	0.1937 (0.0983)	0.2229 (0.0887)
MetaDTA	0.3170 (0.0981)	0.3568 (0.1190)	0.3864 (0.0816)	0.4341 (0.0854)
CML	0.4289 (0.0274)	0.3698 (0.0476)	0.2675 (0.1068)	0.2296 (0.0798)
ZeroBind	0.3928 (0.0973)	0.4647 (0.0822)	0.2324 (0.0685)	0.2681 (0.0714)
PSICHIC	0.3702 (0.0842)	0.3602 (0.0761)	0.1078 (0.0897)	0.1158 (0.0757)
AdaMBind(ours)	0.6279 (0.0739)	0.6239 (0.0701)	0.4717 (0.0971)	0.5543 (0.0900)

The best results are highlighted in bold, while the second-best results are underlined. Five independent replications of each method were performed ($n = 5$). Data are expressed as means (std).

Table R50 Spearman and Pearson on the BindingDB dataset in the novel task split

Methods	Majority		Few-shot	
	Spearman(\uparrow)	Pearson(\uparrow)	Spearman(\uparrow)	Pearson(\uparrow)
DeepDTA	0.4754 (0.0682)	0.4639 (0.0820)	0.3875 (0.0282)	0.3730 (0.0591)
ColdDTA	0.5810 (0.0661)	0.6175 (0.0440)	0.4392 (0.0961)	0.4252 (0.0954)
Co-VAE	0.5481 (0.0757)	0.5991 (0.0946)	0.3996 (0.1228)	0.3936 (0.1067)
HiSIF	0.5873 (0.0786)	0.6102 (0.0817)	0.4489 (0.0419)	0.4756 (0.0573)
MetaDTA	0.6086 (0.0924)	0.6313 (0.0876)	0.4508 (0.0812)	0.4718 (0.1059)
CML	0.5287 (0.0993)	0.5567 (0.0805)	0.4529 (0.0451)	0.5012 (0.0631)
ZeroBind	0.5391 (0.1106)	0.6001 (0.0978)	0.4519 (0.0787)	0.4656 (0.0803)
PSICHIC	0.5328 (0.0816)	0.5282 (0.0709)	0.3100 (0.0891)	0.3662 (0.0752)
AdaMBind(ours)	0.6064 (0.1205)	0.6407 (0.0869)	0.4703 (0.0505)	0.5171 (0.0421)

The best results are highlighted in bold, while the second-best results are underlined. Five independent replications of each method were performed ($n = 5$). Data are expressed as means (std).

On the Davis dataset, in the majority setting, it achieved Spearman and Pearson of 0.5978 and 0.6702, respectively, demonstrating stable performance. In the few-shot setting, AdaMBind showed superior Pearson (0.5350) compared to most baselines.

On the KIBA dataset, in the majority setting, AdaMBind's Spearman (0.6279) and Pearson (0.6239) far exceeded those of the best baseline (ColdDTA: 0.5224 and 0.5352). In the few-shot setting, it maintained a leading position (Spearman 0.4717, Pearson 0.5543).

On the BindingDB dataset, in the majority setting, AdaMBind (Spearman 0.6064, Pearson 0.6407) performed comparably to the strongest baseline, MetaDTA (Spearman 0.6086, Pearson 0.6313). In the few-shot setting, its Spearman reached 0.4703, remaining competitively advantageous, indicating the model's stable cross-task predictive consistency across tasks with varying similarity levels.

In summary, under both conventional random split and the novel task split, AdaMBind consistently outperformed or matched state-of-the-art methods in Spearman

and Pearson correlation metrics across majority and few-shot settings. The results are added to the *Results* section revised manuscript. (The corresponding revision can be seen Line 183-485 on Page 10-28)

6. The details of the relevant hyperparameters are not clearly reported. I recommend that the authors provide a table or similar format specifying the optimal hyperparameter settings for transparency and reproducibility.

Response:

Thank you for your valuable comments on our manuscript. We included all optimal hyperparameter settings for each model in tabular form within the revised manuscript, as shown in **Table R51**.

Table R51 The processes of AdaMBind parameter tuning

Hyper parameters	range
noise	0.1, 0.2, 0.3, 0.4, 0.5 , 0.6
Inner learning rate	1e-3, 5e-4, 1e-4 , 5e-5, 1e-5
Outer learning rate	5e-4, 1e-4, 5e-5, 1e-5
Number of candidate tasks B	12, 13, 14, 15 , 16
Number of selected tasks N_{sct}	4, 6, 8 , 10
Batch size in the support set	2, 4, 8 , 16

Additionally, we provided detailed supplementary notes on the hyperparameter optimization process. For the key hyperparameters of each major component (e.g., noise, outer learning rate, inner learning rate, Batch size in support set, number of candidate tasks B , Number of selected tasks N_{sct} , etc.), we adopted a grid search approach to explore different hyperparameter combinations. First, we conducted a series of experiments to jointly evaluate the influence of the inner learning rate and the batch size in the support set on model performance. Second, we explored the combined effect of the outer learning rate and the number of candidate tasks B . Then, we examined how the number of selected tasks N_{sct} impacts the model performance. Finally, we explored the effect of noise on model performance. As shown in Figure R4 (corresponding to Supplementary Table S9 of the revised Supplementary Information),

the final selected hyperparameter combination demonstrates strong performance in terms of MSE.

The above results have been added to the supplementary information. (corresponding to Supplementary Table S9 and Figure S2 of the revised Supplementary Information)

Figure R4 Impact of hyperparameter on MSE

7. Line 767 mentions that “This noise was sampled from a uniform distribution in the range $[-0.5, 0.5]$.” Intuitively, this level of noise seems relatively large compared with the model's MSE. Did the authors conduct any experiments to test the effect of different noise levels or distributions on model performance? It would be helpful to include such an analysis.

Response:

Thank you for your valuable comment. We conducted the ablation study on this key hyperparameter to further validate and optimize the regularization strategy of our model.

We performed the ablation study across three benchmark datasets (Davis, BindingDB, and KIBA), testing noise levels in the range of [0.1, 0.6] at intervals of 0.1. The results (Figure R5) show that both the BindingDB and KIBA datasets achieved the lowest validation mean squared error (MSE) at a noise level of 0.5, indicating that this setting provides the optimal regularization effect for these two datasets. For the Davis dataset, the best performance was observed at a noise level of 0.2. However, it is worth noting that when the noise level was set to 0.5, the performance (MSE = 0.2490) differed only marginally from the optimal value (MSE = 0.2470). Moreover, the model trained with a noise level of 0.5 exhibited lower performance variability across repeated experiments compared to that trained with a noise level of 0.2.

Based on the above analysis, we ultimately decided to uniformly adopt a noise level of 0.5 in our model.

Figure R5 Impact of Noise on MSE

8. Regarding the task-adaptive module, it is unclear in what form the inputs are provided. Moreover, the rationale for using an LSTM is not evident, as there does not appear to be any sequential or temporal relationship in the data. Could the authors clarify this design choice and, if possible, visualize or provide the learned weights to help interpret the module's behavior?

Response:

Thank you for raising this important question. First, we have supplemented the description of the input format. Second, we clarified the technical rationale behind

selecting LSTM as our feature extractor. Thirdly, we demonstrated its advantages over other architectures through empirical comparisons. Finally, we visualized the learned weights from adaptive task module.

(1) Clarification of the Input Form

In the training phase, the raw data were first partitioned into multiple sub-tasks based on different protein targets, with each task consisting of a set of drugs interacting with the target and their corresponding affinity values. Subsequently, to prevent the model from being dominated by noisy tasks during training, the adaptive task module dynamically evaluates the value of each task and performs task sampling. **The sampling weights are determined by two key factors: (i) the loss on the query set of each task and (ii) the gradient similarity between the support set and the query set of each task.** After sampling, the meta-learning module executes meta-training based on the selected tasks. (corresponding revision can be seen on Line 903-910 on Page 50-51)

To clarify the technical details, during meta-training, the adaptive task module receives two key inputs for the task:

(i) Query set loss: This scalar value reflects the prediction error of a task after inner-loop adaptation, i.e., $Loss(Q_i; \theta_i^{(k)})$, where $\theta_i^{(k)}$ represents the specific-task model parameters for task T_i in k-th meta update, Q_i represents the query set of task T_i .

(ii) Gradient-similarity matrix: This matrix is obtained by computing the cosine similarity between the gradients of the support set and the query set for every parameter group in the base learner, i.e., $\left\langle \nabla_{\theta_0^{(k)}} Loss(Q_i; \theta_0^{(k)}), \nabla_{\theta_0^{(k)}} Loss(S_i; \theta_0^{(k)}) \right\rangle$ where S_i represents the support set of task T_i .

Assuming the adaptive task module is represented as $g_\phi(\cdot)$, the process of generating sampling probabilities in k-th meta update can be described as :

$$\omega_i^{(k)} = g\left(Loss(Q_i; \theta_i^{(k)}), \left\langle \nabla_{\theta_0^{(k)}} Loss(Q_i; \theta_0^{(k)}), \nabla_{\theta_0^{(k)}} Loss(S_i; \theta_0^{(k)}) \right\rangle; \phi^{(k)}\right) \quad (1)$$

(2) Clarification on LSTM Design Choice

We thank the reviewer for raising this important question regarding the LSTM

design in our adaptive task module. We employ the LSTM as a feature extractor rather than for sequence modeling. The key advantage of using LSTM over alternatives like MLP, CNN lies in its unique capability to process this specific input type. Gradient similarity signals naturally contain noise with uneven distribution across layers. The LSTM's gating mechanisms, particularly the forget gates, enable automatic down-weighting of unreliable similarity measurements from certain parameter groups while preserving consistent signals from others³. This dynamic feature weighting proves more effective than other static weighting architectures, especially when handling inherent noise in gradient-based signals. (The corresponding revision can be seen Line 914-920 on Page 51)

(3) Experimental Comparison between different feature extractor architectures

To evaluate the architectural choice for the adaptive task module, we conducted comparative experiments between LSTM, MLP, and CNN implementations on three datasets under the majority setting. The performance was measured using the Mean Squared Error (MSE) on the validation set. As shown in **Table R52**, the LSTM-based adaptive task module consistently outperformed the other architectures across all datasets, demonstrating its effectiveness in handling the task scheduling problem in meta-learning scenarios.

These empirical results validate our design choice and demonstrate that the LSTM's architectural properties are particularly well-suited for the task scheduling problem in meta-learning scenarios.

Table R52 MSE of feature extractor architectures for adaptive task module

	BindingDB	Davis	KIBA
LSTM	0.3140 (0.0749)	0.1733 (0.0290)	0.2595 (0.0325)
MLP	0.4052 (0.0957)	0.2252 (0.0324)	0.3363 (0.0389)
CNN	0.4357 (0.0854)	0.2481 (0.0317)	0.3672 (0.0426)

The best results are highlighted in bold. Five independent replications of each method were performed (n = 5). Data are expressed as means (std).

(4) Visualization of the weights from the adaptive task module

Regarding the visualization of the weights in the adaptive task module, we have presented it in **Fig.6** (Page 26-28 Line 453-484 in the revised manuscript) of the manuscript.

Figure R6. Visualization of the weights from the adaptive task module

As illustrated in Figure R6, we visualized the correlation between task sampling weights and these two factors across three benchmark datasets (BindingDB, KIBA, and Davis). The horizontal axis represents the gradient similarity between the support and query sets of each task, while the vertical axis represents the query loss. Each scatter point corresponds to a specific task, and the color of the point indicates its assigned sampling weight: warmer colors (red/orange) denote higher weights, while cooler colors (blue/cyan) denote lower weights.

Analysis of the visualization reveals two patterns:

- a. Tasks with higher query loss (indicating greater difficulty or noise) tend to receive lower sampling weights.
- b. Tasks with higher gradient similarity between support and query sets (suggesting better alignment in optimization direction) are assigned higher sampling weights.

These results indicated that the adaptive task module follows an "easy-to-hard" learning strategy, prioritizing tasks that are more consistent with the current meta-learner's state and exhibit better generalization potential. This mechanism effectively suppresses noisy or outlier tasks and enhances the model's ability to learn transferable meta-knowledge.

9. The manuscript mentions that the validation set is used to determine the parameters of the adaptive task module. Should this validation set be further divided into support and query subsets, similar to standard meta-learning practice? Clarification on this point would help understand the training and evaluation procedure.

Response:

Thank you for your insightful question regarding the data partitioning of the validation set. We explained the division of support and query subsets within the validation set, and then provided a clarification of the overall training and evaluation procedure.

(1) Explanation of support and query set in the meta-validation tasks

We apologize for not making this point sufficiently explicit in the original manuscript. We have revised the *Methods* section in the revised manuscript to clearly state that the meta-validation set is structured with support and query subsets, thereby enhancing the readability and transparency of our training and evaluation procedure. (The corresponding revision can be seen Line 835-843 on Page 47 of the revised manuscript)

In classical meta-learning algorithms, including the foundational MAML framework, it is indeed standard and correct practice to partition the meta-validation set into support and query subsets¹. In AdaMBind, we strictly adhere to this principle: Our meta-validation set is structured into support and query subsets for each task.

(2) Supplementary Explanation of the AdaMBind Training Process

We apologize for any lack of clarity in the writing of the *Methods* section and have revised it to improve comprehensibility. (The corresponding revision can be seen Line 862-955 on Page 48-53 of the revised manuscript) The training procedures of the meta-learner and the adaptor can be described as follows.

a. Task Sampling & Meta-Training. During a single meta-update iteration, the adaptor (with current parameter ϕ) evaluates all candidate tasks in the meta-training pool. It computes a sampling weight for each task based on its query loss and support-query gradient similarity. A batch of tasks is then sampled according

to these weights for the meta-training, leading to the update of the meta-learner parameters θ_0 (to θ'_0).

b. Validation for Adaptor Guidance. After the meta-learner θ_0 is updated to θ'_0 , we use θ'_0 to evaluate the meta-validation tasks. Crucially, for each meta-validation task, we use its support set to perform a few-step adaptation of the base learner, just as done during training. We then use its query set to compute the prediction loss of this adapted model.

c. Optimizing the Adaptor. The combined loss on the **query sets of the meta-validation tasks serves as the training signal for the adaptor**. If the tasks selected by the adaptor leads to a meta-learner θ'_0 that performs poorly on the meta-validation query sets after adaptation, it indicates a poor sampling strategy of adaptor. Conversely, good performance validates the strategy. The gradients from this validation loss are then used to update the adaptor parameters ϕ to ϕ' .

In addition, we added a pseudocode for the detailed training procedure in Algorithm 1, as shown below (The corresponding revision can be seen Algorithm S1 in Supplementary Information.)

Algorithm 1: AdaMBind training process

Require: learning rates α, β , training set $\{T_k\}_{k=1}^{N_{tr}}$, validation set $\{T_k\}_{k=1}^{N_{val}}$, candidate task size B , selected task set size N_{slect}

- 1: Initialize the meta learner $\theta_0^{(0)}$ and the adaptor $\phi^{(0)}$
 - 2: **for** k-th meta update **do**:
 - 3: Randomly sample B tasks from training set $\{T_k\}_{k=1}^{N_{tr}}$ to form candidate task set $\{T_k\}_{k=1}^B$
 - 4: **for** each task T_i in candidate task set **do**:
 - 5: Initialize task-specific base learner with $\theta_0^{(k)}$
 - 6: update base learner via several steps, compute $Loss(Q_i; \theta_i^{(k)})$ and $\langle \nabla_{\theta_0^{(k)}} Loss(Q_i; \theta_0^{(k)}), \nabla_{\theta_0^{(k)}} Loss(S_i; \theta_0^{(k)}) \rangle$
 - 7: generate sampling probability $\omega_i^{(k)}$ by Eqn.(2)
 - 8: **end for**
 - 9: choose N_{slect} via sampling probability to form selected task set $\{T_k\}_{k=1}^{N_{slect}}$
 - 10: compute approximate $\theta_0^*(\phi)$ by one -step gradient strategy as Eqn.(4)
 - 11: Randomly sample B tasks from validation set $\{T_k\}_{k=1}^{N_{val}}$, update adaptor $\phi^{(k+1)}$ by Eqn.(3)
 - 12: generate sampling probability $\omega_i^{(k+1)}$ for candidate task set by Eqn.(2) via using updated $\phi^{(k+1)}$
 - 13: Resample another N_{slect} tasks to form selected task set $\{T_k\}_{k=1}^{N_{slect}}$
 - 14: update meta learner $\theta_0^{(k+1)}$ by Eqn.(5)
 - 15: **end for**
-

During the k-th meta update, for candidate task $T_i \in \{T_i\}_{i=1}^B$ and meta learner $\theta_0^{(k)}$, we can define an adaptor $g_\phi(\cdot)$ that generates task sampling weights ω_i with parameter ϕ :

$$\omega_i^{(k)} = g\left(Loss(Q_i; \theta_i^{(k)}), \langle \nabla_{\theta_0^{(k)}} Loss(Q_i; \theta_0^{(k)}), \nabla_{\theta_0^{(k)}} Loss(S_i; \theta_0^{(k)}) \rangle; \phi^{(k)}\right) \quad (2)$$

The update and optimization of the adaptor ϕ can be described by the following formula:

$$\phi^{(k+1)} = \phi^{(k)} - \beta \cdot \nabla_\phi E_{T_{val} \sim p(T_{val})} [Loss(T_{val}; \tilde{\theta}_0^{(k+1)}(\phi^{(k)}))] \quad (3)$$

$$\tilde{\theta}_0^{(k+1)}(\phi^{(k)}) = \theta_0^{(k)} - \beta \cdot E_{T_{slect} \sim p(T_{slect})} [\nabla_\theta Loss(T_{slect}; \theta_0^{(k)}, \phi^{(k)})] \quad (4)$$

The update and optimization of the meta learner $\phi^{(k+1)}$ can be described by the following formula:

$$\theta_0^{(k+1)}(\phi^{(k)}) = \theta_0^{(k)} - \beta \cdot E_{T_{slect} \sim P(T_{slect})} [\nabla_{\theta} Loss(T_{slect}; \theta_0^{(k)}, \phi^{(k+1)})] \quad (5)$$

During testing phase, we directly load the trained meta-learner's initial parameters, perform a few fine-tuning steps on the support set of the meta-test task, and then make predictions on the query set.

10. The dataset splitting does not appear to consider sequence similarity (e.g., clustering with CD-HIT). The authors may refer to relevant literature and consider incorporating such a procedure to ensure that the training, validation, and test sets are appropriately non-redundant, which would provide a more rigorous evaluation of model generalization.

Response:

Thank you for your valuable comment. We fully agree that considering protein sequence similarity during dataset splitting is crucial for rigorously evaluating the model's generalization. Accordingly, this point has been addressed in the revised manuscript.

(1) Newly added novel task split based on protein sequence clustering

Specifically, we introduced a new data-splitting strategy based on protein sequence clustering in the revised manuscript, as detailed in the *Data split* subsection of the *Methods* section. (The corresponding revision can be seen Line 232-244 on Page 13, Line 844-861 on Page 47-48 of the revised manuscript) . Following the methodology adopted in related DTA studies^{4,5}, we refer to this task partitioning scheme as the “**novel task split.**”

We employed CD-HIT with a stringent 40% sequence identity threshold to cluster all target proteins. Proteins with sequence similarity exceeding 40% were grouped into the same cluster, while those below this threshold were assigned to different clusters. The clusters were then randomly split into meta-training, meta-validation, and meta-testing tasks in an 8:1:1 ratio. All proteins across training, validation, and testing

clusters share less than 40% sequence similarity. This design ensures that meta-testing tasks consist of targets that are structurally and sequentially distinct from those encountered during training, thereby simulating a realistic and challenging “cold-target” scenario in which the model must generalize to novel protein families with low similarity.

(2) Model comparison under novel task split

We added experimental results based on the novel task split to the revised manuscript (The corresponding revision can be seen Line 302-352 on Page 17-21 of the revised manuscript), comparing the performance of AdaMBind against multiple baseline models under both the Majority and Few-shot settings:

Table R53-Table R55 (corresponding to Fig.3 of the revised manuscript) demonstrated the performance of AdaMBind **under the majority setting with the novel task split**. AdaMBind achieved highly competitive performance across all three benchmark datasets, outperforming other baseline models in the majority of evaluation metrics. Specifically, it attained the best scores in CI (0.7495), R^2 (0.3738), Pearson (0.6279), and Spearman (0.6239) on the KIBA dataset. On the BindingDB dataset, AdaMBind showed superior performance in MSE (0.8758) and R^2 (0.4114), with its CI (0.7689) being nearly identical to the top-performing baseline. Although some baselines achieved optimal results in specific metrics on the Davis dataset (e.g., Co-VAE in MSE and CI, MetaDTA in Spearman), AdaMBind consistently ranked within the top two performers on most of the metrics, demonstrating robust and stable predictive capability. These results confirmed that AdaMBind maintained strong generalization performance even under the challenging cold-target scenario with sufficient support samples.

Table R53 Performance on Davis in novel task split (Majority)

Methods	MSE(↓)	CI(↑)	R2(↑)	Spearman(↑)	Pearson(↑)
DeepDTA	0.6230 (0.2530)	0.7853 (0.0335)	0.3346 (0.0997)	0.5560 (0.0425)	0.6097 (0.0691)
ColdDTA	0.5731 (0.2184)	0.7940 (0.0180)	0.3809 (0.0914)	0.5722 (0.0261)	0.6316 (0.0613)
Co-VAE	0.4960 (0.0831)	0.8187 (0.0103)	0.4168 (0.0281)	0.5704 (0.0291)	0.6517 (0.0215)
HiSIF	0.6407 (0.2614)	0.7887 (0.0325)	0.3215 (0.0723)	0.5673 (0.0654)	0.5966 (0.0533)
MetaDTA	0.6714 (0.1780)	0.8163 (0.0152)	0.4156 (0.0689)	0.6606 (0.0542)	0.6454 (0.0506)
CML	0.6301 (0.2050)	0.7680 (0.0141)	0.3117 (0.0798)	0.5288 (0.0525)	0.5592 (0.0723)
ZeroBind	0.6506 (0.1684)	0.7582 (0.0265)	0.2882 (0.0480)	0.5248 (0.0479)	0.5713 (0.0442)
PSICHIC	0.8463 (0.2298)	0.7739 (0.0487)	0.2915 (0.0681)	0.5668 (0.0513)	0.6051 (0.0544)
AdaMBind(ours)	0.5080 (0.1823)	0.8066 (0.0271)	0.4460 (0.0808)	0.5978 (0.0633)	0.6702 (0.0623)

The best results are highlighted in bold, while the second-best results are underlined. Five independent replications of each method were performed (n = 5). Data are expressed as means (std).

Table R54 Performance on KIBA in novel task split (Majority)

Methods	MSE(↓)	CI(↑)	R2(↑)	Spearman(↑)	Pearson(↑)
DeepDTA	0.5160 (0.1091)	0.6512 (0.0346)	0.1409 (0.1374)	0.3985 (0.0845)	0.4857 (0.0661)
ColdDTA	0.4851 (0.0771)	0.7140 (0.0247)	0.2821 (0.0056)	0.5224 (0.0071)	0.5352 (0.0172)
Co-VAE	0.5159 (0.0950)	0.6595 (0.0164)	0.1071 (0.0319)	0.4194 (0.0353)	0.4781 (0.0131)
HiSIF	0.4754 (0.0911)	0.6632 (0.0524)	0.2105 (0.0906)	0.4260 (0.1290)	0.4669 (0.0995)
MetaDTA	0.5800 (0.0201)	0.6184 (0.0375)	0.0092 (0.1476)	0.3170 (0.0981)	0.3568 (0.1190)
CML	0.5332 (0.0773)	0.6605 (0.0121)	0.1120 (0.0458)	0.4289 (0.0274)	0.3698 (0.0476)
ZeroBind	0.5398 (0.0667)	0.6461 (0.0348)	0.1567 (0.0482)	0.3928 (0.0973)	0.4647 (0.0822)
PSICHIC	0.4680 (0.0465)	0.6411 (0.0369)	0.1113 (0.0549)	0.3702 (0.0842)	0.3602 (0.0761)
AdaMBind(ours)	0.3626 (0.0368)	0.7495 (0.0307)	0.3738 (0.1017)	0.6279 (0.0739)	0.6239 (0.0701)

The best results are highlighted in bold, while the second-best results are underlined. Five independent replications of each method were performed (n = 5). Data are expressed as means (std).

Table R55 Performance on BindingDB in novel task split (Majority)

Methods	MSE(↓)	CI(↑)	R2(↑)	Spearman(↑)	Pearson(↑)
DeepDTA	1.2675 (0.2743)	0.7081 (0.0348)	0.1749 (0.0734)	0.4754 (0.0682)	0.4639 (0.0820)
ColdDTA	0.8902 (0.0820)	0.7412 (0.0644)	0.3837 (0.0063)	0.5810 (0.0661)	0.6175 (0.0440)
Co-VAE	1.1189 (0.4251)	0.7447 (0.0436)	0.2571 (0.2233)	0.5481 (0.0757)	0.5991 (0.0946)
HiSIF	1.0064 (0.1674)	0.7385 (0.0597)	0.2846 (0.0813)	0.5873 (0.0786)	0.6102 (0.0817)
MetaDTA	0.9987 (0.3440)	0.7710 (0.0386)	0.3496 (0.1823)	0.6086 (0.0924)	0.6313 (0.0876)
CML	1.0545 (0.068)	0.7323 (0.0260)	0.2987 (0.0964)	0.5287 (0.0993)	0.5567 (0.0805)
ZeroBind	1.2703 (0.2417)	0.7274 (0.0358)	0.3349 (0.1257)	0.5391 (0.1106)	0.6001 (0.0978)
PSICHIC	1.634 (0.0483)	0.7146 (0.0357)	0.2185 (0.0987)	0.5328 (0.0816)	0.5282 (0.0709)
AdaMBind(ours)	0.8758 (0.0374)	0.7689 (0.0348)	0.4114 (0.1199)	0.6064 (0.1205)	0.6407 (0.0869)

The best results are highlighted in bold, while the second-best results are underlined. Five independent replications of each method were performed (n = 5). Data are expressed as means (std).

Table R56-Table R58 (corresponding to Fig.3 of the revised manuscript) presented the comparison results under the **few-shot setting**. AdaMBind demonstrated more pronounced and consistent advantages in this data-scarce scenario. It outperformed all baseline models on the KIBA dataset across MSE (0.4910), CI (0.6808), R2 (0.2983), and Spearman (0.5543). On the Davis dataset, AdaMBind achieved the best R² (0.2526) and Spearman (0.5350) scores, with its MSE (0.6128) being very close to the top performer. For the BindingDB dataset, AdaMBind ranked first in R² (0.2536) and Spearman (0.5171), while its MSE (1.2968) was comparable to the best-performing baseline. These results highlighted AdaMBind's exceptional capability to rapidly adapt to novel targets with extremely limited data.

Table R56 Performance on Davis in novel task split (Few-shot)

Methods	MSE(↓)	CI(↑)	R2(↑)	Spearman(↑)	Pearson(↑)
DeepDTA	0.6283 (0.1923)	0.7459 (0.0205)	0.2202 (0.0641)	0.4687 (0.0304)	0.4946 (0.0517)
ColdDTA	0.6464 (0.2420)	0.7373 (0.0409)	0.2145 (0.0898)	0.4503 (0.0476)	0.4795 (0.0927)
Co-VAE	0.6069 (0.0789)	0.7114 (0.0076)	0.1536 (0.0352)	0.4043 (0.0083)	0.4358 (0.0222)
HiSIF	0.6682 (0.2423)	0.7412 (0.0541)	0.1811 (0.0892)	0.4618 (0.0835)	0.4784 (0.0972)

Methods	MSE(↓)	CI(↑)	R2(↑)	Spearman(↑)	Pearson(↑)
MetaDTA	0.7664 (0.1847)	0.7784 (0.0156)	0.2347 (0.0652)	0.4749 (0.0281)	0.5224 (0.0529)
CML	0.7499 (0.1971)	0.6556 (0.0190)	0.0627 (0.0579)	0.3028 (0.0582)	0.2957 (0.0654)
ZeroBind	0.7723 (0.1552)	0.6285 (0.0277)	0.0935 (0.0746)	0.3260 (0.0654)	0.3135 (0.0789)
PSICHIC	0.7207 (0.1697)	0.7023 (0.0418)	0.2295 (0.0899)	0.4086 (0.0786)	0.4535 (0.0741)
AdaMBind(ours)	0.6128 (0.2205)	0.7591 (0.0186)	0.2526 (0.1101)	0.4926 (0.0971)	0.5350 (0.0900)

The best results are highlighted in bold, while the second-best results are underlined. Five independent replications of each method were performed (n = 5). Data are expressed as means (std).

Table R57 Performance on KIBA in novel task split (Few-shot)

Methods	MSE(↓)	CI(↑)	R2(↑)	Spearman(↑)	Pearson(↑)
DeepDTA	0.6932 (0.0784)	0.6017 (0.0200)	-0.0118 (0.0973)	0.2775 (0.0571)	0.2937 (0.0576)
ColdDTA	0.6532 (0.1565)	0.6241 (0.0306)	0.1055 (0.0306)	0.3340 (0.0372)	0.3709 (0.0072)
Co-VAE	0.8888 (0.1188)	0.5380 (0.0306)	-0.2686 (0.1261)	0.1065 (0.0867)	0.1121 (0.0907)
HiSIF	0.9372 (0.1552)	0.5681 (0.0510)	0.0514 (0.1056)	0.1937 (0.0983)	0.2229 (0.0887)
MetaDTA	0.6887 (0.1761)	0.6451 (0.0303)	-0.0342 (0.4096)	0.3864 (0.0816)	0.4341 (0.0854)
CML	0.7863 (0.1335)	0.5992 (0.0418)	-0.1383 (0.0940)	0.2675 (0.1068)	0.2296 (0.0798)
ZeroBind	0.7316 (0.1137)	0.5549 (0.0256)	0.0011 (0.1320)	0.2324 (0.0685)	0.2681 (0.0714)
PSICHIC	0.9733 (0.1550)	0.5375 (0.0396)	-0.1230 (0.1546)	0.1078 (0.0897)	0.1158 (0.0757)
AdaMBind(ours)	0.4910 (0.1403)	0.6808 (0.0442)	0.2983 (0.1101)	0.4717 (0.0971)	0.5543 (0.0900)

The best results are highlighted in bold, while the second-best results are underlined. Five independent replications of each method were performed (n = 5). Data are expressed as means (std).

Table R58 Performance on BindingDB in novel task split (Few-shot)

Methods	MSE(↓)	CI(↑)	R2(↑)	Spearman(↑)	Pearson(↑)
DeepDTA	1.6661 (0.3281)	0.6627 (0.0167)	0.0511 (0.1197)	0.3875 (0.0282)	0.3730 (0.0591)
ColdDTA	1.6989 (0.2111)	0.6770 (0.0958)	0.1724 (0.0392)	0.4392 (0.0961)	0.4252 (0.0954)
Co-VAE	1.3085 (0.3372)	0.6779 (0.0665)	0.0681 (0.0738)	0.3996 (0.1228)	0.3936 (0.1067)
HiSIF	1.5272 (0.1251)	0.6715 (0.0129)	0.1233 (0.0568)	0.4489 (0.0419)	0.4756 (0.0573)
MetaDTA	1.3613 (0.3719)	0.7240 (0.0448)	0.2184 (0.1476)	0.4508 (0.0812)	0.4718 (0.1059)
CML	1.4191 (0.2578)	0.6926 (0.0232)	0.1898 (0.0859)	0.4529 (0.0451)	0.5012 (0.0631)
ZeroBind	1.2610 (0.3165)	0.6817 (0.0361)	0.1443 (0.0587)	0.4519 (0.0787)	0.4656 (0.0803)
PSICHIC	1.8581 (0.2548)	0.6341 (0.0250)	0.1207 (0.0878)	0.3100 (0.0891)	0.3662 (0.0752)
AdaMBind(ours)	1.2968 (0.2389)	0.6980 (0.0169)	0.2536 (0.0352)	0.4703 (0.0505)	0.5171 (0.0421)

The best results are highlighted in bold, while the second-best results are underlined. Five independent replications of each method were performed ($n = 5$). Data are expressed as means (std).

(3) Support set size sensitivity analysis under novel task split

In the support set size sensitivity analysis, we supplemented the experimental results under the novel task split scenario. As shown in **Figure R7** (corresponding to Fig.4b of the revised manuscript), under the novel task split, AdaMBind maintained a stable and excellent performance trend. As the support set size per task increased from 5 to 40 samples, AdaMBind consistently achieved leading or near-optimal performance across all three benchmark datasets (BindingDB, KIBA, and Davis). Its R^2 steadily improved with increasing sample size, demonstrating robust and scalable learning capability. Importantly, even with very few support samples (e.g., 5 samples), the model still exhibited a clear performance advantage, highlighting its data-efficient adaptation. Together, these findings further validated that our model can rapidly and efficiently adapt using only limited sample information, enabling effective knowledge transfer even under data-scarce conditions. (The corresponding revision can be seen Line 408-410 on Page 24 of the revised manuscript)

Figure R7. Performance comparison under novel task split. Comparing AdaMBind and baseline models across different support set sizes (5, 10, 20, 30, 40). The evaluation metric is R^2 . Five independent replications of each method were performed ($n=5$). Data are expressed as means \pm std.

11. Regarding the use of five different random seeds, it is unclear whether these correspond to different dataset splits or only to variations in network initialization and training. The authors should clarify this, as it impacts the interpretation of the reported performance variability.

Response:

Thank you for raising this important point regarding the use of random seeds, which is crucial for interpreting the robustness of our results. We apologize for any lack of clarity in our original manuscript and are pleased to provide the following clarification.

In our study, the five different random seeds were employed to control stochasticity across the entire experimental pipeline, encompassing both (a) **task-level data partitioning (i.e., the assignment of tasks to meta-training, meta-validation, and meta-testing sets)** and (b) **model-level randomness (i.e., neural network parameter initialization and stochastic operations during training)**. This approach enables a thorough and rigorous assessment of model robustness, preventing the evaluation results from being potentially influenced by a single specific data partition or favorable initialization.

The above results have been added to the *Methods*. (The corresponding revision can be seen Line 1046-1055 on Page 57-58 of the revised manuscript)

12. It is unclear how task similarity is defined in this study. Additionally, the use of molecule-averaged fingerprints warrants further explanation—what is the physical or chemical significance of averaging the fingerprints across molecules?

Response:

Thank you for your insightful comment regarding the definition of task similarity and the physicochemical meaning of averaged molecular fingerprints. We clarify that the purpose of introducing this task similarity is to evaluate whether the generalization capability of meta-learning models is overly dependent on feature-level resemblance between test and training tasks. In addition, we acknowledge that the arithmetic mean of molecular fingerprints lacks clear physicochemical significance. Consequently, we employed UniMol models based on three-dimensional conformation to extract molecular representations and calculated their mean vectors to aggregate the chemical space of the task.

(1) The purpose of introducing task similarity

When evaluating the generalization capability of meta-learning models to novel tasks, a common assumption is that the model has acquired transferable knowledge from a diverse yet correlated set of training tasks. However, model performance can heavily rely on the similarity in feature distribution between test tasks and training tasks. This dependency may limit the model's applicability in real-world drug discovery scenarios.

In this study, we formulate learning tasks around protein targets: each task consists of a specific protein target, along with a set of drugs known to interact with it and their corresponding affinity data. Consequently, the characteristics of a task can be decomposed into two components: the biological properties encoded in the sequence of the target protein, and the chemical space defined by the drug that binds to that target. To investigate the generalization mechanism of meta-learning models, we introduce task similarity as a quantitative metric to measure the overall proximity between different tasks within the feature space. The primary objective of this analysis is to evaluate whether a model's performance on a novel task is overly reliant on its high feature-level resemblance to certain tasks in the training set. A meta-learning model

with strong generalization capability should be able to effectively apply the transferable knowledge distilled from a series of training tasks to new tasks whose feature distributions may differ significantly, rather than performing well only in scenarios highly similar to the training tasks.

(2) Review of the Initial Method

Therefore, in our initial analysis, we attempted to construct a comprehensive task-level representation vector. Specifically, we used the protein language model ESM-2 to extract a high-dimensional semantic embedding vector from the target's amino acid sequence. Simultaneously, to summarize the chemical space of drugs associated with the task, we performed element-wise arithmetic averaging of the molecular fingerprints (e.g., ECFP) of all drugs within that task, resulting in an "averaged fingerprint" vector. This concept of element-wise arithmetic averaging originates from the research of Qi et al.². The protein embedding and the averaged fingerprint were then concatenated, and the cosine similarity between such task representation vectors was calculated as the task similarity.

(3) Replacing fingerprint averaging with UniMol representation

We fully agree that element-wise averaging of molecular fingerprints lacks clear physicochemical interpretability, as it obscures structural information and reflects only a statistical central tendency.

To address this, we adopted UniMol⁶, a model pre-trained on molecular conformations that captures spatial atomic arrangements and provides a firmer physicochemical foundation, to adopt the more expressive and 3D-aware molecular representation. Specifically, for the set of drugs within a task, we calculate the mean vector of these drugs within the UniMol representation space to serve as a summary of the ligand chemical space for that task. This mean vector is then concatenated with the target protein's ESM-2 embedding to form the task representation. The task-pair cosine similarities have been recomputed using this refined measure.

(4) Re-elaboration of the experimental methodology

To systematically evaluate the robustness of AdaMBind against such distribution shifts, we designed an experiment to quantitatively analyze the relationship between

model performance and task similarity. We selected CML, ZeroBind, and MetaDTA as strong meta-learning baselines for comparison.

First, we define each task as a drug–target affinity prediction problem centered around a specific protein target. The similarity between tasks is jointly determined by two core aspects: the biological characteristics of the protein and the chemical properties of the associated drugs. To construct a task-level feature representation, we adopt the following approach:

For proteins, we utilize the pre-trained protein language model ESM-2 to extract deep semantic embedding vectors from their amino acid sequences. For drugs, we employ UniMol to generate embedding vectors for each molecule. To derive a unified representation of the chemical space at the task level, we compute the element-wise average of the UniMol embeddings of all drugs within a given task, resulting in a comprehensive "task-level drug representation". Finally, we concatenate the protein's ESM embedding with the averaged drug embedding to form the final feature vector for the task.

Based on this representation, we compute the cosine similarity between the feature vectors of any two tasks. The cosine similarity ranges from -1 to 1 , with values closer to 1 indicating higher similarity between tasks. For each test task, we calculate its similarity to all training tasks and define the set of training tasks whose similarity exceeds a predefined threshold (0.6) as the "similar training set", whose size is denoted as N_{sim} . Intuitively, N_{sim} reflects the number of "neighbors" of the test task within the training distribution.

To quantify the model's dependence on task similarity, we define the generalization gap as the difference between the actual performance (e.g., R^2) on the test task and the average performance on its similar training set. A smaller gap suggests that the model's test performance remains close to its performance on similar training tasks, implying lower dependence on task similarity.

(5) Description of results achieved in new task similarity evaluation methods

As the performance of several meta-learning models on the KIBA dataset has been

less than satisfactory, we take the KIBA dataset as an example. Figure R8 illustrates, via scatter plots and linear fitting curves, the relationship between the generalization gap and different models.

In Figure R8, it was observed that the generalization gaps of AdaMBind are clustered around zero and show no systematic shift as N_{sim} increases, indicating that its performance is only minimally influenced by task similarity. Linear regression analysis further confirmed this observation: AdaMBind exhibited only a weak and statistically non-significant positive correlation between its generalization gap and N_{sim} (Pearson $r=0.093$, $p=0.786$). This implies that regardless of whether a test task has many similar “neighbors” (high N_{sim}) or very few (low N_{sim}) in the training distribution, AdaMBind’s actual performance (R^2) on that task remained stable and comparable to the level achieved on its similar training tasks.

In contrast, the other meta-learning baselines demonstrated varying degrees of significant dependence on task similarity. Specifically, the generalization gaps of CML, ZeroBind, and MetaDTA all showed a clear positive correlation trend with N_{sim} , with correlation coefficients that are statistically significant (CML: Pearson $r=0.612$, $p=0.045$; ZeroBind: Pearson $r=0.830$, $p=0.002$) or indicated a strong correlational trend (MetaDTA: Pearson $r=0.486$, $p=0.129$). This pattern revealed a critical limitation: the strong performance of these models heavily relied on the test task having access to a sufficient number of highly similar “support” tasks in the training set. When confronted with a novel target that has few neighbors in the training distribution (i.e., a low N_{sim} scenario), their generalization performance degraded markedly (the generalization gap becomes a large negative value), indicating that the meta knowledge they had acquired possesses limited cross-task transferability and struggled to effectively handle genuine “cold-start” scenarios.

Figure R8 Linear fitting curves of generalization gap with respect to the size of the similar training set.

We have updated the method description and results in the corresponding section of the revised manuscript (The corresponding revision can be seen Line 485-564 on Page 28-32 of the revised manuscript and Figure S4 in revised Supplementary Information).

13. When modeling molecules with GNNs, it is unclear why edge information (e.g., bond types) is not considered.

Response:

Thank you for your insightful comment. Indeed, the GNN backbone currently implemented in AdaMBind only takes node features and graph topology as input and does not explicitly encode edge information such as chemical bond types. We would like to emphasize that the core contribution of this work is a novel meta-learning framework for few-shot DTA prediction. Our primary research objective is to validate the effectiveness of this framework in enhancing generalization. Accordingly, a relatively simple and stable base learner was chosen for affinity prediction to serve this

focused purpose.

Another key advantage of AdaMBind lies in its model-agnostic design. This flexibility yields two major benefits: (1) it enables seamless integration of more advanced base learners, such as GNNs leveraging edge information (e.g., KANGNN or graph transformers), thereby enhancing molecular representation capabilities; (2) it ensures the framework's adaptability to future novel architectures or feature extraction methods.

In future work, we plan to integrate richer drug-target pair representations (such as edge information, 3D structural information, etc.) and explore the construction of more powerful base learners, to further improve the performance ceiling of AdaMBind. (The corresponding revision can be seen Lines 764-798 on Pages 44-45 of the revised manuscript).

14. Regarding the LIT-PCBA dataset and the associated experimental setup, it is unclear whether the training data consist of only 13 tasks \times 15 active samples. Since virtual screening is a common task, the authors should clarify the performance levels of the reported EF and precision values, and discuss how they compare with existing studies. Additionally, it is unclear why BEDROC ($\alpha = 80.5$), a widely used metric for early recognition in virtual screening, was not considered. Including this metric or discussing its omission would strengthen the evaluation.

Response:

Thank you for raising these insightful points. In response, we first clarify the training data composition used in the LIT-PCBA experiment. Subsequently, we have supplemented the evaluation by including the BEDROC metric ($\alpha = 80.5$) in the revised results, along with reporting the performance of baseline models under the same setup. To contextualize the level of the reported EF and precision values, we surveyed existing virtual screening studies on the LIT-PCBA benchmark. Finally, we provide a comparative analysis of our results relative to prior works, discussing the performance positioning and practical implications of AdaMBind in rigorous few-shot virtual screening scenarios.

(1) Clarification on the composition of the training data

From the perspective of inner-loop task adaptation, the base learner of each meta-training task is updated using only the 15 active samples in its support set. Hence, when the model performs rapid adaptation for a single task, the amount of data employed is indeed 13 training tasks \times 15 active samples per task. This setting is designed to simulate the scenario of extreme data scarcity commonly encountered in real-world drug screening. (The corresponding revision can be seen Lines 594-603 on Page 34 of the revised manuscript).

From the perspective of outer-loop meta-optimization, the update of the meta-learner relies on the performance of each task on its query set. Specifically, after the adaptation on the support set is completed for a meta-training task, we evaluate the performance of the adapted model using samples from that task's query set, and the gradients computed from the losses on all task query sets are used to update the initialization parameters of the meta-learner.

(2) Newly added metric and the comparison with baselines

Results of AdaMBind on LIT-PCBA

To assess the practical utility of AdaMBind in virtual screening tasks, we conducted a rigorous few-shot evaluation on the independent LIT-PCBA dataset. This dataset contains 15 targets with notably low proportions of active compounds (e.g., only 2.30% for the ESR target and 1.91% for the TP53 target), closely reflecting the extreme class imbalance typically encountered in real world drug discovery. We constructed meta-test tasks centered on the ESR and TP53 targets. In each test task, only 15 active compounds were used as the support set, while the remaining compounds formed the query set (including both active and inactive molecules). Meanwhile, AdaMBind was re-trained using the remaining 13 tasks with 15 samples in the support set. This training setup simulates the realistic challenge of identifying active molecules from large compound libraries under limited prior knowledge. The performance of AdaMBind on these two targets is summarized in Table R59.

Table R59 Performance of AdaMBind on LIT-PCBA

	EF@1%	EF@5%	Precision@10	Precision@20	BEDROC(80.5)
TP53	4.1966	3.6875	0.1000	0.1000	9.3461
ESR	4.2080	4.3959	0.1000	0.1500	11.4917

Comparison of AdaMBind with baselines on LIT-PCBA

To compare AdaMBind with baseline models on the LIT-PCBA dataset, all models were also retrained on the LIT-PCBA dataset using the identical data partitioning strategy. The performance of each model was then comprehensively evaluated and compared across multiple key metrics, including EF@1%, EF@5%, Precision@10, Precision@20, and BEDROC($\alpha=80.5$).

As shown in Figure R9, AdaMBind achieved EF@1% and EF@5% values of 4.208 and 4.396, respectively, on the ESR target, and 4.197 and 3.687 on the TP53 target, significantly outperforming all baseline models. These results demonstrate that even under extreme data imbalance, AdaMBind can effectively rank active compounds at the very top of the prediction list. Furthermore, precision-based evaluation further confirms its superiority: on ESR, AdaMBind attained precision@10 and precision@20 of 10% and 15%, respectively; on TP53, both precision@10 and precision@20 reached 10%. However, most baseline models failed to identify any effective compounds when predicting the top 10 or top 20 candidates. To provide a more comprehensive assessment of early recognition capability, we employed the BEDROC metric with $\alpha=80.5$, which emphasizes top-ranking performance. AdaMBind achieved BEDROC scores of 11.4917% for ESR and 9.3461% for TP53, the highest among all compared methods. The robust performance of AdaMBind across multiple metrics demonstrates its ability to reliably prioritize active compounds from highly imbalanced libraries. (The corresponding revision can be seen Fig.8 on Page 33 and Line 622-631 on Page 35-36 of the revised manuscript).

In addition, during the re-verification of the experimental results, we identified that an earlier version of the experimental results from the baseline model had been inadvertently used in plotting. We have now corrected this in the revised manuscript by replotting the figure using the final and accurate experimental results from the baseline model (now presented as Figure R9). This correction pertains solely to the graphical

presentation and does not affect any of the study's conclusions. We sincerely apologize for this oversight.

Figure R9 Performance comparison between AdaMBind and baseline methods on the TP53 (a) and ESR (b) targets for identifying active compounds.

(3) A survey of virtual screening metrics reported by existing studies

To understand how the results achieved by AdaMBind under the dual challenges of limited meta-tasks and constrained support-set sample sizes compare to existing approaches, we conducted a comprehensive literature review.

Zhou et al.⁷ and Lan et al.⁸ compared the performance of multiple deep learning methods on the LIT-PCBA dataset. **Table R60** presents their results on three key metrics: BEDROC ($\alpha=80.5$), EF@1%, and EF@5%. Among them, ligand-based virtual screening (LBVS) methods such as ROCS, Phase Shape, LIGSIFT, SHAFTS, and S-MolSearch primarily rely on molecular 3D shape or fingerprint similarity for search, with molecular 3D structural information as input. In contrast, structure-based virtual screening (SBVS) methods such as Glide-SP, Surflex, Gnina, SaBAN and DrugCLIP typically require 3D structural information or binding site data of proteins, with some approaches (e.g., BigBind, DeepDTA) also depending on binding affinity labels for training.

During their evaluation, none of the models were trained on any data from LIT-PCBA. Instead, they were either pre-trained or directly inferred using external data sources such as ChEMBL and public molecular databases.

Table R60 Performance on the LIT-PCBA Dataset as Reported by Zhou and Lan

Methods	BEDROC (%)	EF @1%	EF @5%
ROCS (LBVS)	–	2.48	–
Phase Shape (LBVS)	–	2.98	–
LIGSIFT (LBVS)	–	2.39	–
SHAFTS (LBVS)	–	2.79	–
Surflex (SBVS)	–	2.50	–
Glide-SP (SBVS)	4.00	3.41	2.01
Planet (SBVS)	–	3.87	2.43
Gnina (SBVS)	5.40	4.63	–
DeepDTA (SBVS)	2.53	1.47	–
BigBind (SBVS)	–	3.82	–
DrugCLIP (SBVS)	6.23	5.51	2.27
S-MolSearch _{0.4} (LBVS)	7.58	6.28	2.47
S-MolSearch _{0.9} (LBVS)	8.48	7.36	3.21
SaBAN(SBVS)	13.35	6.33	3.36

Meanwhile, Connor J. Morris et al.⁹ also compared MILCDock with several molecular docking tools and machine learning methods. MILCDock is a classification model that integrates output features from multiple traditional docking tools (including Vina, rDock, Plants, LeDock, and AutoDock4) to construct a multi-layer perceptron model aimed at improving the ranking capability in virtual screening. This method does not directly utilize the raw structural information of proteins or small molecules; instead, it extracts energy scores and inter-conformational RMSD values generated by each docking tool as input features. The baseline models for comparison include individual docking tools, a score-based naive consensus method, and the classical machine learning model XGBoost. Morris et al. partitioned the protein targets in the LIT-PCBA

dataset into training, validation, and test sets, ensuring substantial diversity among targets across different splits. The test set comprises four targets: GBA, ALDH1, IDH1, and ADRB2. The comparative performance of these methods is summarized in two key tables: Table R61 presents the enrichment factor at 1% (EF@1%), while Table R62 details the corresponding BEDROC ($\alpha=80.5$) scores.

Table R61 EF@1% on the LIT-PCBA Dataset as Reported by Morris

Targets	Vina	rDock	Plants	LeDock	AutoDock4	Naïve Consensus	XGBoost	MLP	Ensemble MLP
ADRB2	0.00	0.00	0.00	0.00	0.00	5.88	0.00	11.77	5.88
ALDH1	1.73	1.70	1.66	1.62	2.11	1.97	0.56	0.95	0.97
GBA	4.91	3.07	4.91	1.84	5.52	1.84	4.30	4.91	7.98
IDH1	2.56	2.56	2.56	5.13	0.00	0.00	2.56	5.13	2.56
MEAN	2.30	1.83	2.28	2.15	1.91	2.42	1.85	5.69	4.35
STD	2.04	1.35	2.05	2.15	2.61	2.48	1.96	4.48	3.17

Table R62 BEDROC($\alpha=80.5$) on the LIT-PCBA Dataset as Reported by Morris

Targets	Vina	rDock	Plants	LeDock	AutoDock	Naïve Consensus	XGBoost	MLP	Ensemble MLP
ADRB2	0.000	0.000	0.000	0.000	0.000	0.027	0.000	0.091	0.057
ALDH1	0.094	0.088	0.087	0.084	0.113	0.100	0.038	0.047	0.049
GBA	0.029	0.018	0.032	0.006	0.038	0.016	0.035	0.042	0.047
IDH1	0.025	0.009	0.007	0.026	0.000	0.000	0.015	0.029	0.008
MEAN	0.037	0.029	0.031	0.029	0.038	0.036	0.022	0.052	0.040
STD	0.040	0.040	0.039	0.038	0.053	0.044	0.018	0.027	0.022

(4) Extend the analysis of the performance level of our reported results

In virtual screening, the LIT-PCBA dataset is recognized as a challenging

benchmark task due to its highly imbalanced ratio of active to inactive compounds (with some targets having less than 2% active compounds) and its origin from experimentally validated data. Compared to biased datasets such as DUD-E, which are artificially constructed, LIT-PCBA better reflects the data distribution in real-world drug discovery scenarios, thus achieving good performance on this dataset carries stronger practical significance.

In this comparative analysis, we first examined representative methods that have demonstrated excellent performance on LIT-PCBA in recent years. As shown in Table R11, both ligand-based virtual screening methods (e.g., S-MolSearch) and structure-based virtual screening methods (e.g., DrugCLIP, SaBAN) mostly rely on large-scale pretraining or external labeled data. For instance, S-MolSearch utilized approximately 600,000 protein-molecule pairs from ChEMBL, along with about 19 million unlabeled molecules for semi-supervised contrastive learning training. SBVS methods such as DrugCLIP typically require 3D structural information or binding site data of proteins. These methods exhibit promising screening capabilities under zero-shot or pretrained evaluation settings, particularly showing significant improvements in early enrichment metrics such as BEDROC and EF@1%.

In contrast to these methods that depend on large-scale labeled data or structural information, AdaMBind uses only a very small number of support samples (only 15 active compounds per target) and still demonstrates competitive early enrichment ability on targets such as TP53 and ESR (EF@1% = 4.20 and 4.21, BEDROC = 9.34% and 11.49% respectively). Although the values are lower than those of pretrained methods such as S-MolSearch^{0.9} (EF@1% = 7.36, BEDROC=8.48%) or SaBAN (EF@1% = 6.33, 13.35%), the strength of AdaMBind lies in its extremely low data dependency and powerful few-shot generalization capability. We did not use hundreds of thousands of affinity data points from ChEMBL, nor did we rely on protein 3D structures or large-scale unlabeled molecular libraries. Instead, through a meta-learning framework, AdaMBind rapidly adapts to new targets with minimal support samples, significantly reducing the cost of data acquisition and annotation.

Further comparison with machine learning methods based on ensemble docking,

such as MILCDock (Table R61-Table R62), shows that AdaMBind's EF@1% and BEDROC performance on TP53 and ESR targets exceeds the average EF@1% and BEDROC of MILCDock (MLP mean EF@1% = 5.69, BEDROC=5.2%; Ensemble MLP EF@1% = 4.35, BEDROC=4.0%). Moreover, its BEDROC metric indicates good stability in early recognition. It is particularly noteworthy that MILCDock still requires output features from multiple docking tools for training, and its performance on LIT-PCBA fluctuates considerably (e.g., EF@1% = 0 on the ADRB2 target).

In summary, LIT-PCBA serves as a difficult and realistic virtual screening benchmark that imposes higher demands on model generalizability and data efficiency. Although AdaMBind's absolute performance is slightly inferior to some methods that rely on large-scale pretraining or structural information, it still achieves effective active compound identification with extremely limited labeled samples (only 15 active molecules per task). This highlights its notable data efficiency advantage and potential for cold-target adaptation, offering a practical and valuable solution for virtual screening in data-scarce scenarios. We have added the above analyses to the revised manuscript. (The corresponding revision can be seen Line 632-646 on Page 36-37 of the revised manuscript)

15. I suggest providing the predicted affinity values for all compounds listed in Table 1. It is also unclear why experimental validation was not performed for the other compounds. A more detailed analysis of the binding modes would help interpret the predictions and strengthen the study.

Response:

Thank you for your valuable feedback. We explained why experimental validation was not pursued for the remaining compounds. Ultimately, we analyze the predicted binding modes. Below are our specific responses and clarifications regarding the 3 points you raised:

(1) Supplementary data for the predicted affinities in Table 1

We included the predicted affinity (pIC₅₀) values for all 20 compounds in Table 1 of revised manuscript (Table R63).(The corresponding revision can be seen Table1 on

Table R63 Top 20 drugs ranked by predicted inhibitory potency against FLT3

Rank	Drug	CID	Literature Validation	Docking Simulation	Predicted Affinity (pIC ₅₀)
1	Indolocarbazole Nitrogen Derivative	44299148			13.2539
2	Staurosporine	44259		√	12.7778
3	Midostaurin	9829523	Stone R M et al.		12.5083
4	K-252a	3035817			12.2893
5	Vinblastine	13342		√	11.8991
6	Ingenol Mebutate	6918670			11.5905
7	Picrotoxin	31304		√	11.5486
8	SCHEMBL1649555	133005			11.4881
9	CHEMBL282318	104826			11.4813
10	Tiotropium	5487427			11.3839
11	Vinorelbine	44424639	Ramaswamy K et al.		11.3628
12	Omadacycline	54697325		√	11.3247
13	Fluticasone Furoate	9854489			11.2953
14	SCHEMBL21067566	444853			11.2512
15	Paclitaxel	36314			11.2391
16	Vincristine	5978		√	11.1512
17	4-hydroxybenzoyl-CoA	168718			11.1289
18	Recinnamine	32681		√	11.1275
19	Atracurium Besylate	47320			11.1261
20	Navitoclax	24978538	Kivioja J L et al.		11.1082

(2) The experimental validation of other compounds

For the validation experiments of FLT3 inhibitors, we adopted a multi-step

strategy as follows:(1) Affinity prediction and ranking based on the AdaMBind ;(2) Molecular docking simulations for preliminary validation ;(3) Literature review for supporting evidence ;(4) wet-lab validation of the most promising candidates.

Staurosporine was selected as the primary compound for in-depth experimental validation based on the following systematic considerations. First, Staurosporine ranked second in the model prediction with high predicted affinity ($pIC_{50} = 12.78$), indicating strong potential inhibitory activity. Second, we performed molecular docking simulations to verify the binding modes of top-ranked candidates. The analysis revealed that although certain compounds (e.g., Indolocarbazole Nitrogen derivative) exhibited higher predicted affinity values, their docked conformations with FLT3 showed considerable instability and yielded poor interaction energies, suggesting that their actual binding might be weak or unreliable. Subsequently, we systematically compared the docking poses with the reported binding modes of FLT3 inhibitors in the literature. The results indicated that the docked conformation of Staurosporine shares partial characteristics with Type I inhibitors. Its molecular scaffold formed a stable hydrogen bond network and hydrophobic interactions within the ATP-binding pocket, providing a reasonable structural basis for the predicted high affinity. In contrast, although other high-scoring compounds displayed some binding propensity in docking, their binding modes lacked clear support in the literature, and their interaction strength and stability were inferior to those of Staurosporine. Finally, considering the substantial resource, time, and cost investment required for wet-lab experiments, we prioritized Staurosporine as a lead compound for in-depth functional validation, owing to its high prediction confidence, well-defined structure, and representative binding mode.

(3) Analysis of the binding modes

To further elucidate the binding mode of Staurosporine to the FLT3 kinase as predicted by AdaMBind, we conducted a systematic structural analysis through molecular docking simulations. **Figure R10** demonstrates that Staurosporine stably binds to the ATP-binding pocket of the FLT3 kinase. Its molecular scaffold engages in extensive hydrophobic packing interactions with residues such as Leu818 and Val624. Simultaneously, key hydrogen-bond networks are formed between the carbonyl oxygen

of Staurosporine and the backbone nitrogen of Cys694, as well as between its amino group and the side-chain carboxyl of Asp698, with distances of 2.9 Å and 3.1 Å, respectively. Furthermore, the indole moiety of Staurosporine establishes CH- π interactions with Tyr693, reflecting its favorable conformational adaptability.

According to systematic reviews on the mechanisms of FLT3 inhibitors in the literature, FLT3 inhibitors are primarily classified into three binding modes: Type I inhibitors bind to the ATP pocket in the active kinase conformation; Type II inhibitors bind to the inactive conformation and occupy the hydrophobic pocket formed by the activation loop; covalent inhibitors achieve irreversible inhibition by forming covalent bonds with Cys695¹⁰. These three classes exhibit distinct differences in binding conformation, interaction sites, and resistance profiles.

Comparative analysis reveals that the binding characteristics of Staurosporine exhibit a complex pattern. In certain respects, its binding mode resembles that of Type I FLT3 inhibitors, as it localizes within the ATP-binding pocket without deeply occupying the hydrophobic cavity typically targeted by Type II inhibitors. The hydrogen-bond network formed with the conserved catalytic residues Cys694 and Asp698 aligns with interaction patterns observed in some kinase inhibitors targeting the active conformation. Furthermore, the absence of strong direct interactions with the gatekeeper residue Phe691 distinguishes it from many conventional FLT3 inhibitors, whose efficacy is often compromised by the F691L mutation. This interaction profile, centered on the ATP-binding pocket while displaying unique features, provides a structural rationale for the high affinity predicted by AdaMBind and accounts for the potent inhibitory activity observed in our experimental studies against both wild-type and mutant forms of FLT3. (The corresponding revision can be seen on Lines 691-730 on Page 40-42 of the revised manuscript)

Figure R10 2D Interaction Diagram of FLT3 and Staurosporine.

Minor

1. In line 32, “IC50” is incorrectly written as “IC₅₀”. This typo appears throughout the manuscript, and the authors should correct all instances to ensure consistency and accuracy.

Response:

Thank you for your careful review and this comment. We corrected the typo “IC₅₀” to “IC₅₀” throughout the manuscript, as suggested. We have also double-checked the document to ensure all similar biochemical terms are presented correctly and consistently.

2. In line 205, it is stated that “In the testing phase, the trained meta-learner is used to predict drug–target binding affinities in the majority setting (40 known DTA pairs) or few-shot setting (5 known DTA pairs).” This description seems inaccurate. Shouldn’t the meta-learner be fine-tuned on the support set during testing, as is standard in meta-learning? Clarification on this point is needed.

Response:

Thank you for your comment. The phrase used in the original manuscript was an oversimplification that omitted the essential adaptation step. We revised the manuscript (Lines 212-218 on Page 11, Lines 226-229 on Page 12 of the revised manuscript) to

explicitly describe this two-step process: In the testing phase, the meta-learner is first fine-tuned on the support set of the test task in either the majority setting (40 drug-target pairs) or the few-shot setting (5 drug-target pairs), and then used to predict drug-target binding affinities.

3. In line 608, the manuscript mentions “additional drugs whose affinities are unknown.” However, the samples in the query set appear to have known labels. The authors should clarify this apparent inconsistency.

Response:

We sincerely thank you for this insightful comment. Indeed, the labels of the query set are known and used during the outer loop of the meta-training process. Our previous description of “affinities are unknown” was intended from the perspective of the inner loop, emphasizing that these labels are not exposed to the model during the inner-loop adaptation to align with the realistic evaluation setting of few-shot learning. However, during the outer-loop meta-optimization stage, these labels serve as supervisory signals for updating the parameters of the meta-learner¹¹.

To eliminate any confusion, we revised the manuscript by removing the imprecise statements (The corresponding revision can be seen in Line 835-843 on Page 47 of the revised manuscript). In the revised manuscript, the relevant description has been revised to: “In this study, we frame the DTA prediction problem within a protein-anchored meta-learning paradigm. We assume a distribution of tasks $p(T)$, where each task T_t corresponds to a unique target protein t and comprises all available drug associated with that target. Each task is further divided into a support set S_t and a query set Q_t . The support set $S_t = \{(t, d_i, y_i)\}_{i=1}^n$ contains pairs of the target t and a small number of drugs d_i with affinities y_i , while the query set $Q_t = \{(t, d_j, y_j)\}_{j=1}^m$ includes the same target t with additional drugs d_j with affinities y_j . In this way, the full set of tasks $p(T)$ can be partitioned into meta-training set $\{T_k\}_{k=1}^{N_{tr}}$, meta-validation set $\{T_k\}_{k=1}^{N_{val}}$, and meta-testing

$\text{set}\{T_k\}_{k=1}^{N_{\text{test}}}$.”

4. In Figure 1, “combanation” is a typo and should be corrected to “combination.”

Response:

Thank you for pointing out this typo. The error has been corrected, and “combanation” has been replaced with “combination” in Figure 1 of the revised manuscript. The revised Fig. 1 is presented in Figure R11.

Figure R11 The architecture of AdaMBind model

5. In line 317, “Meta-Learnng” is a typo and should be corrected to “Meta-Learning.”

Response:

Thank you for catching this typo. The word “Meta-Learnng” in line 317 has been corrected to “Meta-Learning” in the revised manuscript.

6. In line 155, “Li dataset” should be corrected to “Li’s dataset” to indicate proper possession.

Response:

Thank you for this grammatical correction. As suggested, 'Li dataset' has been revised to 'Li's dataset'.

7. Throughout the manuscript, “Lit-PCBA” should be corrected to “LIT-PCBA” to maintain consistent capitalization of the dataset name.

Response:

Thank you for highlighting this inconsistency. We have corrected "Lit-PCBA" to "LIT-PCBA" throughout the entire manuscript to adhere to the standard capitalization of the dataset name.

8. In line 668, “Huaxiu Yao et al.” should be corrected to “Yao et al.”

Response:

Thank you for the correction. We have changed the citation in line 668 from "Huaxiu Yao et al." to "Yao et al."

9. In line 713, it is mentioned that “we converted the original Kd and IC50 values into their logarithmic forms (pKd and pIC50).” The authors should also including Ki values.

Response:

Thank you for this suggestion. We agree that including Ki values is important for consistency. The sentence in line 713 (and the corresponding method description) has been revised to state that we converted "Kd, Ki, and IC50 values into their logarithmic forms (pKd, pKi, and pIC50)."

10. In line 412, it is mentioned “and Davis datasets (Supplementary Fig. 2).” However, other supplementary information is not referenced in the main text, and there appears to be no Supplementary Fig. 1. The authors should ensure that all supplementary figures are properly cited and included.

Response:

We thank the reviewer for highlighting this inconsistency in the referencing of supplementary figures. We have carefully reviewed the entire manuscript and the supplementary information. The citation has been corrected to "Supplementary Fig. 1" in line 412. Furthermore, we have ensured that all supplementary figures are now sequentially numbered and appropriately cited in the main text.

11. In line 727, “coefficient of determination (r^2)” should be written as “ R^2 ” to follow standard notation.

Response:

Thank you for this correction. We have revised “ r^2 ” to the standard notation “ R^2 ” in line 727 and throughout the manuscript where applicable to maintain consistency with statistical conventions.

12. In line 762, it is mentioned that “During meta-training, the outer loop was run for 10 episodes, and each episode included 5 epochs of training in the inner loop.” This number of training iterations seems relatively small. The authors should clarify whether this is sufficient and, if possible, provide training curves to support the choice of training schedule.

Response:

Thank you for your comment. In the meta-learning training framework, due to the memory constraints of the computational hardware, we cannot simultaneously load all training tasks (i.e., the data subsets corresponding to different protein targets). To address this, within each training epoch, we repeatedly sample a fixed-sized batch of tasks (containing B independent tasks) from the overall task pool. Based on this task batch, we perform inner-loop adaptation and outer-loop meta-gradient computation, followed by an update to the meta-parameters. This sampling and updating procedure is repeated 100 times per epoch, meaning that each epoch consists of 100 iterations of random task sampling and parameter updates. Through this intensive iterative training with a large number of task batches, the model sufficiently learns robust meta-knowledge from the task distribution. This approach of meta-training via task batches

has been widely adopted in the field of meta-learning, and its effectiveness has been validated in numerous studies^{12,13}. (The corresponding revision can be seen in Line 891-894 on Page 50, and Line 1056-1063 on Page 58 of the revised manuscript)

Figure R12 illustrates the training and validation loss curves of AdaMBind on the BindingDB dataset (majority setting).

Figure R12 Training Loss Curve of AdaMBind.

Reference

- 1 Bai, Y., Chen, M., Zhou, P. et al. How important is the train-validation split in meta-learning? International Conference on Machine Learning. 543–553 (2021).
- 2 Qi, X., Zhao, L., Tian, C. et al. Predicting transcriptional responses to novel chemical perturbations using deep generative model for drug discovery. Nat Commun 15, 9256 (2024).
- 3 Greff, K., Srivastava, R. K., Koutník, J. et al. LSTM: A search space odyssey. IEEE Transactions on Neural Networks and Learning Systems 28, 2222–2232 (2016).
- 4 Wang, Y., Xia, Y., Yan, J. et al. ZeroBind: a protein-specific zero-shot predictor with subgraph matching for drug-target interactions. Nat Commun 14, 7861 (2023).
- 5 Gil-Sorribes, M. & Molina, A. Tensor-DTI: Enhancing Biomolecular Interaction Prediction with Contrastive Embedding Learning. Learning Meaningful

- Representations of Life Workshop at ICLR 2025. (2025).
- 6 Zhou, G., Gao, Z., Ding, Q. et al. Uni-mol: A universal 3d molecular representation learning framework. ChemRxiv. (2023).
 - 7 Zhou, G., Wang, Z., Yu, F. et al. S-molsearch: 3d semi-supervised contrastive learning for bioactive molecule search. Advances in Neural Information Processing Systems 37, 74715–74737 (2024).
 - 8 Lan, J., Ding, H., Chen, H. et al. Structure-Aware Contrastive Learning with Fine-Grained Binding Representations for Drug Discovery. arXiv preprint arXiv:2509.14788 (2025).
 - 9 Morris, C. J., Stern, J. A., Stark, B. et al. MILCDock: machine learning enhanced consensus docking for virtual screening in drug discovery. Journal of Chemical Information and Modeling 62, 5342–5350 (2022).
 - 10 Friedman, R. The molecular mechanisms behind activation of FLT3 in acute myeloid leukemia and resistance to therapy by selective inhibitors. Biochimica et Biophysica Acta (BBA) - Reviews on Cancer 1877, 188666 (2022).
 - 11 Lee, K., Maji, S., Ravichandran, A. et al. Meta-learning with differentiable convex optimization. Proceedings of the IEEE/CVF Conference on Computer Vision and Pattern Recognition. 10657–10665 (2019).
 - 12 Wen, G. & Li, L. MMOSurv: meta-learning for few-shot survival analysis with multi-omics data. Bioinformatics 41, btae684 (2025).
 - 13 Gao, J., Lyu, S., Liu, G. et al. A hybrid model for few-shot text classification using transfer and meta-learning. International Conference on Advanced Algorithms and Control Engineering. 2179–2183 (2025).
- .

Reviewer #2:

This manuscript presents AdaMBind, an innovative drug-target affinity prediction model that integrates the MAML meta-learning framework with a task adaptation mechanism. By adopting an “easy-to-hard” learning strategy, AdaMBind effectively enhances model performance under low-data conditions. The authors conduct comprehensive evaluations across three benchmark datasets, under both majority- and few-shot settings, demonstrating consistent superiority over existing baseline methods. Notably, the study extends beyond computational validation by applying AdaMBind to identify candidate FLT3 inhibitors, culminating in experimental confirmation that Staurosporine exhibits potent inhibitory activity against FLT3 with a significantly improved IC_{50} compared to the positive control. This real-world application underscores the practical utility of the proposed method. Overall, the integration of meta-learning with adaptive task selection is both innovative and well-motivated. The experimental validation is thorough, and the manuscript is clearly written. However, the authors should consider the following comments.

Thank you for reviewing in our manuscript. We carefully revised the manuscript according to your comments. All of your suggestions are helpful to improve our work and make the results more convincing. According to your suggestions, we mainly answer your comments and revise the manuscript from five aspects (**Parts 1-5** below):

1) Incorporation of the latest literature. In the ‘*Introduction*’ section, we included the recent advances from 2023 to 2025, focusing on advances in the areas of meta-learning and drug-target affinity prediction. [**Comment 8**] (Page 38-,Line 56-149)

2) Expanded Description of Experimental Implementation and Methodological Design.

To enhance the clarity, reproducibility, and rationale of our methodological choices, we provided extensive additional details.

a. Hyperparameter Settings: We provided a detailed table specifying the optimal hyperparameter configurations for all experiments to ensure full transparency and reproducibility [**Comment 1-2**]. (Figure S2 and Table S9 in revised Supplementary

Information)

b. Ablation Study on Noise Robustness: To address the concern regarding the noise level used during training, we conducted an ablation study to test the model's performance under different noise levels and distributions. **[Comment 3]**. (Figure S3 in revised Supplementary Information)

3) Refinement of the Discussion. In the *Discussion* section, we clarified the limitations of the model when applied to conformationally flexible protein targets, and briefly outlined future directions for improvement. **[Comment 4]** (Page 43-45 , Line 759-798)

4) Revision and addition of figures

a. Redraw of Figure 4a. We presented the standard deviation across five independent runs in Figure 4a. **[Comment 5]**

b. Visualization of FLT3 structure. We presented the FLT3 structure employed in the docking simulation and the PDB ID. **[Comment 6]** (Fig.9c in revised manuscript)

5) Made corrections of inaccurate expressions, grammar, and representations of figures and tables. **[Comments 7,9-10]**

Below, we provide a point-by-point response to each suggestion. All modifications in the revised manuscript are highlighted in red text.

Points

1. The manuscript needs more details on hyperparameter optimization. The authors should clarify their hyperparameter selection process and training procedures, especially regarding how they optimized the parameters of different components of AdaMBind. This will clarify the effects of hyperparameter tuning on model performance across various datasets.

Response:

Thank you for your valuable comments on our manuscript. We provided detailed additions and explanations on the missing details of hyperparameter optimization. For the key hyperparameters of each major component (e.g., noise, outer learning rate, inner learning rate, Batch size in support set, number of candidate tasks B , Number of

selected tasks N_{slect} , etc.), we adopted a grid search approach to explore different hyperparameter combinations. First, we conducted a series of experiments to jointly evaluate the influence of the inner learning rate and the batch size in the support set on model performance. Next, Second, we systematically explored the combined effect of the outer learning rate and the number of candidate tasks B . Then, we examined how the number of selected tasks N_{slect} impacts the model performance. Finally, we explored the effect of noise on model performance. As shown in **Table R1**, the final selected hyperparameter combination demonstrates strong performance in terms of MSE.

Table R1 The processes of AdaMBind parameter tuning

Hyper parameters	range
noise	0.1, 0.2, 0.3, 0.4, 0.5 , 0.6
Inner learning rate	1e-3, 5e-4, 1e-4 , 5e-5, 1e-5
Outer learning rate	5e-4, 1e-4, 5e-5, 1e-5
Number of candidate tasks B	12, 13, 14, 15 , 16
Number of selected tasks N_{slect}	4, 6, 8 , 10
Batch size in the support set	2, 4, 8 , 16

The results of the hyperparameter experiments are presented in Figure R1. (corresponding to Supplementary Figure S9 and Table S2 of the revised Supplementary Information)

Figure R1 Impact of hyperparameter on MSE

2. As meta-learning frameworks are highly sensitive to hyperparameter settings, the authors are strongly encouraged to include a comprehensive list of all key hyperparameters in the Supplementary Materials. Moreover, more statements or illustrations should be added to clarify why the meta-learning framework helps the prediction.

Response:

We sincerely thank you for these valuable suggestions. We provided detailed additions and explanations on the missing details of hyperparameter optimization. In addition, we provided a more detailed explanation of how the meta-learning framework aids prediction.

(1) Hyperparameter transparency

We agree with the reviewer on the importance of detailed hyperparameter reporting. In response, we have included a comprehensive hyperparameter table in Table R1 (Corresponding to Table S9 and Figure S2 of the Supplementary Information), which lists all key hyperparameters used in AdaMBind, including those related to the meta-learning framework, the adaptive task module, and the noise injection strategy.

(2) Statement of the meta-learning framework

To clarify why the meta-learning framework is particularly suitable for DTA prediction under data-scarce conditions, we have added explanatory statements in the *Introduction* (Line 109-117 on Page 6 of revised manuscript).

The expanded statement in the revised manuscript is shown below: “Meta-learning, also referred to as “learning to learn”, has emerged as a promising framework to address these challenges. Unlike conventional training that merely fits data distributions of specific tasks, meta-learning enables models to acquire shared and transferable “meta knowledge” across diverse tasks. Consequently, when encountering a novel target with extremely limited data, the model can leverage the previously learned meta-knowledge to rapidly fine-tune and adapt using only a few samples, thereby enabling effective prediction of affinities for new targets.”

3. Label noise injection is mentioned in the method section as a means to enhance robustness, but the potential biases it may introduce are not discussed. It is recommended to incorporate an analysis of the impact of noise levels in the ablation studies.

Response:

Thank you for your valuable comment. We have conducted a systematic ablation study on this key hyperparameter to further validate and optimize the regularization strategy of our model.

We performed an experimental evaluation across three benchmark datasets (Davis, BindingDB, and KIBA), testing noise levels in the range of [0.1, 0.6] at intervals of 0.1. **Figure R2** shows that both the BindingDB and KIBA datasets achieved the lowest validation mean squared error (MSE) at a noise level of 0.5, indicating that this setting provides the optimal regularization effect for these two datasets. For the Davis dataset, the best performance was observed at a noise level of 0.2. However, it is worth noting that when the noise level was set to 0.5, the performance (MSE = 0.2490) differed only marginally from the optimal value (MSE = 0.2470). Moreover, the model trained with a noise level of 0.5 exhibited lower performance variability across repeated experiments

compared to that trained with a noise level of 0.2. (Corresponding to Figure S3 of the revised Supplementary Information)

Based on the above analysis, we ultimately decided to uniformly adopt a noise level of 0.5 in our model.

Figure R2 Impact of Noise on MSE

4. Although AdaMBind does not require 3D structural information, which is advantageous for targets with unknown or poorly resolved structures, it may encounter difficulties when applied to highly flexible proteins, such as GPCRs or ion channels, where conformational dynamics play a crucial role in ligand binding. The authors are encouraged to discuss these limitations and boundary conditions in the “Discussion” section. Additionally, they could briefly outline potential future directions, such as incorporating conformational ensembles, molecular dynamics features, or uncertainty estimation, to position the work within the evolving landscape of DTA prediction.

Response:

We sincerely appreciate your insightful suggestions regarding the discussion of AdaMBind’s limitations and potential improvements. We revised and refined the *Discussion* section in our revised manuscript. (Line 764-798 on Page 44-45)

The expanded discussion in the revised manuscript is shown below:

“Second, the current model design does not depend on three-dimensional structural information, which makes it applicable to targets with incomplete structural resolution, but may also limit its predictive capability for highly flexible proteins such

as GPCRs and ion channels. The binding affinity of such targets often involves dynamic processes like “induced fit”, which may not be fully captured by the static feature fusion mechanism based solely on sequence.

To address these limitations and further advance DTA prediction, future work can proceed along the following directions. At the feature representation level, dynamic structural information derived from molecular dynamics simulations or conformational sampling could be integrated to more accurately characterize the dynamics of drug-target interactions. Simultaneously, enhanced graph neural networks capable of explicitly modeling edge information, such as chemical bonds, could be introduced to improve the fine-grained expressiveness of drug representations. At the algorithmic framework level, more advanced pre-trained protein encoders that incorporate three-dimensional structures can be integrated with uncertainty quantification mechanisms. This integration would enhance both the reliability and interpretability of predictions. Furthermore, thanks to the flexible and model-agnostic design of the AdaMBind framework itself, more powerful base learners for DTA prediction can be conveniently integrated in the future, thereby continuously raising the performance ceiling and generalization capability of the model.”

5. Figure 4a presents the ablation study results, but does not include error bars or statistical measures. The authors should report variance metrics (e.g., standard deviation over five independent runs) to support the reliability of the observed performance differences.

Response:

Thank you for your comment. We included error bars representing the standard deviation across five independent runs in Fig.5 in the revised manuscript. The updated figure is presented as **Figure R3**. (Corresponding to Fig.5 of the revised manuscript)

Figure R3 Ablation study on BindingDB dataset

6. In the context of the FLT3 inhibitor validation, the PDB ID of the FLT3 structure employed in the docking simulation should be explicitly stated.

Response:

Thank you for the suggestions regarding image optimization. We stated the FLT3 structure (PDB ID: 6JQR) used in the docking simulation in the revised manuscript. (Corresponding to Fig.9c of the revised manuscript)

Figure R4 Visualization for structure of FLT3 in docking simulation (PDB ID:6JQR)

7. The manuscript employs Enrichment Factor (EF@1%, EF@5%) as a key metric in the validation of ESR/TP53, but it fails to define the EF.

Response:

Thank you for the suggestions. We have now incorporated clear definitions of the relevant metrics in the *Methods* section of the revised manuscript. (Lines 957-997 on

8. To enhance the scholarly relevance and positioning of this work, the authors are advised to update the reference list to include recent advances from 2023 to 2025, particularly in the areas of meta-learning for biomedicine and DTA modeling.

Response:

We sincerely appreciate your valuable suggestions for improving our research. In the revised manuscript, we have updated eight references and expanded the discussion on recent advances in DTA prediction and meta-learning approaches in the Introduction section.

The new additions to the literature are as follows, corresponding to the citations above:

1. Vettoruzzo, A., Bouguelia, M. R., Vanschoren, J. et al. Advances and challenges in meta-learning: A technical review. *IEEE Transactions on Pattern Analysis and Machine Intelligence* 46, 4763–4779 (2024).
2. Xu, Y., Fan, Y., Bao, Y. et al. Task-aware meta-learning paradigm for universal structural damage segmentation using limited images. *Engineering Structures* 284, 115917 (2023).
3. Kumar, R., Deleu, T. & Bengio, Y. The effect of diversity in meta-learning. *Proceedings of the AAAI Conference on Artificial Intelligence* 37, 8396–8404 (2023).
4. Zheng, H., Shen, L., Tang, A. et al. Learning from models beyond fine-tuning. *Nature Machine Intelligence* 7, 6–17 (2025).
5. Zhang, L., Liu, Z., Zhang, W. et al. Style uncertainty based self-paced meta learning for generalizable person re-identification. *IEEE Transactions on Image Processing* 32, 2107–2119 (2023).
6. Wang, Y., Xia, Y., Yan, J. et al. ZeroBind: a protein-specific zero-shot predictor with subgraph matching for drug-target interactions. *Nature Communications* 14, 7861 (2023).
7. Koh, H. Y., Nguyen, A. T. N., Pan, S. et al. Physicochemical graph neural network

for learning protein–ligand interaction fingerprints from sequence data. *Nature Machine Intelligence* 6, 673–687 (2024).

9. Two instances of the word “Table” appear on line 500.

Response:

Thank you for pointing out this error. We have corrected the duplicate "Table" to "Table 1" in the revised manuscript.

10. Several grammatical errors were observed (e.g., inconsistent tenses in the abstract). A thorough proofreading is recommended.

Response:

Thank you for your careful review. We have thoroughly proofread the entire manuscript and corrected all identified grammatical errors, including tense inconsistencies, to ensure clarity and adherence to academic writing standards.

Reviewer #2 (Remarks on code availability):

1. The authors provided comprehensive code accompanied by well-documented instructions, which offers readers a good opportunity for reproducibility of their work. In addition, I suggest that the authors provide case codes showing how to apply AdaMBind in the Table of Contents.

Response:

We thank the reviewer for the code and documentation. We have now added a Jupyter notebook to the code repository. This notebook provides a step-by-step demonstration of the complete workflow, including data preprocessing, model training, and evaluation, thereby further enhancing the reproducibility and accessibility of our method. The code can be accessed at <https://github.com/Moohyun-w/AdaMBind>

Reviewer #3:

The manuscript by Wan et al, provides a tool “AdaMbind” for accurate and robust drug target affinity prediction. Authors have developed a new tool for DTA prediction to address the limitations of the current methods. Two core modules were developed – meta learning and task adaptive. Authors evaluated it under majority and few shot learning setting, using the two settings allows for detection of drug-target patterns and drug target binding pairs. ADAMBind outperformed as compared to existing models. Further testing in AML from drug bank identified staurosporine as FLT3 inhibitor which may hold relevance if significant beyond the kinase activity assay.

Response:

Thank you for reviewing our manuscript. We have carefully revised the manuscript according to your comments. All of your suggestions are helpful to improve our work and make the results more convincing. Below, we provide a point-by-point response to each suggestion. All modifications in the revised manuscript are highlighted in red text.

Points

1. The model though outperforms current models still has dependency on the known structures or binding affinities. This should be elaborated more.

We sincerely thank you for raising this important point regarding the model’s dependency on known structures or binding affinities. It is acknowledged that AdaMBind does not operate completely independently of known affinity data or protein sequences. Achieving such independence, a goal closely associated with pure zero-shot learning, remains a recognized challenge in the research field. AdaMBind is designed to address the practical and critical bottleneck of accurate prediction for novel targets with only minimal prior data (i.e., few-shot learning), which is highly relevant in early-stage drug discovery. We will elaborate on this from four perspectives: (1) The design innovation of the AdaMBind architecture, (2) The practical advantages of AdaMBind in real-world applications, (3) The potential inherent in the AdaMBind framework, and (4) Prospects and improvements toward enabling zero-shot learning.

(1) The design innovation of the AdaMBind architecture

In response to your insightful observation that “the model still relies on known data”, we would first like to clarify: the design intent of AdaMBind is not to achieve absolute zero-shot prediction, **but rather aims to address a more common and highly challenging scenario in drug discovery: enabling rapid and accurate learning and prediction for novel targets (new tasks) with only a minimal amount of known data (few-shot)**. As a result, AdaMBind employs a meta-learning algorithm that integrates an adaptive task mechanism. The core advantage of this architecture lies in its adaptive task sampling module, which dynamically evaluates task value, combined with the bi-level optimization mechanism of the meta-learning framework. This enables the model to extract transferable generalization knowledge from limited historical tasks and rapidly adapt to novel targets with only a small number of support samples, thereby maintaining robust and accurate predictions while reducing data dependency:

The meta-learning module, based on Model-Agnostic Meta-Learning (MAML) framework, **achieves rapid adaptation through a bi-level optimization architecture comprising an outer loop (meta-learner) and an inner loop (base-learner)**. The outer loop learns a set of optimal initialization parameters by distilling universal interaction patterns and generalization knowledge from diverse tasks, each corresponding to a distinct protein target along with its associated drugs and affinities. When encountering a novel target, the inner loop utilizes these pre-trained parameters as a foundation and rapidly adapts to the specific task through a few gradient descent steps, leveraging only a minimal amount of target-specific data (e.g., a support set with as few as 5 DTA pairs). This collaborative mechanism enables the generation of a customized prediction model tailored to the new target while maintaining high data efficiency.

The adaptive task module effectively addresses the limitations of conventional meta-learning methods by dynamically evaluating task value and prioritizing high-value training tasks, thereby enhancing the model's learning efficiency and generalization robustness in data-scarce scenarios. Unlike typical approaches such as MAML, which rely on uniform random task sampling, often leading to inefficiency

and susceptibility to noise tasks. AdaMBind introduces a learnable adapter within its adaptive task module. This adapter dynamically assesses the value of each training task and prioritizes those tasks that most effectively enhance the model's rapid adaptation capability. By strategically selecting high-value tasks, the module ensures that the meta-learner acquires more robust and transferable initialization parameters.

In summary, through the synergistic integration of its meta-learning module and adaptive task module, AdaMBind achieves rapid adaptation to novel targets and maintains stable predictive performance under conditions of extreme data scarcity. This design enables AdaMBind to effectively address real-world scenarios, where rapid and accurate learning and prediction for novel targets can be achieved with only a minimal amount of known data (few-shot learning).

(2) Possesses practical application value under extreme settings

As elaborated in section (1), the model is designed to excel in scenarios involving novel targets with extremely limited data, a capability that has been validated by its performance in real-world applications: achieving performance outperforms methods that rely on massive pretraining data or complex structural information, but with extremely low data requirements. Our experiments on the widely recognized and challenging virtual screening benchmark LIT-PCBA strongly support this point.

To simulate the extreme data scarcity of novel targets in early-stage drug discovery, we established a strict dual-constrained training setup: (i) Extremely few meta tasks. The entire meta-training phase utilized only 13 different target tasks. (ii) Extremely small support set size. Only 15 labeled samples are provided for each task. Overall, the available samples for learning drug features, target features, and their mapping relationships to binding affinities were limited to only 13×15 instances.

As is shown in **Table R1**, under these extreme constraints, AdaMBind achieved competitive early recognition performance on test targets (e.g., TP53 and ESR): EF@1% reached approximately 4.20, and BEDROC ($\alpha = 80.5$) scores were 9.35% and 11.49%, respectively.

Table R1 Performance of AdaMBind on LIT-PCBA

	EF@1%	EF@5%	BEDROC(%)
TP53	4.1966	3.6875	9.3461
ESR	4.2080	4.3959	11.4917

To objectively position these results, we conducted a systematic literature review, comparing them with the reported performances of state-of-the-art methods on this dataset.

As shown in **Table R2**, advanced virtual screening methods (e.g., S-MolSearch_{0.9}, SaBAN¹) typically rely on hundreds of thousands of external affinity data points or protein 3D structural information for training or prediction. Their EF@1% can reach 6.33 to 7.36, with the highest BEDROC at 13.35%. In contrast, AdaMBind did not use any external large-scale pretraining data and does not require protein 3D structures. With only 15 samples per task for fine-tuning, the EF@1% achieved on TP53 (4.19) and ESR(4.20) targets is within a comparable range to methods like DrugCLIP (5.51)². Meanwhile, its BEDROC ($\alpha=80.5$) scores reached 9.34% and 11.49% respectively, further validating the model's robust early recognition capability under extreme data constraints.

Table R2 Virtual screening experimental results of LIT-PCBA reported in other literature

Methods	BEDROC (%)	EF @1%	EF @5%
DrugCLIP (SBVS ^a)	6.23	5.51	2.27
S-MolSearch _{0.4} (LBVS ^b)	7.58	6.28	2.47
S-MolSearch _{0.9} (LBVS)	8.48	7.36	3.21
SaBAN(SBVS)	13.35	6.33	3.36

Note: a: Structure-based virtual screening (SBVS). b: Ligand-based virtual screening (LBVS)

This demonstrates AdaMBind achieves virtual screening efficacy close to some methods that rely on massive data/structures, but at an extremely low data cost (15 samples per target). While its peak absolute performance still has a gap compared to the top-tier, resource-intensive pretrained methods, this gap exists under a huge difference in training data volume (15 per task vs. 600,000+) and with simpler input information (sequences/2D graphs vs. 3D structures). This precisely highlights

AdaMBind's exceptional data efficiency and powerful few-shot generalization capability.

(3) The substantial potential inherent in the AdaMBind framework

AdaMBind is not merely a validated solution under existing data-scarce conditions, but also a research framework with strong evolutionary potential, designed for the continuous exploration of lower data dependency and greater generalization capability.

A core contribution of our work lies in its model-agnostic meta-learning framework design, which establishes a critical architectural foundation for exploring more efficient learning paradigms. The model-agnostic nature of the AdaMBind framework enables seamless integration with any advanced base learner. In the future, we can readily replace the current base learner with more powerful molecular and protein representation models that possess enhanced representational capabilities. This will directly improve the model's ability to extract information from extremely few samples, thereby providing a strong representational foundation for achieving higher-performance few-shot and even zero-shot predictions.

(4) Prospects and improvements toward enabling zero-shot learning.

Building upon the foundation established by AdaMBind, we outlined subsequent research pathways aimed at systematically advancing the model toward lower data dependency and stronger generalization capability.

Specifically, we plan to conduct an in-depth exploration from the following two perspectives:

Developing novel meta-learning mechanisms for zero-shot prediction. We plan to investigate and develop a meta-learning framework capable of directly generating task-specific parameters. Unlike the current approach of learning a shared initialization applicable to all tasks, this new mechanism aims to learn the distribution of task-related model parameters. The goal is to enable the model, even in the complete absence of known affinity labels (i.e., zero supervisory signals) for a novel target, to infer customized model parameters adapted to this new task through acquired meta-knowledge. This would facilitate "plug-and-play" prediction and represents a critical step towards fundamentally reducing dependency on target-specific labeled data.

Integrating richer drug/target information and prior knowledge. We will explore how to more effectively integrate multimodal information and domain knowledge within the AdaMBind framework. This includes, but is not limited to: more refined molecular representations (e.g., bond types, 3D conformations), protein structure and functional information, and physicochemical interaction principles. By encoding such prior knowledge into the base-learner or the meta-learning process, we aim to enhance the model's ability to perform reasonable extrapolation and inference under conditions of extreme data scarcity (or even zero data), thereby improving the accuracy of its "cold-start" predictions.

We believe that the robust generalization capability demonstrated by AdaMBind with extremely few samples provides a solid empirical foundation for achieving the more challenging goals outlined above. We will continue to explore along this clear roadmap, with the ultimate aim of providing more powerful solutions for data-scarce drug discovery.

In the revised manuscript, we have emphasized the model's applicability: AdaMBind operates within a few-shot learning paradigm and still relies on known affinity values. Its goal is to address a key challenge in early-stage drug discovery: effectively leveraging the minimal preliminary experimental data available for a novel target to rapidly develop target-specific understanding and achieve reliable affinity prediction. (Line 102-107 on Page 6, Line 736-739 on Page 42) Furthermore, in the *Discussion* section, we have supplemented the analysis regarding this limitation and explored corresponding potential solutions. (764-798 on Page 44-45 of the revised manuscript)

2. Figure 2. shows the outperformance of AdaMbind as compared to other models, but difference in improvement varies in different comparisons.

Response:

Thank you for your valuable feedback. Figure 2 demonstrates that AdaMBind consistently outperforms other models across different comparative scenarios, yet the extent of performance improvement varies among these comparisons. We provide a

systematic explanation of this phenomenon from the following three perspectives: (1) the inherent differences in evaluation metrics, (2) the impact of data distribution, and (3) model training objective:

(1) The inherent differences in evaluation metrics

We observed that in the comparative experiments between AdaMBind and other baseline models, whether under the majority or few-shot settings, the model sometimes performed well on metrics such as MSE, R^2 , and Pearson, while its performance on CI and Spearman was less satisfactory. This phenomenon arises from the fundamental differences in the performance dimensions evaluated by these metrics.

Pointwise Error Metrics (MSE, R^2): These metrics directly quantify the numerical deviation between predicted and true values, serving as the most direct objectives during model optimization. Our meta-learning framework is fundamentally driven by minimizing the Mean Squared Error (MSE), which explains the substantial advancements in numerical precision.

Ranking Based Metrics (CI, Spearman, Pearson): These metrics assess the model's ability to preserve ordinal relationships among samples, but they differ in their specific focus.

Concordance Index (CI): This metric measures the global consistency between the predicted and true ordering across all comparable sample pairs. When the label distribution is highly concentrated, minor numerical adjustments are often insufficient to flip the relative order of many pairs, thereby limiting observable improvement in CI.

Spearman Rank Correlation: This coefficient measures the monotonic relationship between predictions and true values based on their ranks. It is less sensitive to the exact magnitude of errors and focuses on capturing ranking trends.

Pearson Correlation Coefficient: This measures the linear correlation between raw predicted and true values. While it can reflect ranking trends to some extent, it is primarily influenced by the covariance and variances of the datasets and is more sensitive to large absolute errors, offering a complementary perspective that emphasizes linear agreement.

In summary, the pronounced improvements in pointwise error metrics underscore

the efficacy of our framework in learning accurate numerical estimations from limited data. The more modest improvements in ranking-based metrics, particularly CI and Spearman, highlight the inherent challenge of simultaneously optimizing for precise pointwise accuracy and perfect global ranking consistency, especially under data-scarce conditions where learning a fully calibrated ordinal structure is more difficult.

(2) The Impact of Data Distribution

We visualized the distributions of the Davis, KIBA, and BindingDB datasets in **Figure R1** and quantified their concentration levels using two key metrics: the Coefficient of Variation (CV)³ and the Interquartile Range (IQR)⁴.

Figure R1. Affinity distribution across Davis, KIBA, and BindingDB datasets

Coefficient of Variation (CV): It quantifies the relative dispersion of data through the ratio of the standard deviation to the mean ($CV = \sigma / |\mu| \times 100\%$). A lower CV indicates that the overall distribution of the data is more concentrated, with smaller numerical differences between samples, reflecting stronger internal consistency within the dataset.

Interquartile Range (IQR): It reflects the spread of the central portion of the data by calculating the difference between the third quartile and the first quartile. A smaller IQR indicates that the majority of the data is clustered within a narrower numerical range, demonstrating pronounced aggregation characteristics in the distribution.

The analysis reveals that the KIBA dataset (CV = 7.14%, IQR = 0.72) and the Davis dataset (CV = 16.41%, IQR = 0.52) exhibit highly concentrated distributions, with most affinity values clustering within extremely narrow intervals. In contrast, the BindingDB dataset (CV = 23.65%) shows a relatively broader, though still moderately

concentrated distribution. These concentrated distributions, particularly in Davis and KIBA, critically impact metric evaluation: CI and Spearman exhibit high sensitivity to ranking errors in dense regions, where the true values of numerous samples are extremely close. Consequently, even small and acceptable prediction errors can lead to incorrect ranking for a large number of "comparable" sample pairs, resulting in a disproportionate negative impact on values. In contrast, other metrics are less affected. Metrics such as MSE and R^2 are relatively insensitive to minor differences between values within dense intervals. They primarily reward predictions that are close to the true values in an absolute sense. Therefore, a model can significantly improve its overall numerical fitting while still struggling to perfectly resolve subtle ranking distinctions within highly concentrated regions of the distribution.

This distribution-based analysis provides a clear explanation: substantial improvements in MSE, R^2 and Pearson can coincide with stagnation or even a slight decline in CI and Spearman. The model achieves better global numerical accuracy but faces inherent difficulty in perfectly resolving the ordinal sequence among samples with extremely close true values, a challenge amplified by the specific distribution properties of benchmark datasets like Davis and KIBA.

(3) Model training objective

The core optimization objective of our model framework is to directly minimize the error between predicted and true values, explicitly implemented by continuously reducing the MSE during training. By learning to make each prediction approximate its ground truth as closely as possible, the model prioritizes acquiring the ability to generate accurate numerical predictions swiftly. In contrast, improving metrics such as the CI and Spearman's correlation coefficient represents a more complex and indirect objective. It requires the model not only to ensure individual prediction accuracy but also to maintain correct relative ordering across all predictions. Consequently, the improvement or decline in metrics like MSE and R^2 tends to be more pronounced, whereas changes in CI and Spearman are generally more moderate.

To further bridge the gap between pointwise metrics and correlation-based metrics, we plan to enhance AdaMBind in two directions. First, we intend to introduce a dual-

loss training framework that combines MSE with a ranking-oriented loss function to jointly optimize numerical accuracy and relational consistency. Second, we will explore a weighting strategy based on meta-task similarity, which dynamically adjusts the weight coefficients of the ranking loss according to the data distribution characteristics of different target-specific tasks. This approach aims to preserve pointwise prediction accuracy while improving ranking stability in cross-target generalization (779-798 on Page 44-45 of the revised manuscript).

3. Number of unconventional abbreviations used makes it difficult to read the manuscript.

Response:

We appreciate the reviewer's feedback regarding the use of unconventional abbreviations. We reviewed the manuscript and standardized all abbreviations. Each non-standard abbreviation is now clearly defined upon its first use, and we minimized their overall number, opting for full terms where possible without sacrificing necessary technical precision. These changes have been incorporated into the revised manuscript.

4. The studies on FLT3 to identify compounds with interactions with FLT 3 is interesting, but will need testing in context of FLT3-ITD positive and FLT3-WT staus to establish its therapeutic relevance in AML.

Thank you for raising this important point. We fully agree that evaluating compound activity in the context of both FLT3-ITD mutant and FLT3 wild-type (WT) is crucial for establishing its therapeutic relevance in AML, as the FLT3-ITD mutation is a key driver and therapeutic target in this disease.

We conducted additional experiments to address this specific question. While our study demonstrated the inhibitory effect of compound Staurosporine on FLT3 signaling in cellular models, we performed enzymatic assays to directly evaluate the inhibitory activity of compound Staurosporine against the FLT3-ITD kinase. Our results show that compound Staurosporine exhibits significant inhibitory activity against the FLT3-ITD kinase ($IC_{50} = 0.465$ nM), with over 10-fold higher potency than Quizartinib ($IC_{50} = 5.7$

nM). This finding directly confirms that Staurosporine effectively targets FLT3 in AML.

Thank you for prompting us to clarify this point. We have incorporated the key finding into the revised manuscript (corresponding to Fig.9d and Line 691-708 in the revised manuscript).

Figure R2. Concentration-activity curves of Quizartinib and Staurosporine in the FLT3-IDT kinase ADP-Glo assays, respectively. Quizartinib as the positive control. Data are presented as mean \pm standard deviation (SD), n=3.

Reference

- 1 Zhou, G., Wang, Z., Yu, F. et al. S-molsearch: 3d semi-supervised contrastive learning for bioactive molecule search. *Advances in Neural Information Processing Systems* 37, 74715–74737 (2024).
- 2 Lan, J., Ding, H., Chen, H. et al. Structure-Aware Contrastive Learning with Fine-Grained Binding Representations for Drug Discovery. *arXiv preprint arXiv:2509.14788* (2025).
- 3 Abdi, H. Coefficient of variation. *Encyclopedia of Research Design* 1, 169–171 (2010).
- 4 Wan, X., Wang, W., Liu, J. et al. Estimating the sample mean and standard deviation from the sample size, median, range and/or interquartile range. *BMC Medical Research Methodology* 14, 135 (2014).